# Planning in Strawberry Fields: Evaluating and Improving the Planning and Scheduling Capabilities of LRM o1

## Abstract

The ability to plan a course of action that achieves a desired state of affairs has long been considered a core competence of intelligent agents and has been an integral part of AI research since its inception. With the advent of large language models (LLMs), there has been considerable interest in the question of whether or not they possess such planning abilities, but–despite the slew of new private and open source LLMs since GPT3–progress has remained slow. OpenAI claims that their recent o1 (Strawberry) model has been specifically constructed and trained to escape the normal limitations of autoregressive LLMs–making it a new kind of model: a Large Reasoning Model (LRM). In this paper, we evaluate the planning capabilities of two LRMs (o1-preview and o1-mini) on both planning and scheduling benchmarks. We see that while o1 does seem to offer significant improvements over autoregressive LLMs, this comes at a steep inference cost, while still failing to provide any guarantees over what it generates. We also show that combining o1 models with external verifiers–in a so-called LRM-Modulo system–guarantees the correctness of the combined system's output while further improving performance.

## 1 Introduction

The recent release of OpenAI's o1 (Strawberry) (OpenAI, 2024a) brings with it the opportunity to freshly evaluate the progress of large pre-trained AI models on planning and scheduling benchmarks. Unlike the Large Language Models (LLMs) which came before it–which can roughly be viewed as approximate retrievers–o1 seems to have been trained to be an approximate reasoner, capable of scaling the amount of compute it uses depending on the query.[1] Following OpenAI[2], we draw a distinction between previous Large Language Models (LLMs) and o1, a Large Reasoning Model (or LRM), as its new (unknown) architecture, operation, and capabilities all seem to be fundamentally different from those of vanilla LLMs, likely during both the pre-training phase and at inference time. Our aim in this paper is twofold: to comprehensively evaluate the performance of o1 on established planning and scheduling benchmarks as well as more difficult extensions; and to demonstrate how to provide correctness guarantees and boost performance by embedding an LRM in a loop with a sound verifier, in a similar vein to the LLM-Modulo framework (Kambhampati et al., 2024).

---

[1] We speculate that the complete system learns to improve its ability to make appropriate Chain-of-Thought (CoT) moves useful for reasoning in a pretraining RL step with synthetic data, and does inference time prompt-specific rollouts; see Appendix B. In other words, it may be an RL-trained system in the same vein as AlphaGo, but where the 'moves' being generated and evaluated are Chains of Thought. Note that this is very different use of RL from that in RLHF, which can be seen as a fine tuning stage for an LLM that keeps the inference step unchanged.

[2] Per the blogpost announcing the model: "A new series of reasoning models [...] for complex reasoning tasks this is a significant advancement and represents a new level of AI capability. Given this, we are resetting the counter back to 1 and naming this series OpenAI o1."(OpenAI, 2024b)

To properly evaluate this new kind of model and understand its abilities and limitations will require new tools and evaluation methods, especially if details of the overall model structure are kept secret and internal traces remain inaccessible to outside researchers.[3] In this paper, we evaluate performance on established benchmarks and compare to previous state-of-the-art results, extending these benchmarks to more difficult problems when possible and necessary. For planning, we use PlanBench (Valmeekam et al., 2023), which consists of both specific test sets and a suite of tools intended for evaluating language models on arbitrary IPC planning domains. To evaluate scheduling capabilities, we test on TravelPlanner (Xie et al., 2024), on the three domains from Natural Plan (Zheng et al., 2024), and on graph coloring problems (Stechly et al., 2024b).

Using these benchmarks as our basis for analysis, we investigate the performance jump that LRMs from the o1 family promise. We then demonstrate how these benchmarks can be elaborated on in order to remain relevant metrics for LRMs. We argue that, to be complete, new approaches to measuring LRM reasoning capabilities must take into account efficiency, cost, and guarantees. We also note the steep inference cost of LRMs and discuss the tradeoffs between using LLMs vs LRMs, arguing that in some cases an LLM-Modulo (Kambhampati et al., 2024) approach may be significantly cheaper than o1 models for comparative performance, and with guarantees. Our results also show that that same LLM-Modulo approach can indeed be adapted to LRMs to further improve their performance and provide guarantees. In essence, LRMs can replace LLMs as significantly better–but still fallible–generators in the LLM-Modulo framework.

## 2 BACKGROUND AND RELATED WORK

Though they are trained as text completion systems, Large Language Models (LLMs) have shown some promise on many other tasks. Initial claims were wildly positive, claiming they are general purpose reasoning systems (Bubeck et al., 2023), especially when prompted in just the right way (Kojima et al., 2022; Wei et al., 2022), but later results showed that their seeming reasoning capabilities are brittle and break down even in simple domains (Luo et al., 2023; Dziri et al., 2024; Stechly et al., 2024b; Qian et al., 2022) and may be attributable to dataset contamination (Roberts et al., 2023). In planning, Valmeekam et al. (2023) showed that LLMs fail even on problems as trivial as three block stacking.

Based on what little has been revealed by OpenAI, o1 seems to be a new class of model (a Large Reasoning Model or LRM), designed to combine the fuzzy language capabilities of LLMs with some implementation of approximate reasoning. With this in mind, we believe it is time to bring up the same questions that were asked about LLMs for these LRMs. We use benchmarks from the LLM literature, extending them where possible and necessary to show how well and how robustly o1 does or doesn't perform on various planning and scheduling tasks.

### 2.1 DOMAINS: PLANNING

The LLM literature abounds with claims of the 'emergent' planning capabilities of LLMs (Huang et al., 2022a). However, closer inspection reveals that many of the empirical results supporting these claims comes from evaluations on simpler, commonsense domains, such as ALFworld (Yao et al., 2023), BEHAVIOR (Srivastava et al., 2022), kitchen environments (Ahn et al., 2022; Huang et al., 2022b), and virtual home (Huang et al., 2022a). Not only do the the instances tested on tend to have fewer interactions, but many of them conflate reactive acting and deliberative planning (Ghallab et al., 2016).

In contrast, we focus on classical planning problems, or STRIPS planning problems, which are a formalism for automated planning in discrete, deterministic spaces. To define a planning problem, we specify an *initial state*, a *domain*, and a *goal*. The domain contains all relevant information about the types of objects that may exist and the allowable actions from any given state, specified by defining the preconditions and effects of each named

---

[3]There are reports that OpenAI is threatening to revoke access to o1 from anyone who tries to extract internal reasoning traces (Edwards, 2024).

| Domain | Shots | Claude Models | | OpenAI GPT-4 Models | | | | LLaMA Models | | Gemini Models | |
|---|---|---|---|---|---|---|---|---|---|---|---|
| | | Claude 3.5 (Sonnet) | Claude 3 (Opus) | GPT-4o | GPT-4o -mini | GPT-4 | GPT-4 Turbo | LLaMA 3.1 405B | LLaMA 3 70B | Gemini 1.5 Pro | Gemini 1 Pro |
| **Blocks world** | One Shot | **346/600 (57.6%)** | 289/600 (48.1%) | 170/600 (28.3%) | 49/600 (8.1%) | 206/600 (34.3%) | 138/600 (23%) | 284/600 (47.3%) | 76/600 (12.6%) | 101/600 (16.8%) | 68/600 (11.3%) |
| | Zero Shot | 329/600 (54.8%) | 356/600 (59.3%) | 213/600 (35.5%) | 53/600 (8.8%) | 210/600 (34.6%) | 241/600 (40.1%) | **376/600 (62.6%)** | 205/600 (34.16%) | 143/600 (23.8%) | 3/600 (0.5%) |
| **Mystery Blocks world** | One Shot | 19/600 (3.1%) | 8/600 (1.3%) | 5/600 (0.83%) | 0/600 (0%) | **26/600 (4.3%)** | 5/600 (0.83%) | 21/600 (3.5%) | 15/600 (2.5%) | - | 2/500 (0.4%) |
| | Zero Shot | 0/600 (0%) | 0/600 (0%) | 0/600 (0%) | 0/600 (0%) | 1/600 (0.16%) | 1/600 (0.16%) | **5/600 (0.8%)** | 0/600 (0%) | - | 0/500 (0%) |

Table 1: Performance on 600 instances from the Blocksworld and Mystery Blocksworld domains across large language models from different families, using both zero-shot and one-shot prompts. Best-in-class accuracies are bolded.

action. Problems and domains are represented in the flexible PDDL (Planning Domain and Definition Language) framework (McDermott et al., 1998). Solutions to PDDL problems are correct plans–sequences of actions executable from the initial state which arrive at a goal-satisfying final state. These are problems in which the the planner already knows all relevant facts about the world and which actions are possible–only deliberation is required.

In the current work, we extend the STRIPS planning evaluation pipeline first introduced by Valmeekam et al. (2024). This benchmark provides an extensible suite of domains and tools for evaluating general models, a collection of static test sets across these domains, and ways of easily generating additional instances of problems in these domains. We draw on these static test sets to provide our initial o1 planning results on Blocksworld–a commonsense domain from the International Planning Competition (IPC, 1998)–and Mystery Blocksworld–an obfuscated version of the same. We also extend these sets to harder instances and examine performance on problems from both Logistics and Sokoban, two other well-known IPC domains.

o1's performance on PlanBench is especially interesting in light of the fact that the benchmark remains very challenging for vanilla LLMs (massive transformer models which have been fine-tuned via RLHF). The lackluster performance of LLMs on even the easiest static test set leads us to continue to believe that planning cannot be generally and robustly solved by approximate retrieval alone. In Table 1, we present the results of running current and previous generation LLMs on a static test set of 600 three to five block Blocksworld problems, as well as on a set of 600 semantically identical but syntactically obfuscated instances which Valmeekam et al. (2024) calls Mystery Blocksworld.

Across the models tested, the best performance on regular Blocksworld is achieved by LLaMA 3.1 405B with 62.6% accuracy. Despite the underlying problems being identical, Mystery Blocksworld performance lags far behind–no LLM achieves even 5% on our test set–and performance on one version of the domain does not clearly predict performance on the other. We do not provide Mystery Blocksworld data for Gemini 1.5 Pro only because we haven't been able to generate it. The model refuses to produce any output, instead claiming that responding to these queries would be harmful. We include this output in Appendix D. Vanilla language models perform better when tested on natural language versions of prompts rather than PDDL (Valmeekam et al., 2024), even though natural language can introduce uncertainty due to polysemanticity and syntactic ambiguity. To make our comparisons "fair" for the models being tested, the results we report are the higher percent accuracy natural language prompting numbers, and can be considered an upper bound on LLM performance on PDDL-specified problems.

PlanBench does not explicitly take efficiency into consideration. As the time taken by a vanilla LLM to produce some output is only dependent on the length of that output, but otherwise independent of the semantic content or difficulty of the instance, this may not have particularly mattered in evaluations at the time. However, as LRMs adaptively vary their time taken and dollar cost per instance in response to the input, measuring efficiency

Table 2: Performance and average time taken on 600 instances from the Blocksworld, Mystery Blocksworld and Randomized Mystery Blocksworld domains and 200 instances from the Logistics and Randomized Logistics domains by OpenAI's o1 family of large reasoning models and Fast Downward (F.D.)

| Total Instances | Domain | Shots | Instances correct | | | Average Time Taken (in secs) | | |
|---|---|---|---|---|---|---|---|---|
| | | | o1-preview | o1-mini | F.D (satisficing) | o1-preview | o1-mini | F.D (satisficing) |
| 600 | Blocksworld | Zero Shot | 97.8% | 56.6% | 100% | 40.43 | 10.84 | 0.12 |
| 600 | Mystery Blocksworld | One Shot | 41.6% | - | 100% | 82.03 | - | 0.12 |
| | | Zero Shot | 52.8% | 19.1% | 100% | 83.37 | 35.54 | 0.12 |
| 600 | Randomized Mystery Blocksworld | Zero Shot | 37.3% | 3.5% | 100% | 111.11 | 55.40 | 0.12 |
| 200 | Logistics | Zero Shot | 94% | - | 100% | 84.07 | - | 0.13 |
| 200 | Randomized Mystery Logistics | Zero Shot | 52% | - | 100% | 167.41 | - | 0.13 |

has become much more important. As a comparison point between LRMs and LLMs, we compute prompting costs across models and present them in Table 4.

We also find that, contrary to previous claims, one-shot prompting is *not* a strict improvement over zero-shot. In fact, for many models it seems to do significantly worse![4] This is most notable in our tests of LLaMA family models.

We relegate additional discussion of the particulars of LLM performance to appendix A.

## 2.2 Domains: Scheduling

More recent text-based benchmarks have provided full, static descriptions of their domains, initial states, and goals. However, many of these, despite the word "planning" in their titles, would be better characterized as testing scheduling abilities (Ghallab et al., 2016). Classical planning problems are canonical graph search problems which are PSPACE-complete. Scheduling problems are only NP-Hard (Carlier, 1982), and mainly revolve around resource allocation. These problems are equivalent to constraint satisfaction problems, and thus easier than the planning problems we describe above.

We evaluate o1 on three scheduling benchmarks on which LLMs have failed. Zheng et al. (2024)'s Natural Plan benchmark consists of three scheduling domains: trip planning, calendar scheduling, and meeting planning. Xie et al. (2024)'s Travel Planning benchmark consists of a large dataset of travel information (flights, accommodations, restaurants, etc.) with prompts that ask the model to create a three to seven day itinerary based on natural language instructions. Finally, Stechly et al. (2024b) translate graph coloring, a classical constraint satisfaction problem, into natural language prompts and evaluates GPT-4's accuracy on these problems. We take their test set and extend it to more difficult instances.

## 3 From Approximate Retrieval to Approximate Reasoning

Many researchers have argued that "standard" autoregressive LLMs generate outputs via approximate retrieval, and that, while they show impressive performance on a range of System 1 tasks, they are unlikely to achieve the more System 2-like approximate reasoning capabilities that are critical for planning tasks (c.f. (Kambhampati, 2024)). From our

---

[4]While the reverse is generally true for Mystery Blocksworld problems, it's important to note that the performance of vanilla LLMs on Mystery Blocksworld has consistently and uniformly been poor (the same as it was when this benchmark was first released), so those results do not provide too clear a picture. Most models do not solve even a single instance in zero-shot mode, and only one (LLaMA 3.1 405B) manages more than one.

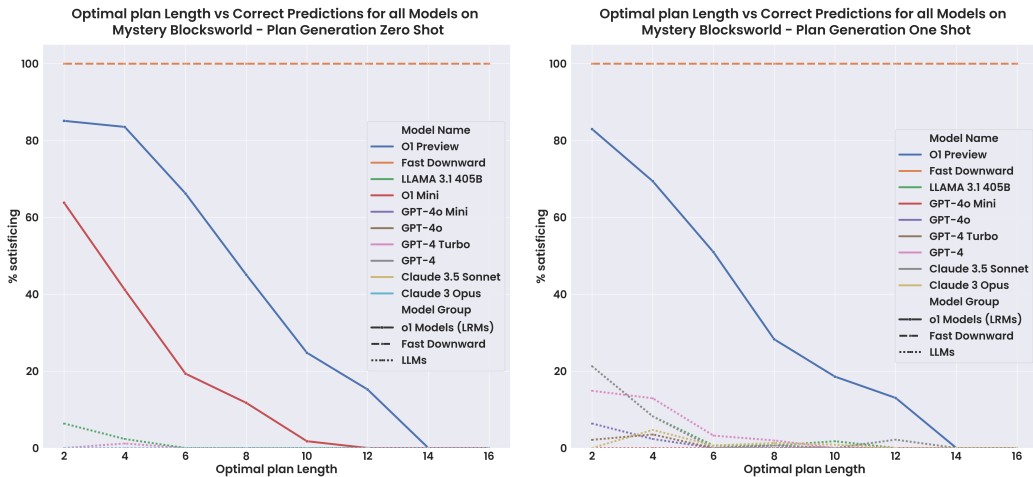

Figure 1: These examples are on Mystery Blocksworld. Fast Downward, a domain-independent planner (Helmert, 2006) solves all given instances near-instantly with guaranteed perfect accuracy. LLMs struggle on even the smallest instances. The two LRMs we tested, o1-preview and o1-mini, are surprisingly effective, but this performance is still not robust, and degrades quickly with length.

analysis, we believe that o1's architecture supplements an underlying LLM with System 2-like abilities, allowing it to outperform previous models.

As far as we can tell, o1 combines an underlying LLM, most likely a modified GPT-4o, into an RL-trained system that steers the creation, curation, and final selection of private Chain-of-Thought reasoning traces. Exact details are currently sparse, and so we can only speculate about its exact mechanisms. Our best guess is that there are two major differences between o1 and LLMs: an additional reinforcement learning pre-training phase (perhaps to learn the q-values of different CoTs from massive amounts of synthetic data) and a new adaptively scaling inference procedure (maybe it further refines learned q-values by something like rollout before selecting a particular CoT; see Appendix B. Regardless, what looks clear from the detail available is that this model is fundamentally different in nature from previous LLMs.

### 3.1 PLANNING

**Evaluating LRMs on PlanBench:** We test o1-preview and o1-mini on the original 600-instance PlanBench test set.[5] The full results can be seen in Table 2. These 600 Blocksworld instances range from three to five blocks, and require plans of between 2 to 16 steps to solve. Far surpassing any LLM, o1 correctly answers 97.8% of these instances. On Mystery Blocksworld, the model does not maintain this level of performance, but it does far surpass all previous models (which barely managed a few percent), answering 52.8% correctly. To test whether the exact obfuscation might be compromised because of data contamination, we also generated a new obfuscation using completely random strings, and presented these problems in a new, semantically equivalent prompt format with fully specified and unambiguous PDDL descriptions of both the domain and problem. This is presented in the table as Randomized Mystery Blocksworld. Exact prompts can be seen in the appendix. While performance did

---

[5]While for previous models, the model itself enforced the desired plan format, some modifications had to be made to accurately test o1's abilities. In its current form, o1-preview does not always conform to explicit formatting restrictions. This is right in line with OpenAI's injunction to keep o1 prompts "simple and direct" (OpenAI, 2024c). In order to extract the generated plans, we used GPT-4o-mini to translate them into PDDL, and wrote a small Python parser to strip any remaining extraneous symbols before evaluating each proposed plan.

Table 3: Rate of claiming that a problem is impossible by OpenAI's o1-preview on 100 unsolvable and 600 solvable instances in the Blocksworld and Randomized Mystery Blocksworld domains. The True Negative rate is the percent of unsolvable instances that were *correctly* marked as unsolvable. The False Negative rate is the percent of solvable instances that were *incorrectly* marked as unsolvable. Previous models are not shown in this table as their true negative and false negative rates were generally 0% across the board.

| Domain | Shots | o1-preview | |
|---|---|---|---|
| | | True Negatives | False Negatives |
| **Blocksworld** | 0-Shot | 27% | 0% |
| **Randomized Mystery Blocksworld** | 1-Shot | 16% | 11.5% |

dip further, 37.3% of instances are answered correctly, sharply contrasting the flat zeroes of previous models. The same pattern can be seen when evaluating Logistics and a freshly generated obfuscation of that domain. Despite the higher branching factor of the domain, o1-preview solves 94% of all 200 problems tested and achieves 52% on the obfuscated variant.

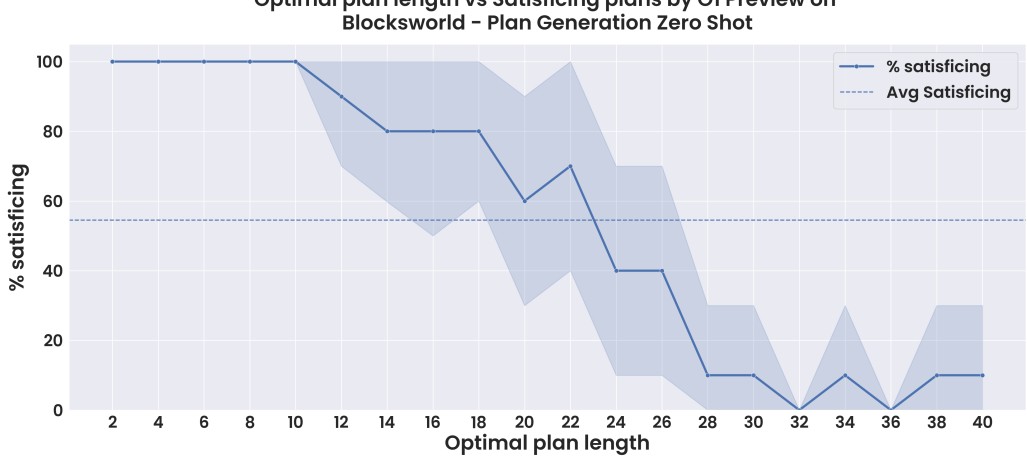

Figure 2: Extending even the (regular, not obfuscated) Blocksworld dataset to problems requiring greater numbers of steps worsens the performance of o1-preview. When tested on 110 instances which each require at least 20 steps to solve, it only manages 23.63%.

**Increasing Problem Size:** Standard LLM chain-of-thought prompting approaches are brittle, do not robustly scale with problem size, and fail to induce general algorithmic procedure-following (Stechly et al., 2024a). We extend planbench to a set of 110 harder Blocksworld problems. Problems in this set range from 6 to 20 blocks in length and require 20 to 40 step optimal plans. Without any obfuscation, we see performance quickly degrade from the 97.8% reported earlier. In fact, over these 110 instances, o1-preview only manages 23.63%, and most of this accuracy comes from correctly solving problems which require fewer than 28 steps. While these models are overall impressive, this shows that their performance is still far from robust. These results are collated together with a representative sampling of smaller instances in Figure 3, showing how performance begins to fall on problems requiring plans of more than 10 steps.

**Performance on Unsolvable Instances:** While planning problems normally require the agent to formulate a course of action to achieve a goal, an equally valid use of planning abilities is to recognize that a given goal *cannot* be accomplished by any plan. A real-world example of this is network vulnerability analysis, where an agent may wish to certify that

no plan of attack exists for a specified system (Boddy et al., 2005). So far, LLMs have struggled to recognize that some problems cannot be solved, instead confidently confabulating nonsensical answers. o1 was launched with the claim that it has started to overcome this issue, and can now accurately identify unsolvable problems (Brown, 2024). To test this systematically, we modified 100 instances from the easier three to five block test set by adding one on(x,y)-type conjunct to each instance's goal state, making the goal unsatisfiable. We ensured our instances were unambiguous by giving the full PDDL representation of both the domain and the instance, to avoid quibbles such as "A is on B because A is on C and C is on B" where the model redefines the meanings of a potentially ambiguous natural language statement. The results are in Table 3. On Blocksworld, only 27% of all instances were correctly and explicitly identified by o1 as unsolvable. In 19% of all cases, the model returned a dot or some kind of "[empty plan]" marker, without any explanation or indication of unsolvability. We consider these incorrect, as "empty plan" is only the correct answer if the goal is already satisfied. In the remaining 54% of cases, the model generated a full (and therefore impossible and incorrect!) plan.

On Randomized Mystery Blocksworld, these numbers are worse: 16% of cases were correctly identified as unsolvable, 5% returned an empty plan, and the remaining 79% were answered with a full (impossible or goal-unsatisfying) plan. Therefore, unsolvable instances continue to be a problem for LRMs. Furthermore, this ability to sometimes note impossible plans correctly comes at a cost: now the model sometimes falsely claims that solvable problems are actually unsolvable. On Randomized Mystery Blocksworld, 11.5% of instances are incorrectly claimed to be impossible. These results can be seen in Table 3.

**o1's Creative Justifications**   While our main focus has been on providing a quantitative evaluation of o1's performance on PlanBench, we have also noticed an o1 idiosyncrasy that is worth commenting on. When the model gives an incorrect answer, it also sometimes provides a creative, but nonsensical, justification for its decision. It is almost as if o1 has gone from *hallucinating* to *gaslighting*! In one case, it decided that an unsolvable problem is solvable because a goal condition, while not present in the final state, had been true for at some point during the execution, and thus should continue to count. In another, it declared that on(a,c) was true because, as it explained in a brief parenthetical, a was on b which was on c, and was thus a was somewhere above c, which should count as being "on top" of it. As we mentioned earlier, we changed our unsolvable instance prompts from natural language to PDDL in order to make it extremely clear that divergences from our exact definitions were disallowed.

**Extending to Harder Domains:**   While STRIPS planning is in principle PSPACE-complete, the problems presented above are actually of a lower computational complexity. Plan existence for both Blocksworld and Logistics is polynomial (Hoffmann et al., 2006). Sokoban is a non-ergodic domain in which an agent moves around a constrained grid, pushing boxes by running into them, where the goal is to move every box to one of a set of final locations. It can be represented in PDDL form, and is thus amenable to classical planning techniques, and is known to be PSPACE-complete (Culberson, 1997).

Using a generator from the 2008 International Planning Competition(IPC, 1998), we generate 55 Sokoban instances with grid sizes ranging from $4 \times 4$ to $10 \times 10$, 1 to 4 boxes, and 1 to 4 walls. We then create corresponding prompts in PDDL. o1-preview answers 12.7% of these instances correctly, while o1-mini is not far behind with 10.9%. For comparison, when evaluated on these same instances, Llama3.1-405B, despite doing the best of all LLMs on the Blocksword sets, does not answer a single question correctly.

### 3.2   SCHEDULING

We also evaluate o1 on a set of scheduling problems that have previously been used in testing LLM capabilities.

**Graph Coloring:**   We evaluated o1-mini on the set of 220 problems included in the codebase associated with Stechly et al. (2024b)'s paper, and found that it solved 96%, surpassing

Table 4: Cost per 100 instances (in USD). LRMs are significantly more expensive than LLMs.

| Costs per 100 instances (in USD) | | | | | | | | | |
|---|---|---|---|---|---|---|---|---|---|
| Large Language Models | | | | | | | | Large Reasoning Models | |
| Claude 3.5 (Sonnet) | Claude 3 (Opus) | GPT-4o | GPT-4o -mini | GPT-4 | GPT-4 Turbo | Gemini 1.5 Pro | Gemini 1 Pro | o1-preview | o1-mini |
| $0.44 | $1.70 | $0.65 | $0.02 | $1.80 | $1.20 | $0.33 | $0.03 | $42.12 | $3.69 |

the 16% reported by that paper for GPT-4. To test the full capabilities of the model, we extended the test set. Using the same Erdős–Rényi procedure with $p = 0.4, n = 20$, we generated 50 more graphs, and did not rejection sample for planarity. As in that paper, we precalculate the chromatic number and provide it in the prompt, asking that the model produce a coloring that uses exactly that number of colors. These harder graphs have 20 vertices and around 60 edges each. On this set, o1-mini solves 50%, while o1-preview solves 64%.

**Travel Planning:** We test o1 models on the 180 instance validation set of the sole-planning mode. In this mode, the model is provided upfront with all relevant information required to construct the requested itinerary. The previous state-of-the-art in direct prompting was 4.4%, achieved by GPT-4-turbo. o1-preview surpasses this, but only barely, solving only 10% of all instances. o1-mini does not beat even GPT-4-turbo, remaining at 1.67%.

**Natural Plan:** This benchmark consists of three domains: calendar scheduling, trip planning, and meeting planning. On calendar scheduling, o1-mini solves 94% of all instances correctly. Given cost constraints, we did not test o1-preview on this domain, as o1-mini tends to be a lower bound on its performance. Neither model performs well on trip planning– o1-preview reaches 4% and o1-mini only 1%, both lower than the previous state-of-the-art set by Gemini 1.5 Pro. When provided with prompts from the meeting planning domain, both models refuse to respond and flag the input as a potential terms violation (see Appendix E).

## 3.3 Accuracy/Cost Tradeoffs and Guarantees

With LRMs showing better performance on planning and scheduling problems, our evaluations must explicitly take into account the trade-offs that come from choosing general models over established deep and narrow systems. While o1-preview may provide higher accuracy than LLMs, it still fails to provide any correctness guarantees, and it is unclear that it is at all cost-effective. Unlike previous models, whose APIs only charge based on the number of input tokens and the number of output tokens (usually at a rate that is five times higher for the latter), o1's price-per-call includes a surcharge based on the number of "reasoning tokens" it used–tokens generated as part of inference and not revealed to the user–which are charged at the significantly higher output token rate. Currently, end users have no control over the number of these tokens generated, a number which is expanded or limited by the model in its own opaque way. We have already run up a bill of over $4000 for just the o1 model experiments reported in this evaluation![6])

Without exposing the ability to scale inference time to particular specifications, influence the internal 'thinking' process in task-specific ways, or ensure that intermediate steps are evaluated by trusted or sound verifiers, the o1 models are a coarse-grained choice in the space of cost, inference time, guarantees, and performance trade-offs. They aren't, however, the only choices in that space, and reasonable LRM evaluations must take this into account (see similar arguments in (Katz et al., 2024; Kapoor et al., 2024)).

Classical planners like Fast Downward (Helmert, 2006) achieve 100% on our dataset in a fraction of the time, compute, and cost, while providing *guarantees that their answers are correct.* Running Fast Downward on a personal computer was essentially free in dollar terms

---

[6]The rich irony of researchers using tax payer provided research funds to pay private companies like OpenAI to evaluate their private commercial models is certainly not lost on us.

Table 5: Performance of LRM-modulo with o1-preview and o1-mini as the underlying LRMs on our hardest test sets: Blocksworld (20+ length plans), Sokoban, Graph Coloring (20 vertex), Travel Planning, and Trip Planning (10 cities). Due to cost constraints, we run each problem set for up to ten iterations, stopping early if the improvement per iteration levels off.

| Total Instances | Domain | o1-preview | | | o1-mini | | |
|---|---|---|---|---|---|---|---|
| | | Direct | LRM Modulo | Iterations | Direct | LRM Modulo | Iterations |
| 110 | Blocksworld (hard) | 23.65% | 98.2% | 7 | 0.90% | 10% | 4 |
| 55 | Sokoban | 12.70% | 43.6% | 7 | 10.90% | 12.70% | 4 |
| 50 | Graph Coloring (hard) | 64% | 94% | 10 | 50% | 84% | 15 |
| 180 | Travel Planning | 10% | 65% | 10 | 1.67% | 41.11% | 10 |
| 200 | Trip Planning (10 cities) | 4% | 15.50% | 4 | 1% | 3.50% | 4 |

and averaged 0.12 seconds per instance, which is many orders of magnitude faster than the average o1 clock times listed in table 2. It is also generally predictable, and can be scaled to harder instances very directly. Vanilla LLMs are typically very good at translating problems between formats, and could be used to do so in concert with a classical planner at a fraction of the cost of LRMs (e.g. Olmo et al. (2021); Liu et al. (2023)). For problems which don't have simple PDDL domain and instance specifications, LLM-Modulo systems may be a safer and cheaper approach: run a smaller, faster LLM in a loop with a sound verifier, so that the combined system will only output guaranteed correct solutions (e.g. Kambhampati et al. (2024); Romera-Paredes et al. (2024); Trinh et al. (2024)).

The correctness guarantees provided by these latter two methods are sorely lacking in LRMs like o1. A general reasoning system cannot be deployed in safety critical and non-ergodic domains if it continues to confidently make incorrect plans. o1 is a fully black box system, even more so than previous models, and OpenAI's decision to not only keep the architecture under wraps and hide the reasoning traces, but to warn away and even ban anyone who attempts to understand what is going on inside them (Edwards, 2024), makes interpretability nearly impossible, and reduces trust in the system overall.[7]

## 4 LRM-Modulo to Improve o1 with Guarantees

We propose augmenting o1 with external verifiers to endow the combined system with soundness guarantees. While o1 is a stride in the direction of general-purpose, expressive planning systems, our results show that it cannot plan robustly when faced with harder instances, nor can it consistently recognize when instances are unsolvable, still providing incoherent plans in a majority of such cases. In other words, o1 is still fallible and without guarantees. Prior to the release of these models, the best way to coax planning capabilities out of LLMs has been to pair them with sound external verifier in generate-test frameworks, in what are known as LLM-Modulo systems (Kambhampati et al., 2024; Trinh et al., 2024). This framework is broadly applicable even beyond LLMs, and–given a sound verifier for some domain–requires only a generator expressive enough to provide guesses for that domain. Moreover, because of the built-in verification, it guarantees that any answer output is correct. For safety-critical systems, this is essential! High accuracies are not sufficient, especially when the underlying system–as is the case for both LLMs and even more so for LRMs–is an opaque black box. Therefore, we investigated integrating LRMs into LRM-modulo systems to both boost their overall performance and to provide much-needed guarantees over their outputs.

---

[7]The current model is also set to a default temperature of 1.0, which further reduces replicability and interpretability–for any given problem, it is never clear whether the result is merely the result of stochasticity. This compounds a problem with OpenAI models that has existed since at least GPT3. Temperature 0 never gave deterministic outputs, and worse, the logprobs provided by the OpenAI API for any given prompt have long been known to fluctuate wildly (Xuan, 2023).

Generate-test systems are limited by how good the generator is. A poor generator, such as one that produces completely random strings, may be capable of eventually producing the correct answer, but be so unlikely to do so at each iteration as to be useless, while an incomplete generator may never output the correct answer at all. LLMs and LRMs can be backprompted–that is, we can take feedback from the sound verifier and send it back to the model or modify the next prompt in some other way to increase the diversity of the responses generated–which may steer their next output towards the correct answer. Based on our results, o1 models are much better generators than anything that came before them, but, a priori, it is unclear if they are any more complete or capable of effectively utilizing backprompts or advice.

We test LRM-modulo setups on our five hardest test sets: 20+ length plan Blocksworld, Sokoban, 20 vertex graph coloring, OSU's Travel Planning, and 10 city trip planning. Due to cost constraints, we limit the number of iterations to a maximum of ten, but we stop the system early once the performance increase from round to round has become mostly flat.[8] Even with so few iterations, we see significant jumps in performance across almost all of our domains. o1-preview's performance on harder Blocksworld saturates within 7 iterations, with the combined system achieving 98.2%. Harder graph coloring shows similar results, going up to 94%. Perhaps most surprising, our most difficult domain, Sokoban, shows a significant jump from 12.7% to 43.6%. o1-mini-Modulo performance, while also impressive, only comes close on graph coloring and travel planning.

Our results seem to imply that these models are not only better generators, but also benefit more from the sound verification signal and provided feedback. However, we can't really know *how* they use the critique provided–this question is likely crucial, but also unanswerable with OpenAI's current stance against revealing the internal workings of the model or the intermediate reasoning tokens it generates.

With the high cost of o1 model queries, it is also crucial to examine in which situations these models are not just the best option, but the cost-effective one. Returning to LLM-modulo, where we use a smaller, cheaper, and faster LLM in a loop with a verifier, can provide similar or better performance in some domains. For example, in calendar scheduling, o1-mini costs $2.70 to run over the entire test set, and has a final accuracy of 94%. GPT-4o-mini can be run for 50 iterations in a modulo loop for only $2.48, beating that performance to get 97%, a figure that comes with guarantees not provided by just the base model.

## 5  CONCLUSION

In this paper, we investigated the performance of o1-preview and o1-mini–the new so-called LRMs–on a variety of planning and scheduling benchmarks. While LLMs have thus far failed to make much progress on the obfuscated (or "Mystery") versions of PlanBench domains, o1 shows the first bit of real progress. In general, it seems to have made impressive headway on benchmarks that were previously unassailable. However, when we evaluated the model on longer problems and on the question of determining solvability of potentially impossible instances, we found that these accuracy gains are not general nor robust. While o1 made some gains on scheduling problems, performing much better on graph coloring than previous models, these were not evenly distributed, only making some progress on OSU's Travel Plan domain and the Natural Plan benchmark suite. We also discussed the critical accuracy/efficiency tradeoffs that are brought up by the fact that o1 that uses (and charges for) significant inference-time compute, as well as how it compares to other LLM-based approaches (such as LLM-Modulo (Kambhampati et al., 2024)) and dedicated solvers. Future evaluations will have to maintain a focus on these factors if they are to remain meaningful or relevant. Finally, we showed that approaches like LLM-Modulo (Kambhampati et al., 2024) can indeed be adapted to LRMs to further improve their performance and to provide much-needed guarantees. In essence, LRMs can replace LLMs as significantly better–but still fallible–generators in LLM-Modulo frameworks.

---

[8]In the case of trip planning, we stopped early because of mounting costs: for just 4 iterations, we incurred an API access cost of $535!

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

## Appendix

### A  Further Discussion of LLM Planning Performance

LLMs are highly capable at providing translations between equivalent representations (Olmo et al., 2021). This fact, combined with their significantly higher performance on the unobfuscated version of the Blocksworld domain, predicts that–if they are capable of composing reasoning operations–the performance gap between Mystery Blocksworld and classic Blocksworld should shrink substantially if the translation from Mystery Blocksworld back into Blocksworld is explicitly provided. However, when we provide this in the prompt (see Appendix G), performance only improves a very small amount: GPT-4 achieves 10%.

### B  Speculations about o1 internal operation

While our evaluation of o1 did not depend on any specific assumption about its operation, we did have a working model of o1 based on the very skimpy description that was provided in the blog post that accompanies o1's release (OpenAI, 2024b). Verifying our model is unfortunately made infeasible by the fact that o1 doesn't actually provide any trace of its operations (even during the costly inference stage), and OpenAI warns that API access will be revoked if any attempts are made to surface its reasoning tokens.

There are two things–"reinforcement learning" and "Private Chain-of-Thought (CoT)" that are mentioned in the writeup. So imagine you are trying to transplant a "generalized AlphaGo"–let's call it GPTGo–onto the underlying LLM token prediction substrate.

To do this, you need to know

1. What are the GPTGo moves? For AlphaGo, we had GO moves). What would be the right moves when the task is just "complete the prompt the right way"?

2. Where is it getting its external success/failure signal from? For AlphaGo, we had simulators/verifiers giving the success/failure signal. The most interesting question in transplanting the self-play idea to a general AI agent is where is it getting this signal?

Our guess is that the moves are auto-generated CoTs (thus the moves have a very high branching factor). Let's assume–for simplification–that we have a CoT-generating LLM, that generates these CoTs conditioned on the prompt. (It is not clear if the CoT's are domain independent of the "think step by step" variety (Kojima et al., 2022) or domain/task specific, or a combination.)

The success signal is likely from massive amounts of synthetic training data with correct answers. When the completed prompt is seen to contain the correct answer (presumably judged by the LLM itself), then the episode is considered a success, and a failure otherwise.

The task for the reinforcement learner then is: Given the original problem prompt, generate and select a CoT, and use it to continue to extend the prompt (possibly generating subgoal CoTs after every few stages). Get the final success/failure signal for the example (for which you do have answer).

The RL stage may involve training on a a huge number of training examples with answers. The training examples with answers can either be coming from benchmarks, or from synthetic data with problems and their solutions–using external solvers. In this phase the RL part attempts to learn the q-values of the CoT moves (much like AlphaGo learns the q-values of the moves of the Go). (The q-values learning may be incorporated into the internal weights of the CoT generator LLM). At this point, we have a CoT move generator that is better than the random one before the RL stage

During the inference stage–which OpenAI says can be indefinitely long (although it is currently capped internally by them, with no external control), like AlphaGo, o1 might be further improving its evaluation of the q-values of the CoT moves in the context of the current prompt. While AlphaGo used MCT-based rollouts, we obviously don't know the

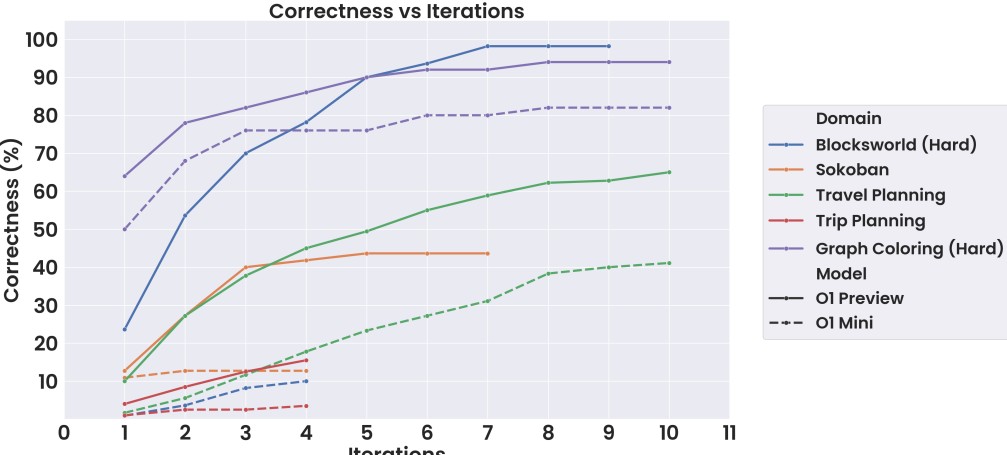

Figure 3: LRM-Modulo significantly improves performance over direct prompting as we increase the number of iterations.

mechanism o1 uses. The announcement only says that at inference stage a long chain of thought is added to the original prompt (and o1 does charge the end users for its "reasoning tokens," which are never seen by the end user, at the same high rate as the output tokens). In this sense, our speculations seem to be consistent, even though it is not clear whether the reasoning tokens are proportional to the entire inference-stage computation, or just represent the final sequence of CoT moves that get selected after the rollout-like inference stage.

Some corollaries of our speculation are:

1. Note that this use of RL is very different from that in RLHF, which can be seen as a fine tuning stage for an LLM that keeps the inference step unchanged. It is also different from techniques, including OpenAI's, that advocated fine tuning both on synthetic data accompanied with derivational traces–these too will be a form of finetuning that leave inference stage unchanged. Here o1 could, in theory, be getting significantly more leverage out of the data by learning move (auto CoT) generators.

2. There still are no guarantees that the answers provided are "correct"–they may be probabilistically a little more correct (subject to the training data). If you want guarantees, you still will need some sort of LLM-Modulo approach even on top of this.

3. It is certainly not clear that anyone will be willing to really wait for long periods of time during inference (it is already painful to wait for 10 sec for a 10 word last letter concatenation!). The kind of people who will wait for longer periods would certainly want guarantees–and there are deep and narrow System 2's a plenty that can be used for many such cases.

4. There is a bit of a *Ship of Theseus* feel to calling o1 an LLM–considering how far it is from the other LLM models (all of which essentially have teacher-forced training and sub-real-time next token prediction. That said, this is certainly an interesting way to build a generalized system 2'ish component on top of LLM substrates–but without guarantees.

## C  LRM-Modulo performance over iterations

See Figure 3

## D  Gemini 1.5 Pro Response to Mystery Blocksworld

```
finish_reason: SAFETY
```

```
safety_ratings {
  category: HARM_CATEGORY_SEXUALLY_EXPLICIT
  probability: NEGLIGIBLE
}
safety_ratings {
  category: HARM_CATEGORY_HATE_SPEECH
  probability: NEGLIGIBLE
}
safety_ratings {
  category: HARM_CATEGORY_HARASSMENT
  probability: NEGLIGIBLE
}
safety_ratings {
  category: HARM_CATEGORY_DANGEROUS_CONTENT
  probability: MEDIUM
}
```

## E  o1-preview and o1-mini Response to Meeting Planning

```
openai.BadRequestError: b'{
    "error": {
      "message": "Invalid prompt: your prompt was flagged as potentially violating
              our usage policy. Please try again with a different prompt.",
    "type": "invalid_request_error",
    "param": null,
    "code": "invalid_prompt"
  }
}'
```

## F  o1 Token Use Versus Problem Difficulty

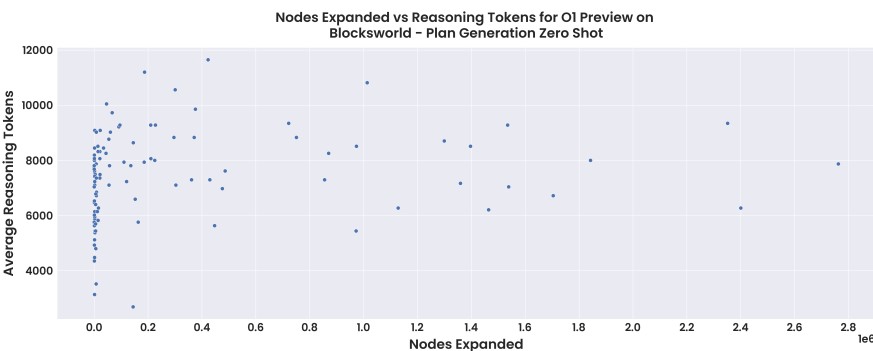

Figure 4: The number of reasoning tokens used by o1-preview when solving Blocksworld instances does not track the number of nodes that need to be expanded to solve the problem.

## G  Prompt to Translate From Mystery Back to Blocksworld

I am playing with a set of objects. Here are the actions I can do

```
    Attack object
    Feast object from another object
    Succumb object
    Overcome object from another object
```

I have the following restrictions on my actions:

To perform Attack action, the following facts need to be true: Province object, Planet object, Harmony.
Once Attack action is performed the following facts will be true: Pain object.
Once Attack action is performed the following facts will be false: Province object, Planet object, Harmony.
To perform Succumb action, the following facts need to be true: Pain object.
Once Succumb action is performed the following facts will be true: Province object, Planet object, Harmony.
Once Succumb action is performed the following facts will be false: Pain object.
To perform Overcome action, the following needs to be true: Province other object, Pain object.
Once Overcome action is performed the following will be true: Harmony, Province object, Object Craves other object.
Once Overcome action is performed the following will be false: Province other object, Pain object.
To perform Feast action, the following needs to be true: Object Craves other object, Province object, Harmony.
Once Feast action is performed the following will be true: Pain object, Province other object.
Once Feast action is performed the following will be false:, Object Craves other object, Province object, Harmony.

You will be given a set of initial conditions and a goal condition. To solve the problem, you will have to tell me which actions to take and in which order in order to achieve the goal.

Please provide your answers using the above terminology. However, you may find it helpful to translate the above description into a common-sense format while working out your solution. Just remember to translate it back later!
Instead of thinking in terms of "objects", think in terms of different alphabet blocks (block A, block B, etc.) which you are stacking (using just one hand) in towers on a table.

Then the "facts" that are true or false at a given time are really facts about the blocks and the hand:
  "Province object a" just means that "block A is clear" or, equivalently, "nothing is on top of block A"
  "Planet object a" is another way of saying "block A is on the table"
  "Harmony" is a codeword for "my hand isn't holding anything"
  "Pain object a" = "the hand is holding block A"
  "object a Craves object b" translates to "block A is on top of block B"

And the "actions" can be seen as stacking and unstacking of blocks (where the restrictions stop us from picking up the bottom block in a tower or holding more than one block in the hand at a time):
  "Attack object a" translates to "pick up block A directly from the table"
  "Feast object a from object b" translates to "pick up block A from directly on top of block B"
  "Succumb object a" translates to "put block A directly on the table"
  "Overcome object a from object b" translates to "put block A directly on top of block B"

If you do use this framing, make sure to remember to translate back into the original terms.

# H   PROMPTS FOR BLOCKSWORLD

## H.1   (SOLVABLE) BLOCKSWORLD INSTANCES - ZERO-SHOT IN NATURAL LANGUAGE

I am playing with a set of blocks where I need to arrange the blocks into stacks. Here are the actions I can do

Pick up a block

Unstack a block from on top of another block
Put down a block
Stack a block on top of another block

I have the following restrictions on my actions:
I can only pick up or unstack one block at a time.
I can only pick up or unstack a block if my hand is empty.
I can only pick up a block if the block is on the table and the block is clear. A
    block is clear if the block has no other blocks on top of it and if the block
    is not picked up.
I can only unstack a block from on top of another block if the block I am unstacking
        was really on top of the other block.
I can only unstack a block from on top of another block if the block I am unstacking
        is clear.
Once I pick up or unstack a block, I am holding the block.
I can only put down a block that I am holding.
I can only stack a block on top of another block if I am holding the block being
    stacked.
I can only stack a block on top of another block if the block onto which I am
    stacking the block is clear.
Once I put down or stack a block, my hand becomes empty.
Once you stack a block on top of a second block, the second block is no longer clear
    .

[STATEMENT]
As initial conditions I have that, the red block is clear, the blue block is clear,
    the yellow block is clear, the hand is empty, the blue block is on top of the
    orange block, the red block is on the table, the orange block is on the table
    and the yellow block is on the table.
My goal is to have that the orange block is on top of the blue block.

What is the plan to achieve my goal? Just give the actions in the plan.

=============o1-preview RESPONSE================
To achieve your goal of stacking the orange block on top of the blue block, you can
    follow these steps:

1. **Unstack the blue block from on top of the orange block.**
   - Since the blue block is clear and your hand is empty, you can unstack it from
       the orange block.
2. **Put down the blue block on the table.**
   - With the blue block in your hand, you can place it onto the table.
3. **Pick up the orange block from the table.**
   - The orange block is now clear (no blocks on top of it), and your hand is empty
       after putting down the blue block, so you can pick it up.
4. **Stack the orange block on top of the blue block.**
   - The blue block is clear on the table, so you can stack the orange block you're
       holding onto it.

**Actions in the plan:**

- **Unstack** the blue block from the orange block.
- **Put down** the blue block on the table.
- **Pick up** the orange block from the table.
- **Stack** the orange block on top of the blue block.

H.2   (Solvable) Harder Blocksworld Instances - Zero-Shot in PDDL

The following is a PDDL specification of a planning problem. The first part, under
    the heading [DOMAIN], is the domain file. The second part, under the heading [
    QUERY PROBLEM], is the problem file. Using this information, which is correct,
    and no further assumptions, find a plan which, when run from the specified
    initial state, satisfies the specified goal. Provide your answer as a sequence

```
972        of actions in PDDL format. An action ACTION which acts on two objects OBJ1 and
973        OBJ2 would be written (ACTION OBJ1 OBJ2). Do not provide anything else in your
974        answer.
975   [DOMAIN]
976   (define (domain blocksworld-4ops)
977     (:requirements :strips)
978   (:predicates (clear ?x)
                   (ontable ?x)
979                (handempty)
980                (holding ?x)
981                (on ?x ?y))
982
983   (:action pick-up
984     :parameters (?ob)
985     :precondition (and (clear ?ob) (ontable ?ob) (handempty))
986     :effect (and (holding ?ob) (not (clear ?ob)) (not (ontable ?ob))
                   (not (handempty))))
987
988   (:action put-down
989     :parameters (?ob)
990     :precondition (holding ?ob)
991     :effect (and (clear ?ob) (handempty) (ontable ?ob)
                   (not (holding ?ob))))
992
993   (:action stack
994     :parameters (?ob ?underob)
995     :precondition (and (clear ?underob) (holding ?ob))
996     :effect (and (handempty) (clear ?ob) (on ?ob ?underob)
                   (not (clear ?underob)) (not (holding ?ob))))
997
998   (:action unstack
999     :parameters (?ob ?underob)
1000    :precondition (and (on ?ob ?underob) (clear ?ob) (handempty))
1001    :effect (and (holding ?ob) (clear ?underob)
                   (not (on ?ob ?underob)) (not (clear ?ob)) (not (handempty)))))
1002
1003  [QUERY PROBLEM]
1004  (define(problem BW-rand-6)
1005  (:domain blocksworld-4ops)
1006  (:objects a b c d e f )
1007  (:init
1008  (handempty)
1009  (ontable a)
1010  (on b f)
1011  (on c a)
1012  (on d b)
1013  (on e c)
1014  (on f e)
1015  (clear d)
1016  )
1017  (:goal
1018  (and
1019  (on a f)
1020  (on b c)
1021  (on c a)
1022  (on d b)
1023  (on f e))
      )
1024  )
1025  )

      [PLAN]

      =============o1-preview RESPONSE===============
      (unstack d b)
```

```
(put-down d)
(unstack b f)
(stack b c)
(pick-up d)
(stack d b)
(unstack f e)
(put-down f)
(unstack e c)
(put-down e)
(pick-up f)
(stack f e)
(unstack c a)
(put-down c)
(pick-up a)
(stack a f)
(pick-up c)
(stack c a)
(pick-up b)
(stack b c)
(pick-up d)
(stack d b)
```

## H.3   (Solvable) Harder Blocksworld Instances - Backprompt

```
The above plan is invalid. The following action at step 4 has an unmet precondition:
stack b c
The unmet precondition is:
(clear c)
Provide your revised plan as a sequence of actions in PDDL format. An action ACTION
    which acts on two objects OBJ1 and OBJ2 would be written (ACTION OBJ1 OBJ2). Do
     not provide anything else in your answer.

[PLAN]
```

## H.4   Unsolvable Blocksworld Instances - Zero-Shot in PDDL

```
The following is a PDDL specification of a planning problem. The first part, under
    the heading [DOMAIN], is the domain file. The second part, under the heading [
    QUERY PROBLEM], is the problem file. Using this information, which is correct,
    and no further assumptions, find a plan which, when run from the specified
    initial state, satisfies the specified goal. Provide your answer as a sequence
    of actions in PDDL format. An action ACTION which acts on two objects OBJ1 and
    OBJ2 would be written (ACTION OBJ1 OBJ2). Do not provide anything else in your
    answer.
[DOMAIN]
(define (domain blocksworld-4ops)
  (:requirements :strips)
(:predicates (clear ?x)
             (ontable ?x)
             (handempty)
             (holding ?x)
             (on ?x ?y))

(:action pick-up
  :parameters (?ob)
  :precondition (and (clear ?ob) (ontable ?ob) (handempty))
  :effect (and (holding ?ob) (not (clear ?ob)) (not (ontable ?ob))
               (not (handempty))))

(:action put-down
  :parameters (?ob)
  :precondition (holding ?ob)
  :effect (and (clear ?ob) (handempty) (ontable ?ob)
```

```
1080                  (not (holding ?ob))))
1081
1082    (:action stack
1083      :parameters (?ob ?underob)
1084      :precondition (and (clear ?underob) (holding ?ob))
1085      :effect (and (handempty) (clear ?ob) (on ?ob ?underob)
1086                  (not (clear ?underob)) (not (holding ?ob))))
1087    (:action unstack
1088      :parameters (?ob ?underob)
1089      :precondition (and (on ?ob ?underob) (clear ?ob) (handempty))
1090      :effect (and (holding ?ob) (clear ?underob)
1091                  (not (on ?ob ?underob)) (not (clear ?ob)) (not (handempty)))))
1092    [QUERY PROBLEM]
1093    (define(problem BW-rand-4)
1094    (:domain blocksworld-4ops)
1095    (:objects a b c d )
1096    (:init
1097    (handempty)
1098    (on a b)
1099    (on b c)
1100    (on c d)
1101    (ontable d)
1102    (clear a)
1103    )
1104    (:goal
1105        (and
1106                (on d c)
1107                (on c d)
1108        )
1109    )
1110    )
1111
1112    [PLAN]
1113    =============o1-preview RESPONSE===============
1114    **No plan exists that satisfies the goal with the given initial state and domain.**
```

# I  PROMPTS FOR MYSTERY BLOCKSWORLD

## I.1  MYSTERY BLOCKSWORLD INSTANCES - ZERO-SHOT IN NATURAL LANGUAGE

I am playing with a set of objects. Here are the actions I can do

   Attack object
   Feast object from another object
   Succumb object
   Overcome object from another object

I have the following restrictions on my actions:
   To perform Attack action, the following facts need to be true: Province object,
       Planet object, Harmony.
   Once Attack action is performed the following facts will be true: Pain object.
   Once Attack action is performed the following facts will be false: Province
       object, Planet object, Harmony.
   To perform Succumb action, the following facts need to be true: Pain object.
   Once Succumb action is performed the following facts will be true: Province
       object, Planet object, Harmony.
   Once Succumb action is performed the following facts will be false: Pain object.
   To perform Overcome action, the following needs to be true: Province other
       object, Pain object.
   Once Overcome action is performed the following will be true: Harmony, Province
       object, Object Craves other object.

Once Overcome action is performed the following will be false: Province other
            object, Pain object.
        To perform Feast action, the following needs to be true: Object Craves other
            object, Province object, Harmony.
        Once Feast action is performed the following will be true: Pain object, Province
            other object.
        Once Feast action is performed the following will be false:, Object Craves other
            object, Province object, Harmony.

[STATEMENT]
As initial conditions I have that, object b craves object c, harmony, planet object
    a, planet object c, planet object d, province object a, province object b and
    province object d.
My goal is to have that object c craves object b.

What is the plan to achieve my goal? Just give the actions in the plan.
==============o1-preview RESPONSE================
To achieve the goal where object c craves object b, you can perform the following
    sequence of actions:

1. **Feast object b from object c**
   - Preconditions:
     - `Object b craves object c` (initial fact)
     - `Province object b` (initial fact)
     - `Harmony` (initial fact)
   - Effects:
     - Adds: `Pain object b`, `Province object c`
     - Deletes: `Object b craves object c`, `Province object b`, `Harmony`

2. **Succumb object b**
   - Preconditions:
     - `Pain object b` (from previous action)
   - Effects:
     - Adds: `Province object b`, `Planet object b`, `Harmony`
     - Deletes: `Pain object b`

3. **Attack object c**
   - Preconditions:
     - `Province object c` (from step 1)
     - `Planet object c` (initial fact)
     - `Harmony` (restored in step 2)
   - Effects:
     - Adds: `Pain object c`
     - Deletes: `Province object c`, `Planet object c`, `Harmony`

4. **Overcome object c from object b**
   - Preconditions:
     - `Province object b` (restored in step 2)
     - `Pain object c` (from step 3)
   - Effects:
     - Adds: `Harmony`, `Province object c`, `Object c craves object b` (goal
         achieved)
     - Deletes: `Province object b`, `Pain object c`

By following this plan, you manipulate the states of the objects through the actions
    provided, ultimately achieving the desired state where `Object c craves object
    b`.

**Answer:**
To achieve the goal, perform these actions in order:

1. Feast object b from object c
2. Succumb object b
3. Attack object c
4. Overcome object c from object b

## J   Prompts for Randomized Mystery Blocksworld

### J.1   Randomized Mystery Blocksworld Instances - Zero-Shot in Natural Language

I am playing with a set of objects. Here are the actions I can do

J4gv801gnu2it0yj object_0 object_1.
U64y1a9apusmslxb object_0.
Kip9uw781pv62umn object_0.
Wio5amhq7814n006 object_0 object_1.

I have the following restrictions on my actions:
To perform j4gv801gnu2it0yj action, the following facts need to be true:
    tv30k33pzoulql6w object_1, and a46zhwykn6jvbabk object_0
Once j4gv801gnu2it0yj is performed the following facts will be true:
    cql4o62p1yeke3ok, tv30k33pzoulql6w object_0, and b6e9q4r60gagvdcn object_0
    object_1
Once j4gv801gnu2it0yj is performed the following facts will be false:
    tv30k33pzoulql6w object_1, and a46zhwykn6jvbabk object_0
To perform u64y1a9apusmslxb action, the following facts need to be true:
    tv30k33pzoulql6w object_0, af9tse23ljclsqad object_0, and cql4o62p1yeke3ok
Once u64y1a9apusmslxb is performed the following facts will be true:
    a46zhwykn6jvbabk object_0
Once u64y1a9apusmslxb is performed the following facts will be false:
    tv30k33pzoulql6w object_0, af9tse23ljclsqad object_0, and cql4o62p1yeke3ok
To perform kip9uw781pv62umn action, the following facts need to be true:
    a46zhwykn6jvbabk object_0
Once kip9uw781pv62umn is performed the following facts will be true:
    tv30k33pzoulql6w object_0, cql4o62p1yeke3ok, and af9tse23ljclsqad object_0
Once kip9uw781pv62umn is performed the following facts will be false:
    a46zhwykn6jvbabk object_0
To perform wio5amhq7814n006 action, the following facts need to be true:
    b6e9q4r60gagvdcn object_0 object_1, tv30k33pzoulql6w object_0, and
    cql4o62p1yeke3ok
Once wio5amhq7814n006 is performed the following facts will be true:
    a46zhwykn6jvbabk object_0, and tv30k33pzoulql6w object_1
Once wio5amhq7814n006 is performed the following facts will be false:
    b6e9q4r60gagvdcn object_0 object_1, tv30k33pzoulql6w object_0, and
    cql4o62p1yeke3ok

[STATEMENT]
As initial conditions I have that, af9tse23ljclsqad object_0, af9tse23ljclsqad
    object_2, af9tse23ljclsqad object_3, b6e9q4r60gagvdcn object_1 object_2,
    cql4o62p1yeke3ok, tv30k33pzoulql6w object_0, tv30k33pzoulql6w object_1 and
    tv30k33pzoulql6w object_3.
My goal is to have that b6e9q4r60gagvdcn object_2 object_1.

What is the plan to achieve my goal? Just give the actions in the plan.
===============o1-preview RESPONSE================
To achieve the goal `b6e9q4r60gagvdcn object_2 object_1`, you need to perform the
    following sequence of actions:

1. **wio5amhq7814n006 object_1 object_2**
   - This action reverses the relationship between `object_1` and `object_2`,
       setting up the necessary conditions for the next steps.
2. **kip9uw781pv62umn object_1**
   - By performing this action on `object_1`, you restore certain facts (`
       cql4o62p1yeke3ok`) required for subsequent actions and adjust the state of `
       object_1`.
3. **u64y1a9apusmslxb object_2**
   - This action modifies `object_2` to have the required property `a46zhwykn6jvbabk
        object_2`, which is a precondition for the final action.
4. **j4gv801gnu2it0yj object_2 object_1**

- Performing this action with `object_2` and `object_1` establishes the goal fact
    `b6e9q4r60gagvdcn object_2 object_1`.

**Plan Summary:**

- **Action 1:** `wio5amhq7814n006 object_1 object_2`
- **Action 2:** `kip9uw781pv62umn object_1`
- **Action 3:** `u64y1a9apusmslxb object_2`
- **Action 4:** `j4gv801gnu2it0yj object_2 object_1`

This sequence of actions satisfies all the necessary preconditions step-by-step and
    ultimately achieves your goal.

## J.2 Unsolvable Randomized Mystery Blocksworld Instances - Zero-Shot in PDDL

The following is a PDDL specification of a planning problem. The first part, under
    the heading [DOMAIN], is the domain file. The second part, under the heading [
    QUERY PROBLEM], is the problem file. Using this information, which is correct,
    and no further assumptions, find a plan which, when run from the specified
    initial state, satisfies the specified goal. Provide your answer as a sequence
    of actions in PDDL format. An action ACTION which acts on two objects OBJ1 and
    OBJ2 would be written (ACTION OBJ1 OBJ2). Do not provide anything else in your
    answer.
[DOMAIN]
```
(define (domain xaji0y)
    (:requirements :equality)
    (:predicates
        (tv30k33pzoulql6w ?x1 - object)
        (af9tse23ljclsqad ?x1 - object)
        (cql4o62p1yeke3ok )
        (a46zhwykn6jvbabk ?x1 - object)
        (b6e9q4r60gagvdcn ?x1 - object ?x2 - object)
    )

    (:action u64y1a9apusmslxb
     :parameters (?ob1 - object)
     :precondition (and (tv30k33pzoulql6w ?ob1) (af9tse23ljclsqad ?ob1) (
        cql4o62p1yeke3ok ))
     :effect (and
        (a46zhwykn6jvbabk ?ob1)
        (not (tv30k33pzoulql6w ?ob1))
        (not (af9tse23ljclsqad ?ob1))
        (not (cql4o62p1yeke3ok )))
    )

    (:action kip9uw781pv62umn
     :parameters (?ob1 - object)
     :precondition (a46zhwykn6jvbabk ?ob1)
     :effect (and
        (tv30k33pzoulql6w ?ob1)
        (cql4o62p1yeke3ok )
        (af9tse23ljclsqad ?ob1)
        (not (a46zhwykn6jvbabk ?ob1)))
    )

    (:action j4gv801gnu2it0yj
     :parameters (?ob1 - object ?ob2 - object)
     :precondition (and (tv30k33pzoulql6w ?ob2) (a46zhwykn6jvbabk ?ob1))
     :effect (and
        (cql4o62p1yeke3ok )
        (tv30k33pzoulql6w ?ob1)
```

```
        (b6e9q4r60gagvdcn ?ob1 ?ob2)
        (not (tv30k33pzoulql6w ?ob2))
        (not (a46zhwykn6jvbabk ?ob1)))
    )

    (:action wio5amhq7814n006
     :parameters (?ob1 - object ?ob2 - object)
     :precondition (and (b6e9q4r60gagvdcn ?ob1 ?ob2) (tv30k33pzoulql6w ?ob1) (
         cql4o62p1yeke3ok ))
     :effect (and
        (a46zhwykn6jvbabk ?ob1)
        (tv30k33pzoulql6w ?ob2)
        (not (b6e9q4r60gagvdcn ?ob1 ?ob2))
        (not (tv30k33pzoulql6w ?ob1))
        (not (cql4o62p1yeke3ok )))
    )

)

[QUERY PROBLEM]
(define(problem instance-1-4)
(:domain xaji0y)
(:objects a b c d )
(:init
(cql4o62p1yeke3ok)
(b6e9q4r60gagvdcn a d)
(af9tse23ljclsqad b)
(b6e9q4r60gagvdcn c a)
(b6e9q4r60gagvdcn d b)
(tv30k33pzoulql6w c)
)
(:goal
      (and
              (b6e9q4r60gagvdcn a d)
              (b6e9q4r60gagvdcn b a)
              (b6e9q4r60gagvdcn b d)
      )
)
)

[PLAN]
==============o1-preview RESPONSE===============
There is no sequence of actions that, when run from the specified initial state,
    satisfies the specified goal. Thus, no plan can be provided.
```

## K  Prompts for Logistics

### K.1  Logistics - Zero-Shot in PDDL

The following is a PDDL specification of a planning problem. The first part, under
    the heading [DOMAIN], is the domain file. The second part, under the heading [
    QUERY PROBLEM], is the problem file. Using this information, which is correct,
    and no further assumptions, find a plan which, when run from the specified
    initial state, satisfies the specified goal. Provide your answer as a sequence
    of actions in PDDL format. An action ACTION which acts on two objects OBJ1 and
    OBJ2 would be written (ACTION OBJ1 OBJ2). Do not provide anything else in your
    answer.
[DOMAIN]
```
(define (domain logistics-strips)
  (:requirements :strips)
  (:predicates (OBJ ?obj)
               (TRUCK ?truck)
               (LOCATION ?loc)
```

```
                       (AIRPLANE ?airplane)
                       (CITY ?city)
                       (AIRPORT ?airport)
                       (at ?obj ?loc)
                       (in ?obj1 ?obj2)
                       (in-city ?obj ?city))

  ; (:types )          ; default object

(:action LOAD-TRUCK
   :parameters
    (?obj
     ?truck
     ?loc)
   :precondition
    (and (OBJ ?obj) (TRUCK ?truck) (LOCATION ?loc)
    (at ?truck ?loc) (at ?obj ?loc))
   :effect
    (and (not (at ?obj ?loc)) (in ?obj ?truck)))

(:action LOAD-AIRPLANE
   :parameters
    (?obj
     ?airplane
     ?loc)
   :precondition
    (and (OBJ ?obj) (AIRPLANE ?airplane) (LOCATION ?loc)
    (at ?obj ?loc) (at ?airplane ?loc))
   :effect
    (and (not (at ?obj ?loc)) (in ?obj ?airplane)))

(:action UNLOAD-TRUCK
   :parameters
    (?obj
     ?truck
     ?loc)
   :precondition
    (and (OBJ ?obj) (TRUCK ?truck) (LOCATION ?loc)
         (at ?truck ?loc) (in ?obj ?truck))
   :effect
    (and (not (in ?obj ?truck)) (at ?obj ?loc)))

(:action UNLOAD-AIRPLANE
   :parameters
    (?obj
     ?airplane
     ?loc)
   :precondition
    (and (OBJ ?obj) (AIRPLANE ?airplane) (LOCATION ?loc)
         (in ?obj ?airplane) (at ?airplane ?loc))
   :effect
    (and (not (in ?obj ?airplane)) (at ?obj ?loc)))

(:action DRIVE-TRUCK
   :parameters
    (?truck
     ?loc-from
     ?loc-to
     ?city)
   :precondition
    (and (TRUCK ?truck) (LOCATION ?loc-from) (LOCATION ?loc-to) (CITY ?city)
    (at ?truck ?loc-from)
    (in-city ?loc-from ?city)
```

```
     (in-city ?loc-to ?city))
    :effect
    (and (not (at ?truck ?loc-from)) (at ?truck ?loc-to)))

  (:action FLY-AIRPLANE
   :parameters
    (?airplane
     ?loc-from
     ?loc-to)
   :precondition
    (and (AIRPLANE ?airplane) (AIRPORT ?loc-from) (AIRPORT ?loc-to)
         (at ?airplane ?loc-from))
   :effect
    (and (not (at ?airplane ?loc-from)) (at ?airplane ?loc-to)))
  )

  [QUERY PROBLEM]
  (define(problem logistics-c2-s1-p1-a2)
  (:domain logistics-strips)
  (:objects a0 a1
          c0 c1
          t0 t1
          l0-0 l1-0
          p0
  )
  (:init
     (AIRPLANE a0)
     (AIRPLANE a1)
     (CITY c0)
     (CITY c1)
     (TRUCK t0)
     (TRUCK t1)
     (LOCATION l0-0)
     (in-city l0-0 c0)
     (LOCATION l1-0)
     (in-city l1-0 c1)
     (AIRPORT l0-0)
     (AIRPORT l1-0)
     (OBJ p0)
     (at t0 l0-0)
     (at t1 l1-0)
     (at p0 l1-0)
     (at a0 l0-0)
     (at a1 l0-0)
  )
  (:goal
     (and
        (at p0 l0-0)
     )
  )
  )
  )

  [PLAN]
  =============o1-preview RESPONSE===============
  (FLY-AIRPLANE a0 l0-0 l1-0)

  (LOAD-AIRPLANE p0 a0 l1-0)

  (FLY-AIRPLANE a0 l1-0 l0-0)

  (UNLOAD-AIRPLANE p0 a0 l0-0)
```

K.2   RANDOMIZED LOGISTICS - ZERO-SHOT IN PDDL

The following is a PDDL specification of a planning problem. The first part, under the heading [DOMAIN], is the domain file. The second part, under the heading [QUERY PROBLEM], is the problem file. Using this information, which is correct, and no further assumptions, find a plan which, when run from the specified initial state, satisfies the specified goal. Provide your answer as a sequence of actions in PDDL format. An action ACTION which acts on two objects OBJ1 and OBJ2 would be written (ACTION OBJ1 OBJ2). Do not provide anything else in your answer.

[DOMAIN]
```
(define (domain dxpflc3up53obvi)
    (:requirements :strips)
    (:predicates (a46zhwykn6jvbabk ?airport) (abwaom5mhyf979wg ?obj1 ?obj2) (
        b6e9q4r60gagvdcn ?city) (cql4o62p1yeke3ok ?airplane) (dvr5qjofrygv9zlx ?obj
        ?loc) (fe8cgp3y92p6lb85 ?loc) (j3bxzst7l100cyne ?obj ?city) (
        lwiiiuk750o3u28e ?obj) (rvgqqfhogwl9yspo ?truck))
    (:action af9tse23ljclsqad
        :parameters (?ob_1 ?ob_2 ?ob_3 ?ob_4)
        :precondition (and (rvgqqfhogwl9yspo ?ob_1) (fe8cgp3y92p6lb85 ?ob_2) (
            fe8cgp3y92p6lb85 ?ob_3) (b6e9q4r60gagvdcn ?ob_4) (dvr5qjofrygv9zlx ?ob_1
             ?ob_2) (j3bxzst7l100cyne ?ob_2 ?ob_4) (j3bxzst7l100cyne ?ob_3 ?ob_4))
        :effect (and (not (dvr5qjofrygv9zlx ?ob_1 ?ob_2)) (dvr5qjofrygv9zlx ?ob_1 ?
            ob_3))
    )
     (:action j4gv801gnu2it0yj
        :parameters (?ob_1 ?ob_2 ?ob_3)
        :precondition (and (lwiiiuk750o3u28e ?ob_1) (rvgqqfhogwl9yspo ?ob_2) (
            fe8cgp3y92p6lb85 ?ob_3) (dvr5qjofrygv9zlx ?ob_2 ?ob_3) (abwaom5mhyf979wg
             ?ob_1 ?ob_2))
        :effect (and (not (abwaom5mhyf979wg ?ob_1 ?ob_2)) (dvr5qjofrygv9zlx ?ob_1 ?
            ob_3))
    )
     (:action kip9uw781pv62umn
        :parameters (?ob_1 ?ob_2 ?ob_3)
        :precondition (and (lwiiiuk750o3u28e ?ob_1) (cql4o62p1yeke3ok ?ob_2) (
            fe8cgp3y92p6lb85 ?ob_3) (dvr5qjofrygv9zlx ?ob_1 ?ob_3) (dvr5qjofrygv9zlx
             ?ob_2 ?ob_3))
        :effect (and (not (dvr5qjofrygv9zlx ?ob_1 ?ob_3)) (abwaom5mhyf979wg ?ob_1 ?
            ob_2))
    )
     (:action tv30k33pzoulql6w
        :parameters (?ob_1 ?ob_2 ?ob_3)
        :precondition (and (cql4o62p1yeke3ok ?ob_1) (a46zhwykn6jvbabk ?ob_2) (
            a46zhwykn6jvbabk ?ob_3) (dvr5qjofrygv9zlx ?ob_1 ?ob_2))
        :effect (and (not (dvr5qjofrygv9zlx ?ob_1 ?ob_2)) (dvr5qjofrygv9zlx ?ob_1 ?
            ob_3))
    )
     (:action u64y1a9apusmslxb
        :parameters (?ob_1 ?ob_2 ?ob_3)
        :precondition (and (lwiiiuk750o3u28e ?ob_1) (rvgqqfhogwl9yspo ?ob_2) (
            fe8cgp3y92p6lb85 ?ob_3) (dvr5qjofrygv9zlx ?ob_2 ?ob_3) (dvr5qjofrygv9zlx
             ?ob_1 ?ob_3))
        :effect (and (not (dvr5qjofrygv9zlx ?ob_1 ?ob_3)) (abwaom5mhyf979wg ?ob_1 ?
            ob_2))
    )
     (:action wio5amhq7814n006
        :parameters (?ob_1 ?ob_2 ?ob_3)
        :precondition (and (lwiiiuk750o3u28e ?ob_1) (cql4o62p1yeke3ok ?ob_2) (
            fe8cgp3y92p6lb85 ?ob_3) (abwaom5mhyf979wg ?ob_1 ?ob_2) (dvr5qjofrygv9zlx
             ?ob_2 ?ob_3))
        :effect (and (not (abwaom5mhyf979wg ?ob_1 ?ob_2)) (dvr5qjofrygv9zlx ?ob_1 ?
            ob_3))
    )
)
```

[QUERY PROBLEM]

```
(define(problem vd24g3dfuion115)
    (:domain dxpflc3up53obvi)
    (:objects o1 o2 o3 o4 o5 o6 o7 o8 o9)
    (:init (a46zhwykn6jvbabk o4) (a46zhwykn6jvbabk o7) (b6e9q4r60gagvdcn o1) (
        b6e9q4r60gagvdcn o9) (cql4o62p1yeke3ok o2) (cql4o62p1yeke3ok o6) (
        dvr5qjofrygv9zlx o2 o7) (dvr5qjofrygv9zlx o3 o4) (dvr5qjofrygv9zlx o5 o4) (
        dvr5qjofrygv9zlx o6 o7) (dvr5qjofrygv9zlx o8 o7) (fe8cgp3y92p6lb85 o4) (
        fe8cgp3y92p6lb85 o7) (j3bxzst7l100cyne o4 o9) (j3bxzst7l100cyne o7 o1) (
        lwiiiuk750o3u28e o5) (rvgqqfhogwl9yspo o3) (rvgqqfhogwl9yspo o8))
    (:goal (dvr5qjofrygv9zlx o5 o7))
)

[PLAN]
==============o1-preview RESPONSE===============
(tv30k33pzoulql6w o2 o7 o4)
(kip9uw781pv62umn o5 o2 o4)
(tv30k33pzoulql6w o2 o4 o7)
(wio5amhq7814n006 o5 o2 o7)
```

## L   PROMPTS FOR SOKOBAN

### L.1   SOKOBAN - ZERO-SHOT IN PDDL

The following is a PDDL specification of a planning problem. The first part, under
    the heading [DOMAIN], is the domain file. The second part, under the heading [
    QUERY PROBLEM], is the problem file. Using this information, which is correct,
    and no further assumptions, find a plan which, when run from the specified
    initial state, satisfies the specified goal. Provide your answer as a sequence
    of actions in PDDL format. An action ACTION which acts on two objects OBJ1 and
    OBJ2 would be written (ACTION OBJ1 OBJ2). Do not provide anything else in your
    answer.

```
[DOMAIN]
(define (domain typed-sokoban)
(:requirements :typing)
(:types LOC DIR BOX)
(:predicates
        (at-robot ?l - LOC)
        (at ?o - BOX ?l - LOC)
        (adjacent ?l1 - LOC ?l2 - LOC ?d - DIR)
        (clear ?l - LOC)
)

(:action move
:parameters (?from - LOC ?to - LOC ?dir - DIR)
:precondition (and (clear ?to) (at-robot ?from) (adjacent ?from ?to ?dir))
:effect (and (at-robot ?to) (not (at-robot ?from)))
)

(:action push
:parameters (?rloc - LOC ?bloc - LOC ?floc - LOC ?dir - DIR ?b - BOX)
:precondition (and (at-robot ?rloc) (at ?b ?bloc) (clear ?floc)
                (adjacent ?rloc ?bloc ?dir) (adjacent ?bloc ?floc ?dir))

:effect (and (at-robot ?bloc) (at ?b ?floc) (clear ?bloc)
        (not (at-robot ?rloc)) (not (at ?b ?bloc)) (not (clear ?floc)))
)
)

[QUERY PROBLEM]
(define(problem typed-sokoban-grid7-boxes1-walls2)
(:domain typed-sokoban)
(:objects
```

```
        up down left right - DIR
        box0 - BOX
        f0-0f f0-1f f0-2f f0-3f f0-4f f0-5f f0-6f
        f1-0f f1-1f f1-2f f1-3f f1-4f f1-5f f1-6f
        f2-0f f2-1f f2-2f f2-3f f2-4f f2-5f f2-6f
        f3-0f f3-1f f3-2f f3-3f f3-4f f3-5f f3-6f
        f4-0f f4-1f f4-2f f4-3f f4-4f f4-5f f4-6f
        f5-0f f5-1f f5-2f f5-3f f5-4f f5-5f f5-6f
        f6-0f f6-1f f6-2f f6-3f f6-4f f6-5f f6-6f - LOC
)
(:init
(adjacent f0-0f f0-1f right)
(adjacent f0-0f f1-0f down)
(adjacent f0-1f f0-0f left)
(adjacent f0-1f f0-2f right)
(adjacent f0-1f f1-1f down)
(adjacent f0-2f f0-1f left)
(adjacent f0-2f f0-3f right)
(adjacent f0-2f f1-2f down)
(adjacent f0-3f f0-2f left)
(adjacent f0-3f f0-4f right)
(adjacent f0-3f f1-3f down)
(adjacent f0-4f f0-3f left)
(adjacent f0-4f f0-5f right)
(adjacent f0-4f f1-4f down)
(adjacent f0-5f f0-4f left)
(adjacent f0-5f f0-6f right)
(adjacent f0-5f f1-5f down)
(adjacent f0-6f f0-5f left)
(adjacent f0-6f f1-6f down)
(adjacent f1-0f f1-1f right)
(adjacent f1-0f f0-0f up)
(adjacent f1-0f f2-0f down)
(adjacent f1-1f f1-0f left)
(adjacent f1-1f f1-2f right)
(adjacent f1-1f f0-1f up)
(adjacent f1-1f f2-1f down)
(adjacent f1-2f f1-1f left)
(adjacent f1-2f f1-3f right)
(adjacent f1-2f f0-2f up)
(adjacent f1-2f f2-2f down)
(adjacent f1-3f f1-2f left)
(adjacent f1-3f f1-4f right)
(adjacent f1-3f f0-3f up)
(adjacent f1-3f f2-3f down)
(adjacent f1-4f f1-3f left)
(adjacent f1-4f f1-5f right)
(adjacent f1-4f f0-4f up)
(adjacent f1-4f f2-4f down)
(adjacent f1-5f f1-4f left)
(adjacent f1-5f f1-6f right)
(adjacent f1-5f f0-5f up)
(adjacent f1-5f f2-5f down)
(adjacent f1-6f f1-5f left)
(adjacent f1-6f f0-6f up)
(adjacent f1-6f f2-6f down)
(adjacent f2-0f f2-1f right)
(adjacent f2-0f f1-0f up)
(adjacent f2-0f f3-0f down)
(adjacent f2-1f f2-0f left)
(adjacent f2-1f f2-2f right)
(adjacent f2-1f f1-1f up)
(adjacent f2-1f f3-1f down)
(adjacent f2-2f f2-1f left)
(adjacent f2-2f f2-3f right)
```

```
(adjacent f2-2f f1-2f up)
(adjacent f2-2f f3-2f down)
(adjacent f2-3f f2-2f left)
(adjacent f2-3f f2-4f right)
(adjacent f2-3f f1-3f up)
(adjacent f2-3f f3-3f down)
(adjacent f2-4f f2-3f left)
(adjacent f2-4f f2-5f right)
(adjacent f2-4f f1-4f up)
(adjacent f2-4f f3-4f down)
(adjacent f2-5f f2-4f left)
(adjacent f2-5f f2-6f right)
(adjacent f2-5f f1-5f up)
(adjacent f2-5f f3-5f down)
(adjacent f2-6f f2-5f left)
(adjacent f2-6f f1-6f up)
(adjacent f2-6f f3-6f down)
(adjacent f3-0f f3-1f right)
(adjacent f3-0f f2-0f up)
(adjacent f3-0f f4-0f down)
(adjacent f3-1f f3-0f left)
(adjacent f3-1f f3-2f right)
(adjacent f3-1f f2-1f up)
(adjacent f3-1f f4-1f down)
(adjacent f3-2f f3-1f left)
(adjacent f3-2f f3-3f right)
(adjacent f3-2f f2-2f up)
(adjacent f3-2f f4-2f down)
(adjacent f3-3f f3-2f left)
(adjacent f3-3f f3-4f right)
(adjacent f3-3f f2-3f up)
(adjacent f3-3f f4-3f down)
(adjacent f3-4f f3-3f left)
(adjacent f3-4f f3-5f right)
(adjacent f3-4f f2-4f up)
(adjacent f3-4f f4-4f down)
(adjacent f3-5f f3-4f left)
(adjacent f3-5f f3-6f right)
(adjacent f3-5f f2-5f up)
(adjacent f3-5f f4-5f down)
(adjacent f3-6f f3-5f left)
(adjacent f3-6f f2-6f up)
(adjacent f3-6f f4-6f down)
(adjacent f4-0f f4-1f right)
(adjacent f4-0f f3-0f up)
(adjacent f4-0f f5-0f down)
(adjacent f4-1f f4-0f left)
(adjacent f4-1f f4-2f right)
(adjacent f4-1f f3-1f up)
(adjacent f4-1f f5-1f down)
(adjacent f4-2f f4-1f left)
(adjacent f4-2f f4-3f right)
(adjacent f4-2f f3-2f up)
(adjacent f4-2f f5-2f down)
(adjacent f4-3f f4-2f left)
(adjacent f4-3f f4-4f right)
(adjacent f4-3f f3-3f up)
(adjacent f4-3f f5-3f down)
(adjacent f4-4f f4-3f left)
(adjacent f4-4f f4-5f right)
(adjacent f4-4f f3-4f up)
(adjacent f4-4f f5-4f down)
(adjacent f4-5f f4-4f left)
(adjacent f4-5f f4-6f right)
(adjacent f4-5f f3-5f up)
```

```
1674    (adjacent f4-5f f5-5f down)
1675    (adjacent f4-6f f4-5f left)
1676    (adjacent f4-6f f3-6f up)
1677    (adjacent f4-6f f5-6f down)
1678    (adjacent f5-0f f5-1f right)
1679    (adjacent f5-0f f4-0f up)
        (adjacent f5-0f f6-0f down)
1680    (adjacent f5-1f f5-0f left)
1681    (adjacent f5-1f f5-2f right)
1682    (adjacent f5-1f f4-1f up)
1683    (adjacent f5-1f f6-1f down)
        (adjacent f5-2f f5-1f left)
1684    (adjacent f5-2f f5-3f right)
1685    (adjacent f5-2f f4-2f up)
1686    (adjacent f5-2f f6-2f down)
1687    (adjacent f5-3f f5-2f left)
1688    (adjacent f5-3f f5-4f right)
        (adjacent f5-3f f4-3f up)
1689    (adjacent f5-3f f6-3f down)
1690    (adjacent f5-4f f5-3f left)
1691    (adjacent f5-4f f5-5f right)
1692    (adjacent f5-4f f4-4f up)
1693    (adjacent f5-4f f6-4f down)
        (adjacent f5-5f f5-4f left)
1694    (adjacent f5-5f f5-6f right)
1695    (adjacent f5-5f f4-5f up)
1696    (adjacent f5-5f f6-5f down)
1697    (adjacent f5-6f f5-5f left)
1698    (adjacent f5-6f f4-6f up)
        (adjacent f5-6f f6-6f down)
1699    (adjacent f6-0f f6-1f right)
1700    (adjacent f6-0f f5-0f up)
1701    (adjacent f6-1f f6-0f left)
1702    (adjacent f6-1f f6-2f right)
1703    (adjacent f6-1f f5-1f up)
        (adjacent f6-2f f6-1f left)
1704    (adjacent f6-2f f6-3f right)
1705    (adjacent f6-2f f5-2f up)
1706    (adjacent f6-3f f6-2f left)
1707    (adjacent f6-3f f6-4f right)
1708    (adjacent f6-3f f5-3f up)
        (adjacent f6-4f f6-3f left)
1709    (adjacent f6-4f f6-5f right)
1710    (adjacent f6-4f f5-4f up)
1711    (adjacent f6-5f f6-4f left)
1712    (adjacent f6-5f f6-6f right)
1713    (adjacent f6-5f f5-5f up)
        (adjacent f6-6f f6-5f left)
1714    (adjacent f6-6f f5-6f up)
1715    (at box0 f1-2f)
1716    (clear f0-0f)
1717    (clear f0-2f)
1718    (clear f0-3f)
1719    (clear f0-4f)
        (clear f0-5f)
1720    (clear f0-6f)
1721    (clear f1-0f)
1722    (clear f1-1f)
1723    (clear f1-3f)
1724    (clear f1-4f)
1725    (clear f1-5f)
        (clear f1-6f)
1726    (clear f2-0f)
1727    (clear f2-1f)
        (clear f2-2f)
```

```
1728   (clear f2-3f)
1729   (clear f2-4f)
1730   (clear f2-5f)
1731   (clear f2-6f)
1732   (clear f3-0f)
1733   (clear f3-3f)
1734   (clear f3-4f)
1735   (clear f3-5f)
1736   (clear f3-6f)
1737   (clear f4-0f)
1738   (clear f4-1f)
1739   (clear f4-2f)
1740   (clear f4-3f)
1741   (clear f4-4f)
1742   (clear f4-5f)
1743   (clear f4-6f)
1744   (clear f5-0f)
1745   (clear f5-1f)
1746   (clear f5-2f)
1747   (clear f5-3f)
1748   (clear f5-4f)
1749   (clear f5-5f)
1750   (clear f5-6f)
1751   (at-robot f6-0f)
1752   (clear f6-0f)
1753   (clear f6-1f)
1754   (clear f6-2f)
1755   (clear f6-3f)
1756   (clear f6-4f)
1757   (clear f6-6f)
1758   )
1759   (:goal
1760   (and
1761   (at box0 f6-6f)
       )
       )
       )

       [PLAN]
       ==============o1-preview RESPONSE================
       (move f6-0f f5-0f up)
       (move f5-0f f4-0f up)
       (move f4-0f f3-0f up)
       (move f3-0f f2-0f up)
       (move f2-0f f1-0f up)
       (move f1-0f f1-1f right)
       (push f1-1f f1-2f f1-3f right box0)
       (push f1-2f f1-3f f1-4f right box0)
       (push f1-3f f1-4f f1-5f right box0)
       (push f1-4f f1-5f f1-6f right box0)
       (move f1-5f f0-5f up)
       (move f0-5f f0-6f right)
       (push f0-6f f1-6f f2-6f down box0)
       (push f1-6f f2-6f f3-6f down box0)
       (push f2-6f f3-6f f4-6f down box0)
       (push f3-6f f4-6f f5-6f down box0)
       (push f4-6f f5-6f f6-6f down box0)
```

## L.2   SOKOBAN - BACKPROMPT

The above plan is invalid. The following action at step 20 has an unmet precondition
:
push f8-5f f8-6f f8-7f right box1
The unmet precondition is:

```
(clear f8-7f)
```
Provide your revised plan as a sequence of actions in PDDL format. An action ACTION
    which acts on two objects OBJ1 and OBJ2 would be written (ACTION OBJ1 OBJ2). Do
      not provide anything else in your answer.

[PLAN]

## M   Prompts for Graph Coloring

### M.1   Graph Coloring - Hard

    Color the following graph, described as a set of edges, such that no two
        vertices on the same edge share a color.
You may use at most 5 colors.
Vertex 0 is connected to vertex 3.
Vertex 0 is connected to vertex 6.
Vertex 0 is connected to vertex 8.
Vertex 0 is connected to vertex 13.
Vertex 0 is connected to vertex 15.
Vertex 0 is connected to vertex 17.
Vertex 0 is connected to vertex 19.
Vertex 1 is connected to vertex 4.
Vertex 1 is connected to vertex 5.
Vertex 1 is connected to vertex 8.
Vertex 1 is connected to vertex 11.
Vertex 1 is connected to vertex 12.
Vertex 1 is connected to vertex 13.
Vertex 1 is connected to vertex 14.
Vertex 1 is connected to vertex 17.
Vertex 1 is connected to vertex 18.
Vertex 1 is connected to vertex 19.
Vertex 2 is connected to vertex 3.
Vertex 2 is connected to vertex 4.
Vertex 2 is connected to vertex 6.
Vertex 2 is connected to vertex 8.
Vertex 2 is connected to vertex 13.
Vertex 2 is connected to vertex 14.
Vertex 2 is connected to vertex 15.
Vertex 2 is connected to vertex 16.
Vertex 2 is connected to vertex 17.
Vertex 3 is connected to vertex 5.
Vertex 3 is connected to vertex 8.
Vertex 3 is connected to vertex 11.
Vertex 3 is connected to vertex 12.
Vertex 3 is connected to vertex 14.
Vertex 3 is connected to vertex 16.
Vertex 4 is connected to vertex 5.
Vertex 4 is connected to vertex 7.
Vertex 4 is connected to vertex 9.
Vertex 4 is connected to vertex 11.
Vertex 4 is connected to vertex 17.
Vertex 4 is connected to vertex 18.
Vertex 5 is connected to vertex 11.
Vertex 5 is connected to vertex 14.
Vertex 5 is connected to vertex 15.
Vertex 6 is connected to vertex 11.
Vertex 6 is connected to vertex 16.
Vertex 6 is connected to vertex 17.
Vertex 7 is connected to vertex 9.
Vertex 7 is connected to vertex 10.
Vertex 7 is connected to vertex 13.
Vertex 7 is connected to vertex 14.
Vertex 7 is connected to vertex 16.
Vertex 7 is connected to vertex 17.

```
Vertex 8 is connected to vertex 10.
Vertex 8 is connected to vertex 12.
Vertex 8 is connected to vertex 13.
Vertex 8 is connected to vertex 16.
Vertex 8 is connected to vertex 19.
Vertex 9 is connected to vertex 11.
Vertex 9 is connected to vertex 13.
Vertex 9 is connected to vertex 17.
Vertex 10 is connected to vertex 11.
Vertex 10 is connected to vertex 13.
Vertex 10 is connected to vertex 16.
Vertex 10 is connected to vertex 17.
Vertex 10 is connected to vertex 18.
Vertex 10 is connected to vertex 19.
Vertex 11 is connected to vertex 13.
Vertex 11 is connected to vertex 17.
Vertex 11 is connected to vertex 18.
Vertex 11 is connected to vertex 19.
Vertex 12 is connected to vertex 13.
Vertex 12 is connected to vertex 15.
Vertex 12 is connected to vertex 16.
Vertex 12 is connected to vertex 18.
Vertex 12 is connected to vertex 19.
Vertex 13 is connected to vertex 14.
Vertex 13 is connected to vertex 18.
Vertex 14 is connected to vertex 15.
Vertex 14 is connected to vertex 18.
Vertex 14 is connected to vertex 19.
There are a total of 20 vertices. Please label every vertex, even if it is
    disconnected from the rest of the graph.Please provide each vertex's color. Do
    not skip any vertices. Each color must be provided on a new line in the
    response and should be formatted as "{VERTEX NUMBER}: {VERTEX COLOR ASSIGNMENT
    }". Please do not provide anything else in your response, and end your response
     with '[ANSWER END]'
=============o1-preview RESPONSE===============
0: 1
1: 3
2: 1
3: 2
4: 4
5: 5
6: 3
7: 1
8: 4
9: 3
10: 3
11: 1
12: 1
13: 2
14: 4
15: 2
16: 5
17: 2
18: 5
19: 2
[ANSWER END]
```

## N   Graph Coloring Backprompt - Hard

This is incorrect. Feedback:
```
Vertex 0 was not given a value in the coloring.
Vertex 1 was not given a value in the coloring.
Vertex 0 was not given a value in the coloring.
Vertex 2 was not given a value in the coloring.
```

```
Vertex 0 was not given a value in the coloring.
Vertex 3 was not given a value in the coloring.
Vertex 0 was not given a value in the coloring.
Vertex 4 was not given a value in the coloring.
Vertex 0 was not given a value in the coloring.
Vertex 10 was not given a value in the coloring.
Vertex 0 was not given a value in the coloring.
Vertex 12 was not given a value in the coloring.
Vertex 0 was not given a value in the coloring.
Vertex 13 was not given a value in the coloring.
Vertex 0 was not given a value in the coloring.
Vertex 15 was not given a value in the coloring.
Vertex 0 was not given a value in the coloring.
Vertex 16 was not given a value in the coloring.
Vertex 0 was not given a value in the coloring.
Vertex 17 was not given a value in the coloring.
Vertex 0 was not given a value in the coloring.
Vertex 18 was not given a value in the coloring.
Vertex 1 was not given a value in the coloring.
Vertex 2 was not given a value in the coloring.
Vertex 1 was not given a value in the coloring.
Vertex 3 was not given a value in the coloring.
Vertex 1 was not given a value in the coloring.
Vertex 8 was not given a value in the coloring.
Vertex 1 was not given a value in the coloring.
Vertex 9 was not given a value in the coloring.
Vertex 1 was not given a value in the coloring.
Vertex 10 was not given a value in the coloring.
Vertex 1 was not given a value in the coloring.
Vertex 15 was not given a value in the coloring.
Vertex 1 was not given a value in the coloring.
Vertex 18 was not given a value in the coloring.
Vertex 2 was not given a value in the coloring.
Vertex 3 was not given a value in the coloring.
Vertex 2 was not given a value in the coloring.
Vertex 5 was not given a value in the coloring.
Vertex 2 was not given a value in the coloring.
Vertex 6 was not given a value in the coloring.
Vertex 2 was not given a value in the coloring.
Vertex 8 was not given a value in the coloring.
Vertex 2 was not given a value in the coloring.
Vertex 11 was not given a value in the coloring.
Vertex 2 was not given a value in the coloring.
Vertex 12 was not given a value in the coloring.
Vertex 2 was not given a value in the coloring.
Vertex 14 was not given a value in the coloring.
Vertex 2 was not given a value in the coloring.
Vertex 15 was not given a value in the coloring.
Vertex 2 was not given a value in the coloring.
Vertex 19 was not given a value in the coloring.
Vertex 3 was not given a value in the coloring.
Vertex 18 was not given a value in the coloring.
Vertex 3 was not given a value in the coloring.
Vertex 19 was not given a value in the coloring.
Vertex 4 was not given a value in the coloring.
Vertex 6 was not given a value in the coloring.
Vertex 4 was not given a value in the coloring.
Vertex 7 was not given a value in the coloring.
Vertex 4 was not given a value in the coloring.
Vertex 9 was not given a value in the coloring.
Vertex 4 was not given a value in the coloring.
Vertex 11 was not given a value in the coloring.
Vertex 4 was not given a value in the coloring.
Vertex 13 was not given a value in the coloring.
Vertex 4 was not given a value in the coloring.
```

```
Vertex 18 was not given a value in the coloring.
Vertex 4 was not given a value in the coloring.
Vertex 19 was not given a value in the coloring.
Vertex 5 was not given a value in the coloring.
Vertex 6 was not given a value in the coloring.
Vertex 5 was not given a value in the coloring.
Vertex 7 was not given a value in the coloring.
Vertex 5 was not given a value in the coloring.
Vertex 8 was not given a value in the coloring.
Vertex 5 was not given a value in the coloring.
Vertex 10 was not given a value in the coloring.
Vertex 5 was not given a value in the coloring.
Vertex 14 was not given a value in the coloring.
Vertex 6 was not given a value in the coloring.
Vertex 7 was not given a value in the coloring.
Vertex 6 was not given a value in the coloring.
Vertex 9 was not given a value in the coloring.
Vertex 6 was not given a value in the coloring.
Vertex 10 was not given a value in the coloring.
Vertex 6 was not given a value in the coloring.
Vertex 12 was not given a value in the coloring.
Vertex 6 was not given a value in the coloring.
Vertex 13 was not given a value in the coloring.
Vertex 6 was not given a value in the coloring.
Vertex 14 was not given a value in the coloring.
Vertex 6 was not given a value in the coloring.
Vertex 15 was not given a value in the coloring.
Vertex 6 was not given a value in the coloring.
Vertex 16 was not given a value in the coloring.
Vertex 6 was not given a value in the coloring.
Vertex 17 was not given a value in the coloring.
Vertex 6 was not given a value in the coloring.
Vertex 19 was not given a value in the coloring.
Vertex 7 was not given a value in the coloring.
Vertex 8 was not given a value in the coloring.
Vertex 7 was not given a value in the coloring.
Vertex 9 was not given a value in the coloring.
Vertex 7 was not given a value in the coloring.
Vertex 11 was not given a value in the coloring.
Vertex 7 was not given a value in the coloring.
Vertex 12 was not given a value in the coloring.
Vertex 7 was not given a value in the coloring.
Vertex 13 was not given a value in the coloring.
Vertex 7 was not given a value in the coloring.
Vertex 14 was not given a value in the coloring.
Vertex 7 was not given a value in the coloring.
Vertex 15 was not given a value in the coloring.
Vertex 7 was not given a value in the coloring.
Vertex 18 was not given a value in the coloring.
Vertex 8 was not given a value in the coloring.
Vertex 11 was not given a value in the coloring.
Vertex 8 was not given a value in the coloring.
Vertex 13 was not given a value in the coloring.
Vertex 8 was not given a value in the coloring.
Vertex 14 was not given a value in the coloring.
Vertex 8 was not given a value in the coloring.
Vertex 19 was not given a value in the coloring.
Vertex 9 was not given a value in the coloring.
Vertex 10 was not given a value in the coloring.
Vertex 9 was not given a value in the coloring.
Vertex 11 was not given a value in the coloring.
Vertex 9 was not given a value in the coloring.
Vertex 13 was not given a value in the coloring.
Vertex 9 was not given a value in the coloring.
Vertex 14 was not given a value in the coloring.
```

```
Vertex 9 was not given a value in the coloring.
Vertex 15 was not given a value in the coloring.
Vertex 10 was not given a value in the coloring.
Vertex 12 was not given a value in the coloring.
Vertex 10 was not given a value in the coloring.
Vertex 16 was not given a value in the coloring.
Vertex 10 was not given a value in the coloring.
Vertex 17 was not given a value in the coloring.
Vertex 11 was not given a value in the coloring.
Vertex 12 was not given a value in the coloring.
Vertex 11 was not given a value in the coloring.
Vertex 13 was not given a value in the coloring.
Vertex 11 was not given a value in the coloring.
Vertex 14 was not given a value in the coloring.
Vertex 11 was not given a value in the coloring.
Vertex 15 was not given a value in the coloring.
Vertex 11 was not given a value in the coloring.
Vertex 17 was not given a value in the coloring.
Vertex 11 was not given a value in the coloring.
Vertex 19 was not given a value in the coloring.
Vertex 12 was not given a value in the coloring.
Vertex 13 was not given a value in the coloring.
Vertex 12 was not given a value in the coloring.
Vertex 18 was not given a value in the coloring.
Vertex 13 was not given a value in the coloring.
Vertex 17 was not given a value in the coloring.
Vertex 14 was not given a value in the coloring.
Vertex 19 was not given a value in the coloring.
Vertex 15 was not given a value in the coloring.
Vertex 19 was not given a value in the coloring.
Vertex 16 was not given a value in the coloring.
Vertex 17 was not given a value in the coloring.
Vertex 17 was not given a value in the coloring.
Vertex 18 was not given a value in the coloring.
```

Using this feedback, please try again.
Please provide each vertex's color. Do not skip any vertices. Each color must be
    provided on a new line in the response and should be formatted as "{VERTEX
    NUMBER}: {VERTEX COLOR ASSIGNMENT}". Please do not provide anything else in
    your response, and end your response with '[ANSWER END]'

## O   PROMPTS FOR OSU TRAVEL PLANNING

### O.1   FIRST ITERATION

You are a proficient planner. Based on the provided information and query, please
    give me a detailed plan, including specifics such as flight numbers (e.g.,
    F0123456),
restaurant names, and accommodation names. Note that all the information in your
    plan should be derived from the provided data. You should give a travel plan in
     JSON format as shown in the
example below. Additionally, all details should align with commonsense. The symbol
    '-' indicates that information is unnecessary. For example, in the provided
    sample, you do not need to
plan after returning to the departure city. When you travel to two cities in one day
    , you should note it in the 'current_city' section as in the example (i.e.,
    from A to B).

***** Example *****
Query: Could you create a travel plan for 7 people from Ithaca to Charlotte spanning
     3 days, from March 8th to March 14th, 2022, with a budget of $30,200?
Travel Plan:
[
    {

```
        "day": 1,
        "people_number": 7,
        "current_city": "from Ithaca to Charlotte",
        "transportation": "Flight Number: F3633413, from Ithaca to Charlotte,
            Departure Time: 05:38, Arrival Time: 07:46",
        "breakfast": "Nagaland's Kitchen, Charlotte",
        "attraction": "The Charlotte Museum of History, Charlotte",
        "lunch": "Cafe Maple Street, Charlotte",
        "dinner": "Bombay Vada Pav, Charlotte",
        "accommodation": "Affordable Spacious Refurbished Room in Bushwick!,
            Charlotte"
    },
    {
        "day": 2,
        "people_number": 7,
        "current_city": "Charlotte",
        "transportation": "-",
        "breakfast": "Olive Tree Cafe, Charlotte",
        "attraction": "The Mint Museum, Charlotte;Romare Bearden Park, Charlotte",
        "lunch": "Birbal Ji Dhaba, Charlotte",
        "dinner": "Pind Balluchi, Charlotte",
        "accommodation": "Affordable Spacious Refurbished Room in Bushwick!,
            Charlotte"
    },
    {
        "day": 3,
        "people_number": 7,
        "current_city": "from Charlotte to Ithaca",
        "transportation": "Flight Number: F3786167, from Charlotte to Ithaca,
            Departure Time: 21:42, Arrival Time: 23:26",
        "breakfast": "Books Monument, Charlotte",
        "attraction": "Books Monument, Charlotte",
        "lunch": "Olive Tree Cafe, Charlotte",
        "dinner": "Kylin Skybar, Charlotte",
        "accommodation": "-"
    }
]
***** Example Ends *****

Given information: [{'Description': 'Attractions in Nashville', 'Content': '
                        Name Latitude Longitude
                                    Address
Phone
Website    City\nCountry Music Hall of Fame and Museum 36.158263 -86.776126 222 Rep.
    John Lewis Way S, Nashville, TN 37203, USA (615) 416-2001
https://countrymusichalloffame.org/ Nashville\n    Nashville Zoo at Grassmere
    36.089705 -86.742096 3777 Nolensville Pk, Nashville, TN 37211, USA (615)
    833-1534
http://www.nashvillezoo.org/ Nashville\n Belle Meade Historic Site & Winery
    36.104916 -86.864695 5025 Harding Pike, Nashville, TN 37205, USA (615) 356-0501
https://visitbellemeade.com/ Nashville\n          Johnny Cash Museum 36.160939
    -86.775757      119 3rd Ave S, Nashville, TN 37201, USA (615) 256-1777
http://www.johnnycashmuseum.com/ Nashville\n          Centennial Park
    36.148946 -86.812750    2500 West End Ave, Nashville, TN 37203, USA (615)
    862-8400
https://www.nashville.gov/Parks-and-Recreation/Parks/Centennial-Park.aspx Nashville\
    n          Grand Ole Opry 36.206857 -86.692108    600 Opry Mills Dr,
    Nashville, TN
37214, USA (615) 871-6779
https://www.opry.com/?utm_campaign=opry&utm_medium=organicsearch&utm_source=
    googlemybusiness&utm_audience=tofu_googlemybusiness&utm_content=
    brandstory_google_my_business_website_link
Nashville\n              Frist Art Museum 36.157897 -86.783853        919
    Broadway, Nashville, TN 37203, USA (615) 244-3340
```

https://fristartmuseum.org/ Nashville\n                        The Parthenon 36.149674
     -86.813347     2500 West End Ave, Nashville, TN 37203, USA (615) 862-8431
https://www.nashvilleparthenon.com/ Nashville\n                 Nashville Shores
     36.158209 -86.604915          4001 Bell Rd, Hermitage, TN 37076, USA (615)
     889-7050
http://www.nashvilleshores.com/lodging Nashville\n Musicians Hall of Fame and Museum
     36.167668 -86.782399     401 Gay St, Nashville, TN 37219, USA (615) 244-3263
https://www.musicianshalloffame.com/ Nashville\n Bicentennial Capitol Mall State
     Park 36.170887 -86.787589 600 James Robertson Pkwy, Nashville, TN 37243, USA
     (888) 867-2757
https://tnstateparks.com/parks/bicentennial-mall Nashville\n              Lane Motor
     Museum 36.140197 -86.734580 702 Murfreesboro Pike, Nashville, TN 37210, USA
     (615) 742-7445
http://www.lanemotormuseum.org/ Nashville\n               Honky Tonk Highway 36.160393
     -86.778432       501 Broadway, Nashville, TN 37203, USA (800) 657-6910
https://www.visitmusiccity.com/visitors/honkytonkhighway Nashville\n     Adventure
     Science Center 36.146614 -86.775481 800 Fort Negley Blvd, Nashville, TN 37203,
     USA (615)
862-5160
http://www.adventuresci.org/ Nashville\n               Patsy Cline Museum 36.160904
     -86.776030      119 3rd Ave S, Nashville, TN 37201, USA (615) 454-4722
https://www.patsymuseum.com/ Nashville\n     Andrew  Jacksons  Hermitage 36.213757
     -86.615310     4580 Rachels Ln, Hermitage, TN 37076, USA (615) 889-2941
http://www.thehermitage.com/ Nashville\n               NashTrash Tours 36.169474
     -86.788254 900 Rosa L Parks Blvd, Nashville, TN 37208, USA (615) 226-7300
http://www.nashtrash.com/ Nashville\n               Fort Nashborough 36.164268
     -86.775430     170 1st Ave N, Nashville, TN 37201, USA (615) 862-8400
https://www.nashville.gov/Parks-and-Recreation/Historic-Sites/Fort-Nashborough.aspx
     Nashville\n        Ryman Auditorium 36.161248 -86.778471           116 5th Ave N,
Nashville, TN 37219, USA (615) 889-3060
https://www.ryman.com/?utm_campaign=ryman&utm_medium=organicsearch&utm_source=
     googlemybusiness&utm_audience=tofu_googlemybusiness&utm_content=
     brandstory_google_my_business_website_link
Nashville\n      Nashville Public Square Park 36.166611 -86.778126 Union St & 3rd
     Ave N, Nashville, TN 37201, USA (615) 743-3090
https://www.nashvilledowntown.com/go/public-square-park Nashville'}, {'Description':
     'Restaurants in Nashville', 'Content': "            Name Average Cost
Cuisines Aggregate Rating    City\n                     Bangkok 1        44
                         Cafe, Seafood       3.3 Nashville\n
                 Bablu Fast
Food       13   Desserts, Pizza, Mexican, BBQ, Fast Food        0.0 Nashville\
     n                 Full Dabba       77  Pizza, Mexican, Fast Food, Cafe,
     American
0.0 Nashville\n                  Twigly       14 Pizza, French, Fast Food,
     Chinese, Seafood       4.5 Nashville\n                  Veg Hut
             35
Tea, Mexican, Seafood          0.0 Nashville\n            The Toddy Shop
          92               Desserts, Fast Food       3.4 Nashville\n
GoGourmet        34       Tea, Cafe, Indian, Mediterranean        3.7
     Nashville\n      Govinda's Confectionery     15     Tea, Bakery,
     Mediterranean, Fast Food
3.1 Nashville\n            Kitchen King     13    Pizza, Italian, BBQ,
     Cafe, Mediterranean       0.0 Nashville\n             Town Hall
           55
Tea, Seafood          3.8 Nashville\n          Smoke House Deli        59
             Cafe, Mediterranean, Seafood      4.0 Nashville\n
     Dialogue
Lounge & Caf        83   Tea, Indian, BBQ, Cafe, American, Seafood        3.0
     Nashville\n             Meenakshi Bhawan     24
                 Tea,
Desserts        3.1 Nashville\n          Oh! Calcutta       83
             Tea, Mexican, BBQ, Cafe, Indian      4.4 Nashville\n
                 Chicago

```
Pizza              20                    Tea, French, Pizza, BBQ            3.2 Nashville\
    n          Malhotra Restaurant          76                    Tea, Desserts,
    Seafood
2.9 Nashville\n                      Madras Cafe          88                    Pizza,
    Desserts, Seafood          0.0 Nashville\nDakshin - Sheraton New Delhi Hotel
           63
Cafe, Pizza, Indian, Bakery          4.0 Nashville\n          1911 Bar - The
    Imperial        76 Desserts, Pizza, Italian, American, Seafood          3.2
    Nashville\n
Punjabi Virsa          35          Desserts, Italian, BBQ, Cafe, Seafood          0.0
    Nashville\n                      Sagar Dhaba          55          Tea, Bakery, BBQ,
    Cafe,
Mediterranean          0.0 Nashville\n                      Kettle & Kegs          60
                            Tea, Seafood          0.0 Nashville\n
Kargo          84                    Tea, Pizza, Desserts          4.4 Nashville
    "}, {'Description': 'Accommodations in Nashville', 'Content': '
NAME price      room type                              house_rules minimum nights
    maximum occupancy review rate number  city\nHuge 2 Bedroom, Great Location,
    Express Metro
745.0 Entire home/apt No smoking & No children under 10 & No pets      4.0
                 4                5.0 Nashville\n Clean and large bedroom in a
    private house 474.0
Private room                              No smoking          1.0                1
                4.0 Nashville\n                      Brooklyn Heights gem 993.0
    Entire
home/apt                              No pets          2.0                6
                5.0 Nashville\n     Lovely room in heart of Williamsburg 61.0
    Private room
No pets          2.0                1          4.0 Nashville\n *Light & Love*
    vibrant, historic, sleeps 4 679.0 Entire home/apt                      No
    pets
5.0                2                2.0 Nashville\n     FiDi Cozy room overlooking
    East River 870.0 Private room                      No parties
    1.0
2                5.0 Nashville\n     Charming 1BR with sun-nook in Brooklyn 256.0
    Entire home/apt No children under 10 & No pets & No smoking  3.0
                 2
4.0 Nashville\n          Cozy bedroom close to Manhattan 576.0 Private room
                No children under 10          1.0                1
                3.0
Nashville'}, {'Description': 'Attractions in Knoxville', 'Content': "
                            Name Latitude Longitude
                            Address
Phone

    Website    City\n                      World's Fair Park 35.962577 -83.924192
525 Henley St, Knoxville, TN 37902, USA (865) 215-1158
                                            http://
    worldsfairpark.org/ Knoxville\n
Knoxville Museum of Art 35.962426 -83.925229 1050 Worlds Fair Park Dr, Knoxville, TN
    37916, USA (865) 525-6101
http://www.knoxart.org/ Knoxville\n                      Sunsphere 35.961707
    -83.923353        810 Clinch Ave, Knoxville, TN 37902, USA (865) 314-0660
http://www.sunspheretickets.com/ Knoxville\n              Ijams Nature Center
    35.956454 -83.866775    2915 Island Home Ave, Knoxville, TN 37920, USA (865)
    577-4717
http://www.ijams.org/ Knoxville\n          Knoxville Walking Tours 35.966448
    -83.919167        301 S Gay St, Knoxville, TN 37902, USA (865) 309-4522
http://knoxvillewalkingtours.com/ Knoxville\n                      Muse Knoxville
    35.997617 -83.885467        516 N Beaman St, Knoxville, TN 37914, USA (865)
    594-1494
http://www.themuseknoxville.org/ Knoxville\n Knoxville Botanical Garden and
    Arboretum 35.982160 -83.881077    2743 Wimpole Ave, Knoxville, TN 37914, USA
    (865) 862-8717
```

```
http://www.knoxgarden.org/ Knoxville\n          Haunted Knoxville Ghost Tours
    35.965963 -83.919553 36 Market Square #1404, Knoxville, TN 37902, USA (865)
    377-9677
http://www.hauntedknoxville.net/ Knoxville\n               Three Rivers Rambler
    35.952810 -83.940376 2560 University Commons Way, Knoxville, TN 37919, USA (865)
    524-9411
http://www.threeriversrambler.com/ Knoxville\n                 Charles Krutch Park
    35.964414 -83.918695      504 Market St, Knoxville, TN 37902, USA (865)
    215-4248
https://www.knoxvilletn.gov/government/city_departments_offices/parks_and_recreation
    /parks/krutch_park Knoxville\n McClung Museum of Natural History & Culture
    35.952005 -83.927209
1327 Cir Park Dr, Knoxville, TN 37996, USA (865) 974-2144
                                                       http://mcclungmuseum
    .utk.edu/ Knoxville\n
Knoxville Sightseeing 35.992790 -83.904128       2519 Mitchell St, Knoxville, TN
    37917, USA (865) 566-0634
Unknown Knoxville\n      Chilhowee Park & Exposition Center 35.996296 -83.884032
        3301 E Magnolia Ave, Knoxville, TN 37914, USA (865) 215-1450
https://chilhoweepark.com/ Knoxville\n                    Augusta Quarry
    35.944772 -83.911886 3000 Fort Dickerson Rd SW, Knoxville, TN 37920, USA
    Unknown
Unknown Knoxville\n                   Zoo Knoxville 35.999812 -83.888250
    3500 Knoxville Zoo Dr, Knoxville, TN 37914, USA (865) 637-5331
https://www.zooknoxville.org/ Knoxville\n   Outdoor Knoxville Adventure Center
    35.961902 -83.912663 900 Volunteer Landing Ln, Knoxville, TN 37915, USA (865)
    228-8424
http://www.outdoorknoxville.com Knoxville\n      James White Fort Association
    35.962962 -83.912169      205 E Hill Ave, Knoxville, TN 37915, USA (865)
    525-6514
http://jameswhitesfort.org/ Knoxville\n   Rowing Man Statue in Knoxville, TN
    35.963611 -83.917519 W. Church Avenue &, S Gay St, Knoxville, TN 37902, USA
    Unknown
Unknown Knoxville\n                 UT Gardens Knoxville 35.944000 -83.938260
            2518 Jacob Dr, Knoxville, TN 37996, USA (865) 974-7324
http://utgardens.tennessee.edu/locations/knoxville Knoxville\nEast Tennessee
    Historical Society and Museum 35.964117 -83.917794   601 S Gay St, Knoxville, TN
    37902, USA (865)
215-8830                                                  http://www.
    easttnhistory.org/ Knoxville"}, {'Description': 'Restaurants in Knoxville', '
    Content': '
Name  Average Cost                           Cuisines Aggregate Rating
    City\n        Cafe Arabelle        29               French, BBQ, Desserts,
    Seafood
3.6 Knoxville\n     Les 3 Brasseurs       24        Desserts, Pizza, Italian,
    Cafe, Indian        4.6 Knoxville\n        Sky On 57        150
            Tea,
Cafe, Pizza, Chinese, Seafood       3.4 Knoxville\n     Cafe El Chico
        67             Tea, French, Indian, Seafood        3.6
    Knoxville\n
TcozY        85        Tea, Pizza, Mexican, Fast Food, Cafe        0.0
    Knoxville\nRama Vaishnav Bhojnalaya       46              Tea, Cafe, Pizza
    , Desserts
3.0 Knoxville\n      Burger Planet       45              Tea, Bakery,
    American, Cafe       3.2 Knoxville\n    Cafe Coffee Day       82
            Fast
Food, American, BBQ, Italian       2.8 Knoxville\n        Mamagoto
        14       Indian, Mediterranean, Desserts, Seafood       4.1
    Knoxville\n   Ali Baba &
41 Dishes        70              Pizza, Desserts, Fast Food        3.5
    Knoxville\n     Punjabi Flavour       41        Tea, Bakery, Cafe,
    Indian, Seafood
0.0 Knoxville\n      Tandoori Tadka       23        Chinese, Pizza, Desserts,
    Fast Food       0.0 Knoxville\n     Coalition Cafe       35
    Desserts,
```

Tea, BBQ, Cafe, Mediterranean              3.4 Knoxville\n          Chaat Corner
            67                              Cafe, Pizza              3.2
    Knoxville\n
El Posto           63                         Tea, Cafe, Pizza, BBQ            3.5
    Knoxville\n           La-Nawaab           17                 French, Bakery,
    Cafe, Seafood
3.0 Knoxville\n       Peppers & Pipes           86                 Chinese, Pizza,
    Cafe, Italian          3.1 Knoxville\n         Chit Chat           15
Pizza, Fast Food           3.1 Knoxville\n       Anand Sweets         45
                Fast Food, Pizza, Indian, Seafood         0.0 Knoxville\n
    Open Kitchen
13 Pizza, Bakery, BBQ, Cafe, Indian, Mediterranean         3.0 Knoxville\n
    Biryani By Kilo         17                 Chinese, Pizza, Fast Food
            4.1
Knoxville\n    Chinese Food Corner         30                 Indian, Desserts,
    Fast Food         0.0 Knoxville\nThe Indian Kaffe Express       71
Tea, Desserts           3.8 Knoxville'}, {'Description': 'Accommodations in
    Knoxville', 'Content': '                              NAME price      room
    type
house_rules minimum nights maximum occupancy review rate number  city\n  Cozy
    Private Room in Chinatown/ Lower East Side 132.0 Private room
                        No pets &
No visitors           7.0               2           4.0 Knoxville\n
                        The Diamond Room 1008.0 Private room
                        No parties &
No visitors           1.0               1           5.0 Knoxville\n
                Light-filled Room in Renovated Apt 310.0 Private room
No pets           2.0               2           2.0 Knoxville\n
                        Private Room 922.0   Private room
                        No
visitors           1.0               1           4.0 Knoxville\n
    Beautiful & Private Manhattan Room 721.0 Private room               No
    parties & No
smoking           1.0               1           2.0 Knoxville\n         Lg
    Quiet Artist Home -Ditmas Park - 561.0 Entire home/apt     No visitors & No
    pets & No
smoking           24.0               4           2.0 Knoxville\n Beautiful 1-
    bdrm apt in tranquil Inwood building 1091.0 Entire home/apt           No
    smoking &
No pets           2.0               4           4.0 Knoxville\n 1,600sq ft
    modern duplex in new harlem brownstone 1159.0 Entire home/apt No smoking & No
    children under 10 &
No pets           2.0               5           2.0 Knoxville\n
    Brooklyn Sunny room 5 min to subway 793.0 Private room       No visitors &
    No children
under 10           2.0               2           5.0 Knoxville\n Amazing Large
     Sunny Studio in Greenwich Village 144.0 Entire home/apt           No
    parties & No
smoking           30.0               3           4.0 Knoxville\n    Private 1
    Bdrm Suite in Historic Brownstone 479.0 Private room
                        No
visitors           2.0               2           2.0 Knoxville\n Charming
    bedroom with huge terrace in Greenpoint 712.0 Private room No parties & No
    children under 10 & No
visitors           2.0               1           3.0 Knoxville\n
                Huge quiet modern oasis 1165.0 Entire home/apt
                No smoking & No
parties           2.0               8           2.0 Knoxville\nPrivate large
    room near LGA airport with queen bed 552.0 Private room
No pets           1.0               1           2.0 Knoxville\n   12 east 86
    th Steet Upper East Side Building 632.0 Entire home/apt
                        No
smoking           30.0               3           2.0 Knoxville\n   Spacious
    Garden Apt in Clinton Hill/Ft Greene 1006.0 Entire home/apt No children under
    10 & No visitors & No

```
smoking                3.0                3                3.0 Knoxville\n
                                        LES Penthouse 993.0 Entire home/apt        No
    children under 10 & No
parties                2.0                8                4.0 Knoxville'}, {'Description
    ': 'Attractions in Chattanooga', 'Content': "
Name Latitude Longitude                                        Address
    Phone
Website      City\n                                The Chattanooga Zoo at Warner Park
    35.042632 -85.282099   301 N Holtzclaw Ave, Chattanooga, TN 37404, USA (423)
    697-1322
http://www.chattzoo.org/ Chattanooga\n                                        Rock
    City Gardens 34.973392 -85.350177    1400 Patten Rd, Lookout Mountain, GA
    30750, USA (706)
820-2531                                        http://seerockcity.
    com/?utm_source=gmb&utm_medium=organic Chattanooga\n
Tennessee Aquarium 35.055823 -85.311065            1 Broad St, Chattanooga, TN
    37402, USA (423) 265-0695
http://www.tnaqua.org/ Chattanooga\n                                Lookout Mountain
    Incline Railway 35.009546 -85.328564    3917 St Elmo Ave, Chattanooga, TN
    37409, USA (423)
821-4224

    http://www.ridetheincline.com/ Chattanooga\n
Coolidge Park 35.060506 -85.306857            150 River St, Chattanooga, TN 37405,
    USA (423) 643-6311
http://www.chattanooga.gov/public-works/parks/directory-of-park-facilities
    Chattanooga\n                                Creative Discovery Museum 35.052700
    -85.312280
321 Chestnut St, Chattanooga, TN 37402, USA (423) 756-2738

    http://www.cdmfun.org/
Chattanooga\n                                Chattanooga Ghost Tours Inc
    35.051231 -85.309057            57 E 5th St, Chattanooga, TN 37402, USA (423)
    800-5998
https://chattanoogaghosttours.com/ Chattanooga\n
                                        Ross's Landing 35.056741 -85.310302
        201 Riverfront Pkwy, Chattanooga, TN 37402,
USA (423) 643-6311                                http://www.chattanooga.gov
    /public-works/parks/directory-of-park-facilities Chattanooga\n
Hunter Museum of American Art 35.055894 -85.306405    10 Bluff View Ave,
    Chattanooga, TN 37403, USA (423) 267-0968
http://www.huntermuseum.org/ Chattanooga\n
    Walnut Street Bridge 35.058330 -85.307296            1 Walnut St, Chattanooga,
    TN 37403, USA
(423) 643-6096                                http://www.chattanooga.gov/
    public-works/parks/directory-of-park-facilities Chattanooga\n
Tennessee Riverpark 35.095938 -85.245311    4301 Amnicola Hwy, Chattanooga, TN
    37402, USA (423) 209-5370
http://parks.hamiltontn.gov/ Chattanooga\n            Moccasin Bend National
    Archeological District 35.053024 -85.328656 301 Manufacturers Rd, Chattanooga,
    TN 37405, USA
(423) 648-5623                        https://www.nps.gov/chch/learn/historyculture/
    moccasin-bend-national-archeological-district.htm Chattanooga\n
Ruby Falls 35.019084 -85.339355        1720 Scenic Hwy, Chattanooga, TN 37409, USA
    (423) 821-2544
http://www.rubyfalls.com/ Chattanooga\n
                                        Cravens House 35.013782 -85.341495
    Point Park Visitor Ctr, Lookout Mountain, TN 37350, USA
(423) 821-7786
    https://www.nps.gov/places/cravenshouselom.htm Chattanooga\n
Williams Echo Dome 35.054215 -85.304836            430 E 2nd St, Chattanooga, TN
    37403, USA    Unknown
Unknown Chattanooga\n                        Reflection Riding Arboretum & Nature
    Center 35.010130 -85.364585        400 Garden Rd, Chattanooga, TN 37419, USA
    (423) 821-1160
```

https://www.reflectionriding.org/ Chattanooga\n
                                                    Audubon Acres 34.999350 -85.180340
        900 N Sanctuary Rd, Chattanooga, TN 37421,
USA (423) 892-1499
                                                                            http
    ://www.chattanoogaaudubon.org/ Chattanooga\n
Sculpture Fields at Montague Park 35.026175 -85.293624        1800 Polk St,
    Chattanooga, TN 37408, USA (423) 266-7288
http://www.sculpturefields.org/ Chattanooga\nBessie Smith Cultural Center and
    Chattanooga African American Museum 35.044634 -85.306467 200 E M L King Blvd,
    Chattanooga, TN 37403,
USA (423) 266-8658 http://www.bessiesmithcc.org/?utm_source=google&utm_medium=
    wix_google_business_profile&utm_campaign=3948799342172393035 Chattanooga\n
Wooden Horse 35.055836 -85.306713      10 Bluff View Ave, Chattanooga, TN 37403,
    USA   Unknown
Unknown Chattanooga"}, {'Description': 'Restaurants in Chattanooga', 'Content': "
                  Name Average Cost                                    Cuisines
     Aggregate
Rating    City\n          P.F. Chang's         33
    Cafe, Bakery, BBQ, Seafood            4.2 Chattanooga\n            L'amandier
28                Cafe, BBQ, Mediterranean, Seafood        4.3 Chattanooga\n
             Anupam Sweet        82                            Desserts,
    American, BBQ
3.4 Chattanooga\n           Nirula's          64
    French, Desserts, Seafood        2.6 Chattanooga\n Anupam Sweets &
    Restaurant       62
Tea, Cafe, Desserts, Seafood        3.0 Chattanooga\n        Bisque Bakery
         90                    French, Desserts, Seafood         3.9
    Chattanooga\n
Nooba          93                  Tea, Mexican, Bakery, Seafood        3.8
    Chattanooga\n          The Royal        39                    French,
    Pizza,
Bakery, Fast Food        3.3 Chattanooga\n        Curry n Phulka        40
                                 Tea, Chinese, Fast Food        2.8 Chattanooga\n
    DCK-
Dana Choga's Kitchen        67                    Cafe, Pizza, Mediterranean
         3.5 Chattanooga\n          Chaayos        54
Desserts, Cafe, BBQ, Chinese, Seafood        3.3 Chattanooga\n
    Truffles        53     Cafe, Bakery, BBQ, Fast Food, Chinese, American
         3.2
Chattanooga\n           Liquid        15        Tea, Pizza, Bakery,
    Mediterranean, Seafood        4.0 Chattanooga\n           Tpot
         14
Desserts, Tea, Pizza, Mexican, Cafe, Indian        0.0 Chattanooga\n  Sardar A
    Pure Meat Shop        24                 Bakery, Pizza, American
3.4 Chattanooga\n        Warehouse Cafe        90                        Fast
    Food, Pizza, Seafood        3.7 Chattanooga\n        The Beer Cafe
         62
Tea, Pizza, Fast Food        3.8 Chattanooga\n           Subway
    66               Tea, Cafe, Bakery, Desserts        2.4 Chattanooga
    \n
Indi-QUE        57              Tea, Italian, BBQ, Fast Food, Cafe
    3.3 Chattanooga\nEssex Collections Patisserie       48
Chinese, BBQ, Fast Food        3.4 Chattanooga\n        Food Adda
         100     Desserts, Pizza, Fast Food, Mediterranean, Seafood        0.0
    Chattanooga\n
Bikanervala        65                            Cafe, Bakery
         3.2 Chattanooga\n        Havmor Ice Cream        75
    Pizza, Bakery, Fast
Food, Chinese, Seafood        3.6 Chattanooga\n        Moti Mahal Delux
         52        Tea, Pizza, BBQ, Cafe, Mediterranean        2.7
    Chattanooga\n
Aggarwal Sweet India        30                           Tea, Desserts
         2.8 Chattanooga\n        Jack Po!tato's        63
             Tea,

Bakery, Mediterranean, Seafood          3.3 Chattanooga\n          Muradabadi
         16                          Cafe, Pizza, Bakery, BBQ          3.0
Chattanooga\n       Vishal Restaurant          27                     Tea,
    Pizza, BBQ, Seafood          3.1 Chattanooga\n          Raj Sweets
         69
Desserts, Tea, BBQ, Cafe, Mediterranean          2.8 Chattanooga\n          Punjabee's
    Darbar          41          Tea, French, Bakery, Fast Food, Chinese
         3.0
Chattanooga\n          Basil Tree          20          Bakery, Indian,
    American, Desserts          2.4 Chattanooga\n          Domino's Pizza
         77
Bakery, Pizza, American, BBQ          2.8 Chattanooga\n          Pizza Hut Delivery
         25                     Tea, Cafe, American, Indian          3.9
    Chattanooga\n
Ichiban          87                          Mexican, Bakery, Desserts
    3.9 Chattanooga\n          Habibi          95
Cafe, BBQ, Seafood          3.7 Chattanooga\n          Cook Du Kdu          69
    Pizza, Fast Food, Cafe, Indian, Mediterranean, Seafood          3.1 Chattanooga\n
Go Krazy          64                          French, BBQ, Fast Food
    3.5 Chattanooga\n          Bentoya          41                     Tea,
    Fast
Food, Bakery, Seafood          3.8 Chattanooga\n          Burger's King
    16                          Tea, Bakery, Desserts          4.1 Chattanooga
"},
{'Description': 'Accommodations in Chattanooga', 'Content': '
                          NAME price     room type
                     house_rules minimum
nights maximum occupancy review rate number    city\n    Affordable Private
    Spacious Room in Brooklyn 790.0 Private room                     No
    parties
2.0              2          3.0 Chattanooga\n
    Sunny One Bedroom 722.0 Entire home/apt       No pets & No children under 10
         1.0
5               3.0 Chattanooga\nUpper West / Morningside Heights Apt, Near Subway
    290.0 Entire home/apt                No visitors          3.0
              3
3.0 Chattanooga\n          Sunny room+Pvte office in huge loft 728.0 Private room
                   No parties          4.0          2
              5.0
Chattanooga\n     Extra Cozy Room in Center of Williamsburg 1033.0 Private room
                   No pets          1.0          1
         1.0
Chattanooga\n   Luxury & Charm. Steps from Christopher Park! 301.0 Entire home/apt
       No pets & No visitors & No parties          5.0          2
         4.0
Chattanooga\n                     Fort Greene Room 722.0  Private room
         No visitors & No children under 10          2.0          1
         2.0
Chattanooga\n                                   973.0
    Entire home/apt                     No pets          10.0
              6
4.0 Chattanooga\n     4 bdrm/2 bath apt. Central Pk, Columbia U. 564.0 Entire home/
    apt          No visitors & No pets          30.0          4
         3.0
Chattanooga\n     Cozy room in Bushwick- 15 min to the city 933.0 Private room
              No parties & No pets          1.0          2
         2.0
Chattanooga\n          Modern apartment w/ gorgeous view 921.0 Private room
                   No pets          3.0          2
         1.0
Chattanooga\n     Artsy Private BR in Fort Greene Cumberland 727.0 Private room
              No visitors          30.0          1
         3.0

Chattanooga\n            Studio Deluxe 1 - Wyndham Midtown 45 848.0 Private room No pets & No smoking & No children under 10      3.0                1               2.0
Chattanooga\n      Trendy and Stylish Downtown apartment !!! 110.0 Entire home/apt                  No pets          3.0                2               2.0
Chattanooga\n    Sonder | 180 Water | Luxurious 1BR + Rooftop 676.0 Entire home/apt                  No pets & No smoking          29.0                5       3.0
Chattanooga'}, {'Description': 'Flight from Houston to Nashville on 2022-03-21', 'Content': 'Flight Number Price DepTime ArrTime ActualElapsedTime FlightDate OriginCityName DestCityName
Distance\n   F3827247   241   14:22   16:10 1 hours 48 minutes 2022-03-21    Houston    Nashville   657.0\n   F3827724   147   18:35   20:33 1 hours 58 minutes 2022-03-21
Houston   Nashville   657.0\n   F3827820   157   12:07   14:02 1 hours 55 minutes 2022-03-21    Houston   Nashville   657.0\n   F3827930   305   07:36   09:41 2 hours 5
minutes 2022-03-21    Houston   Nashville   657.0\n   F3956407   264   11:38   13:15 1 hours 37 minutes 2022-03-21    Houston   Nashville   670.0\n   F3956408   246
08:08   09:47 1 hours 39 minutes 2022-03-21    Houston   Nashville   670.0\n   F3956409   145   17:36   19:12 1 hours 36 minutes 2022-03-21    Houston   Nashville   670.0\n
F3956532   225   07:14   08:52 1 hours 38 minutes 2022-03-21    Houston   Nashville   657.0\n   F3956533   292   11:42   13:19 1 hours 37 minutes 2022-03-21   Houston
Nashville   657.0\n   F3956534   248   18:20   20:06 1 hours 46 minutes 2022-03-21   Houston   Nashville   657.0\n   F4038756   272   10:11   11:52 1 hours 41 minutes
2022-03-21    Houston   Nashville   657.0\n   F4038966   149   16:33   18:13 1 hours 40 minutes 2022-03-21    Houston   Nashville   657.0'}, {'Description': 'Self-driving
from Houston to Nashville', 'Content': 'self-driving, from Houston to Nashville, duration: 11 hours 50 mins, distance: 1,253 km, cost: 62'}, {'Description': 'Taxi from Houston to
Nashville', 'Content': 'taxi, from Houston to Nashville, duration: 11 hours 50 mins, distance: 1,253 km, cost: 1253'}, {'Description': 'Flight from Nashville to Knoxville on 2022-03-23',
'Content': 'There is no flight from Nashville to Knoxville on 2022-03-23.'}, {'Description': 'Self-driving from Nashville to Knoxville', 'Content': 'self-driving, from Nashville to
Knoxville, duration: 2 hours 42 mins, distance: 290 km, cost: 14'}, {'Description': 'Taxi from Nashville to Knoxville', 'Content': 'taxi, from Nashville to Knoxville, duration: 2 hours 42
mins, distance: 290 km, cost: 290'}, {'Description': 'Flight from Knoxville to Chattanooga on 2022-03-25', 'Content': 'There is no flight from Knoxville to Chattanooga on 2022-03-25.'},
{'Description': 'Self-driving from Knoxville to Chattanooga', 'Content': 'self-driving, from Knoxville to Chattanooga, duration: 1 hour 41 mins, distance: 180 km, cost: 9'}, {'Description':
'Taxi from Knoxville to Chattanooga', 'Content': 'taxi, from Knoxville to Chattanooga, duration: 1 hour 41 mins, distance: 180 km, cost: 180'}, {'Description': 'Flight from Chattanooga to
Houston on 2022-03-27', 'Content': 'There is no flight from Chattanooga to Houston on 2022-03-27.'}, {'Description': 'Self-driving from Chattanooga to Houston', 'Content': 'self-driving,
from Chattanooga to Houston, duration: 11 hours 47 mins, distance: 1,309 km, cost: 65'}, {'Description': 'Taxi from Chattanooga to Houston', 'Content': 'taxi, from Chattanooga to Houston,
duration: 11 hours 47 mins, distance: 1,309 km, cost: 1309'}]
Query: Could you design a one-week travel itinerary for two, departing from Houston and touring three cities in Tennessee from March 21st to March 27th, 2022? Our budget is now $8,200. We

require accommodations that allow smoking and should ideally be private rooms. As
    for transportation, we would prefer not to self-drive.
Travel Plan (please only output the JSON string without explanatory information):

==============RESPONSE OF o1-preview=================

```
[
    {
        "day": 1,
        "people_number": 2,
        "current_city": "from Houston to Nashville",
        "transportation": "Flight Number: F3827820, from Houston to Nashville,
            Departure Time: 12:07, Arrival Time: 14:02",
        "breakfast": "-",
        "attraction": "Country Music Hall of Fame and Museum, Nashville",
        "lunch": "Cafe Arabelle, Nashville",
        "dinner": "Les 3 Brasseurs, Nashville",
        "accommodation": "FiDi Cozy room overlooking East River, Nashville"
    },
    {
        "day": 2,
        "people_number": 2,
        "current_city": "Nashville",
        "transportation": "-",
        "breakfast": "Burger Planet, Nashville",
        "attraction": "Johnny Cash Museum; Centennial Park, Nashville",
        "lunch": "Mamagoto, Nashville",
        "dinner": "Oh! Calcutta, Nashville",
        "accommodation": "FiDi Cozy room overlooking East River, Nashville"
    },
    {
        "day": 3,
        "people_number": 2,
        "current_city": "from Nashville to Knoxville",
        "transportation": "Taxi, from Nashville to Knoxville, Departure Time: 09:00,
            Arrival Time: 11:42, Duration: 2 hours 42 minutes, Cost: $290",
        "breakfast": "Cafe El Chico, Nashville",
        "attraction": "World's Fair Park, Knoxville",
        "lunch": "Ali Baba & 41 Dishes, Knoxville",
        "dinner": "Sky On 57, Knoxville",
        "accommodation": "Light-filled Room in Renovated Apt, Knoxville"
    },
    {
        "day": 4,
        "people_number": 2,
        "current_city": "Knoxville",
        "transportation": "-",
        "breakfast": "Chaat Corner, Knoxville",
        "attraction": "Ijams Nature Center; Sunsphere, Knoxville",
        "lunch": "Anand Sweets, Knoxville",
        "dinner": "La-Nawaab, Knoxville",
        "accommodation": "Light-filled Room in Renovated Apt, Knoxville"
    },
    {
        "day": 5,
        "people_number": 2,
        "current_city": "from Knoxville to Chattanooga",
        "transportation": "Taxi, from Knoxville to Chattanooga, Departure Time:
            09:00, Arrival Time: 10:41, Duration: 1 hour 41 minutes, Cost: $180",
        "breakfast": "Anand Sweets, Knoxville",
        "attraction": "Tennessee Aquarium, Chattanooga",
        "lunch": "Anupam Sweet, Chattanooga",
        "dinner": "Bisque Bakery, Chattanooga",
        "accommodation": "Trendy and Stylish Downtown apartment !!!, Chattanooga"
    },
```

```
     {
         "day": 6,
         "people_number": 2,
         "current_city": "Chattanooga",
         "transportation": "-",
         "breakfast": "Anupam Sweets & Restaurant, Chattanooga",
         "attraction": "Rock City Gardens; Ruby Falls, Chattanooga",
         "lunch": "Chaayos, Chattanooga",
         "dinner": "L'amandier, Chattanooga",
         "accommodation": "Trendy and Stylish Downtown apartment !!!, Chattanooga"
     },
     {
         "day": 7,
         "people_number": 2,
         "current_city": "from Chattanooga to Houston",
         "transportation": "Taxi, from Chattanooga to Houston, Departure Time: 08:00,
             Arrival Time: 19:47, Duration: 11 hours 47 minutes, Cost: $1309",
         "breakfast": "Bisque Bakery, Chattanooga",
         "attraction": "Coolidge Park, Chattanooga",
         "lunch": "Nooba, Chattanooga",
         "dinner": "-",
         "accommodation": "-"
     }
]
=============FEEDBACK FROM METACONTROLLER================
By inspecting your plan we find the following issue. Please refine your plan
    according to the feedback below:
1. The room type should be private room :-> Trendy and Stylish Downtown apartment
    !!!
2. The restaurant in day 5 breakfast is repeated.
3. The accommodation Trendy and Stylish Downtown apartment !!!, Chattanooga do not
    obey the minumum nights rule.
4. The lunch in day 1 is invalid or not in the data provided.
```

## O.2  BACK PROMPT

You are a proficient planner. Based on the provided information and query, please
    give me a detailed plan, including specifics such as flight numbers (e.g.,
    F0123456), restaurant names, and
accommodation names. Note that all the information in your plan should be derived
    from the provided data. You should give a travel plan in JSON format as shown
    in the example below.
Additionally, all details should align with commonsense. The symbol '-' indicates
    that information is unnecessary. For example, in the provided sample, you do
    not need to plan after
returning to the departure city. When you travel to two cities in one day, you
    should note it in the 'current_city' section as in the example (i.e., from A to
    B).

```
***** Example *****
Query: Could you create a travel plan for 7 people from Ithaca to Charlotte spanning
    3 days, from March 8th to March 14th, 2022, with a budget of $30,200?
Travel Plan:
[
     {
         "day": 1,
         "people_number": 7,
         "current_city": "from Ithaca to Charlotte",
         "transportation": "Flight Number: F3633413, from Ithaca to Charlotte,
             Departure Time: 05:38, Arrival Time: 07:46",
         "breakfast": "Nagaland's Kitchen, Charlotte",
         "attraction": "The Charlotte Museum of History, Charlotte",
         "lunch": "Cafe Maple Street, Charlotte",
         "dinner": "Bombay Vada Pav, Charlotte",
```

```
        "accommodation": "Affordable Spacious Refurbished Room in Bushwick!,
            Charlotte"
    },
    {
        "day": 2,
        "people_number": 7,
        "current_city": "Charlotte",
        "transportation": "-",
        "breakfast": "Olive Tree Cafe, Charlotte",
        "attraction": "The Mint Museum, Charlotte;Romare Bearden Park, Charlotte",
        "lunch": "Birbal Ji Dhaba, Charlotte",
        "dinner": "Pind Balluchi, Charlotte",
        "accommodation": "Affordable Spacious Refurbished Room in Bushwick!,
            Charlotte"
    },
    {
        "day": 3,
        "people_number": 7,
        "current_city": "from Charlotte to Ithaca",
        "transportation": "Flight Number: F3786167, from Charlotte to Ithaca,
            Departure Time: 21:42, Arrival Time: 23:26",
        "breakfast": "Books Monument, Charlotte",
        "attraction": "Books Monument, Charlotte",
        "lunch": "Olive Tree Cafe, Charlotte",
        "dinner": "Kylin Skybar, Charlotte",
        "accommodation": "-"
    }
]
***** Example Ends *****

Given information: [{'Description': 'Attractions in Nashville', 'Content': '
                        Name Latitude Longitude
                                        Address
Phone
Website    City\nCountry Music Hall of Fame and Museum 36.158263 -86.776126 222 Rep.
    John Lewis Way S, Nashville, TN 37203, USA (615) 416-2001
https://countrymusichalloffame.org/ Nashville\n    Nashville Zoo at Grassmere
    36.089705 -86.742096 3777 Nolensville Pk, Nashville, TN 37211, USA (615)
    833-1534
http://www.nashvillezoo.org/ Nashville\n Belle Meade Historic Site & Winery
    36.104916 -86.864695 5025 Harding Pike, Nashville, TN 37205, USA (615) 356-0501
https://visitbellemeade.com/ Nashville\n        Johnny Cash Museum 36.160939
    -86.775757      119 3rd Ave S, Nashville, TN 37201, USA (615) 256-1777
http://www.johnnycashmuseum.com/ Nashville\n           Centennial Park
    36.148946 -86.812750    2500 West End Ave, Nashville, TN 37203, USA (615)
    862-8400
https://www.nashville.gov/Parks-and-Recreation/Parks/Centennial-Park.aspx Nashville\
    n         Grand Ole Opry 36.206857 -86.692108    600 Opry Mills Dr,
    Nashville, TN
37214, USA (615) 871-6779
https://www.opry.com/?utm_campaign=opry&utm_medium=organicsearch&utm_source=
    googlemybusiness&utm_audience=tofu_googlemybusiness&utm_content=
    brandstory_google_my_business_website_link
Nashville\n            Frist Art Museum 36.157897 -86.783853         919
    Broadway, Nashville, TN 37203, USA (615) 244-3340
https://fristartmuseum.org/ Nashville\n            The Parthenon 36.149674
    -86.813347    2500 West End Ave, Nashville, TN 37203, USA (615) 862-8431
https://www.nashvilleparthenon.com/ Nashville\n         Nashville Shores
    36.158209 -86.604915        4001 Bell Rd, Hermitage, TN 37076, USA (615)
    889-7050
http://www.nashvilleshores.com/lodging Nashville\n Musicians Hall of Fame and Museum
    36.167668 -86.782399      401 Gay St, Nashville, TN 37219, USA (615) 244-3263
https://www.musicianshalloffame.com/ Nashville\n Bicentennial Capitol Mall State
    Park 36.170887 -86.787589 600 James Robertson Pkwy, Nashville, TN 37243, USA
    (888) 867-2757
```

https://tnstateparks.com/parks/bicentennial-mall Nashville\n          Lane Motor Museum 36.140197 -86.734580 702 Murfreesboro Pike, Nashville, TN 37210, USA (615) 742-7445

http://www.lanemotormuseum.org/ Nashville\n          Honky Tonk Highway 36.160393 -86.778432      501 Broadway, Nashville, TN 37203, USA (800) 657-6910

https://www.visitmusiccity.com/visitors/honkytonkhighway Nashville\n     Adventure Science Center 36.146614 -86.775481 800 Fort Negley Blvd, Nashville, TN 37203, USA (615) 862-5160

http://www.adventuresci.org/ Nashville\n          Patsy Cline Museum 36.160904 -86.776030     119 3rd Ave S, Nashville, TN 37201, USA (615) 454-4722

https://www.patsymuseum.com/ Nashville\n     Andrew Jacksons Hermitage 36.213757 -86.615310     4580 Rachels Ln, Hermitage, TN 37076, USA (615) 889-2941

http://www.thehermitage.com/ Nashville\n          NashTrash Tours 36.169474 -86.788254 900 Rosa L Parks Blvd, Nashville, TN 37208, USA (615) 226-7300

http://www.nashtrash.com/ Nashville\n          Fort Nashborough 36.164268 -86.775430     170 1st Ave N, Nashville, TN 37201, USA (615) 862-8400

https://www.nashville.gov/Parks-and-Recreation/Historic-Sites/Fort-Nashborough.aspx Nashville\n          Ryman Auditorium 36.161248 -86.778471     116 5th Ave N, Nashville, TN 37219, USA (615) 889-3060

https://www.ryman.com/?utm_campaign=ryman&utm_medium=organicsearch&utm_source= googlemybusiness&utm_audience=tofu_googlemybusiness&utm_content= brandstory_google_my_business_website_link

Nashville\n     Nashville Public Square Park 36.166611 -86.778126 Union St & 3rd Ave N, Nashville, TN 37201, USA (615) 743-3090

https://www.nashvilledowntown.com/go/public-square-park Nashville'}, {'Description': 'Restaurants in Nashville', 'Content': "          Name Average Cost Cuisines Aggregate Rating   City\n          Bangkok 1     44          Cafe, Seafood     3.3 Nashville\n          Bablu Fast Food     13   Desserts, Pizza, Mexican, BBQ, Fast Food     0.0 Nashville\n          Full Dabba     77   Pizza, Mexican, Fast Food, Cafe, American

0.0 Nashville\n          Twigly     14 Pizza, French, Fast Food, Chinese, Seafood     4.5 Nashville\n          Veg Hut     35 Tea, Mexican, Seafood     0.0 Nashville\n          The Toddy Shop     92          Desserts, Fast Food     3.4 Nashville\n GoGourmet     34     Tea, Cafe, Indian, Mediterranean     3.7 Nashville\n     Govinda's Confectionery     15     Tea, Bakery, Mediterranean, Fast Food

3.1 Nashville\n          Kitchen King     13   Pizza, Italian, BBQ, Cafe, Mediterranean     0.0 Nashville\n          Town Hall     55 Tea, Seafood     3.8 Nashville\n     Smoke House Deli     59          Cafe, Mediterranean, Seafood     4.0 Nashville\n     Dialogue Lounge & Caf     83   Tea, Indian, BBQ, Cafe, American, Seafood     3.0 Nashville\n     Meenakshi Bhawan     24          Tea, Desserts     3.1 Nashville\n          Oh! Calcutta     83     Tea, Mexican, BBQ, Cafe, Indian     4.4 Nashville\n     Chicago Pizza     20          Tea, French, Pizza, BBQ     3.2 Nashville\n     Malhotra Restaurant     76          Tea, Desserts, Seafood

2.9 Nashville\n          Madras Cafe     88          Pizza, Desserts, Seafood     0.0 Nashville\nDakshin - Sheraton New Delhi Hotel     63 Cafe, Pizza, Indian, Bakery     4.0 Nashville\n     1911 Bar - The Imperial     76 Desserts, Pizza, Italian, American, Seafood     3.2 Nashville\n Punjabi Virsa     35     Desserts, Italian, BBQ, Cafe, Seafood     0.0 Nashville\n          Sagar Dhaba     55     Tea, Bakery, BBQ, Cafe,

Mediterranean                 0.0 Nashville\n                    Kettle & Kegs            60
                         Tea, Seafood          0.0 Nashville\n
Kargo          84                  Tea, Pizza, Desserts          4.4 Nashville
    "}, {'Description': 'Accommodations in Nashville', 'Content': '
NAME price      room type                             house_rules minimum nights
    maximum occupancy review rate number  city\nHuge 2 Bedroom, Great Location,
    Express Metro
745.0 Entire home/apt No smoking & No children under 10 & No pets     4.0
                     4                    5.0 Nashville\n Clean and large bedroom in a
    private house 474.0
Private room                                   No smoking           1.0             1
                    4.0 Nashville\n                   Brooklyn Heights gem 993.0
    Entire
home/apt                                 No pets         2.0             6
                    5.0 Nashville\n     Lovely room in heart of Williamsburg 61.0
    Private room
No pets         2.0             1               4.0 Nashville\n *Light & Love*
     vibrant, historic, sleeps 4 679.0 Entire home/apt                    No
     pets
5.0                 2                 2.0 Nashville\n    FiDi Cozy room overlooking
    East River 870.0 Private room                    No parties
    1.0
2              5.0 Nashville\n    Charming 1BR with sun-nook in Brooklyn 256.0
    Entire home/apt No children under 10 & No pets & No smoking  3.0
                  2
4.0 Nashville\n         Cozy bedroom close to Manhattan 576.0 Private room
                    No children under 10           1.0             1
               3.0
Nashville'}, {'Description': 'Attractions in Knoxville', 'Content': "
                         Name Latitude Longitude
                                       Address
Phone

    Website     City\n                       World's Fair Park 35.962577 -83.924192
525 Henley St, Knoxville, TN 37902, USA (865) 215-1158
                                                     http://
    worldsfairpark.org/ Knoxville\n
Knoxville Museum of Art 35.962426 -83.925229 1050 Worlds Fair Park Dr, Knoxville, TN
     37916, USA (865) 525-6101
http://www.knoxart.org/ Knoxville\n                       Sunsphere 35.961707
    -83.923353        810 Clinch Ave, Knoxville, TN 37902, USA (865) 314-0660
http://www.sunspheretickets.com/ Knoxville\n              Ijams Nature Center
    35.956454 -83.866775    2915 Island Home Ave, Knoxville, TN 37920, USA (865)
    577-4717
http://www.ijams.org/ Knoxville\n           Knoxville Walking Tours 35.966448
    -83.919167         301 S Gay St, Knoxville, TN 37902, USA (865) 309-4522
http://knoxvillewalkingtours.com/ Knoxville\n                  Muse Knoxville
    35.997617 -83.885467        516 N Beaman St, Knoxville, TN 37914, USA (865)
    594-1494
http://www.themuseknoxville.org/ Knoxville\n Knoxville Botanical Garden and
    Arboretum 35.982160 -83.881077    2743 Wimpole Ave, Knoxville, TN 37914, USA
    (865) 862-8717
http://www.knoxgarden.org/ Knoxville\n        Haunted Knoxville Ghost Tours
    35.965963 -83.919553 36 Market Square #1404, Knoxville, TN 37902, USA (865)
    377-9677
http://www.hauntedknoxville.net/ Knoxville\n               Three Rivers Rambler
    35.952810 -83.940376 2560 University Commons Way, Knoxville, TN 37919, USA (865)
     524-9411
http://www.threeriversrambler.com/ Knoxville\n               Charles Krutch Park
    35.964414 -83.918695         504 Market St, Knoxville, TN 37902, USA (865)
    215-4248
https://www.knoxvilletn.gov/government/city_departments_offices/parks_and_recreation
    /parks/krutch_park Knoxville\n McClung Museum of Natural History & Culture
    35.952005 -83.927209

2808 1327 Cir Park Dr, Knoxville, TN 37996, USA (865) 974-2144
http://mcclungmuseum
.utk.edu/ Knoxville\n
Knoxville Sightseeing 35.992790 -83.904128        2519 Mitchell St, Knoxville, TN
37917, USA (865) 566-0634
Unknown Knoxville\n        Chilhowee Park & Exposition Center 35.996296 -83.884032
3301 E Magnolia Ave, Knoxville, TN 37914, USA (865) 215-1450
https://chilhoweepark.com/ Knoxville\n                Augusta Quarry
35.944772 -83.911886 3000 Fort Dickerson Rd SW, Knoxville, TN 37920, USA
Unknown
Unknown Knoxville\n                Zoo Knoxville 35.999812 -83.888250
3500 Knoxville Zoo Dr, Knoxville, TN 37914, USA (865) 637-5331
https://www.zooknoxville.org/ Knoxville\n    Outdoor Knoxville Adventure Center
35.961902 -83.912663 900 Volunteer Landing Ln, Knoxville, TN 37915, USA (865)
228-8424
http://www.outdoorknoxville.com Knoxville\n        James White Fort Association
35.962962 -83.912169        205 E Hill Ave, Knoxville, TN 37915, USA (865)
525-6514
http://jameswhitesfort.org/ Knoxville\n    Rowing Man Statue in Knoxville, TN
35.963611 -83.917519 W. Church Avenue &, S Gay St, Knoxville, TN 37902, USA
Unknown
Unknown Knoxville\n                UT Gardens Knoxville 35.944000 -83.938260
2518 Jacob Dr, Knoxville, TN 37996, USA (865) 974-7324
http://utgardens.tennessee.edu/locations/knoxville Knoxville\nEast Tennessee
Historical Society and Museum 35.964117 -83.917794   601 S Gay St, Knoxville, TN
37902, USA (865)
215-8830                                http://www.
easttnhistory.org/ Knoxville"}, {'Description': 'Restaurants in Knoxville', '
Content': '
Name  Average Cost                                Cuisines Aggregate Rating
City\n        Cafe Arabelle        29                French, BBQ, Desserts,
Seafood
3.6 Knoxville\n     Les 3 Brasseurs        24        Desserts, Pizza, Italian,
Cafe, Indian        4.6 Knoxville\n        Sky On 57        150
Tea,
Cafe, Pizza, Chinese, Seafood        3.4 Knoxville\n        Cafe El Chico
67                Tea, French, Indian, Seafood        3.6
Knoxville\n
TcozY        85        Tea, Pizza, Mexican, Fast Food, Cafe        0.0
Knoxville\nRama Vaishnav Bhojnalaya        46                Tea, Cafe, Pizza
, Desserts
3.0 Knoxville\n        Burger Planet        45                Tea, Bakery,
American, Cafe        3.2 Knoxville\n    Cafe Coffee Day        82
Fast
Food, American, BBQ, Italian        2.8 Knoxville\n        Mamagoto
14        Indian, Mediterranean, Desserts, Seafood        4.1
Knoxville\n  Ali Baba &
41 Dishes        70                Pizza, Desserts, Fast Food        3.5
Knoxville\n    Punjabi Flavour        41        Tea, Bakery, Cafe,
Indian, Seafood
0.0 Knoxville\n        Tandoori Tadka        23        Chinese, Pizza, Desserts,
Fast Food        0.0 Knoxville\n    Coalition Cafe        35
Desserts,
Tea, BBQ, Cafe, Mediterranean        3.4 Knoxville\n        Chaat Corner
67                Cafe, Pizza        3.2
Knoxville\n
El Posto        63                Tea, Cafe, Pizza, BBQ        3.5
Knoxville\n        La-Nawaab        17                French, Bakery,
Cafe, Seafood
3.0 Knoxville\n    Peppers & Pipes        86                Chinese, Pizza,
Cafe, Italian        3.1 Knoxville\n        Chit Chat        15
Pizza, Fast Food        3.1 Knoxville\n        Anand Sweets        45
Fast Food, Pizza, Indian, Seafood        0.0 Knoxville\n
Open Kitchen

```
13 Pizza, Bakery, BBQ, Cafe, Indian, Mediterranean        3.0 Knoxville\n
    Biryani By Kilo          17                         Chinese, Pizza, Fast Food
                4.1
Knoxville\n   Chinese Food Corner        30                    Indian, Desserts,
    Fast Food             0.0 Knoxville\nThe Indian Kaffe Express      71
Tea, Desserts           3.8 Knoxville'}, {'Description': 'Accommodations in
    Knoxville', 'Content': '                                   NAME price      room
    type
house_rules minimum nights maximum occupancy review rate number  city\n   Cozy
    Private Room in Chinatown/ Lower East Side 132.0 Private room
                            No pets &
No visitors            7.0                2              4.0 Knoxville\n
                                The Diamond Room 1008.0 Private room
                        No parties &
No visitors            1.0                1              5.0 Knoxville\n
                Light-filled Room in Renovated Apt 310.0 Private room
No pets           2.0                2                 2.0 Knoxville\n
                                Private Room 922.0   Private room
                                No
visitors           1.0                1              4.0 Knoxville\n
    Beautiful & Private Manhattan Room 721.0 Private room                   No
    parties & No
smoking            1.0                1              2.0 Knoxville\n          Lg
    Quiet Artist Home -Ditmas Park - 561.0 Entire home/apt      No visitors & No
    pets & No
smoking           24.0                4              2.0 Knoxville\n Beautiful 1-
    bdrm apt in tranquil Inwood building 1091.0 Entire home/apt            No
    smoking &
No pets           2.0                4              4.0 Knoxville\n 1,600sq ft
    modern duplex in new harlem brownstone 1159.0 Entire home/apt No smoking & No
    children under 10 &
No pets           2.0                5              2.0 Knoxville\n
    Brooklyn Sunny room 5 min to subway 793.0 Private room        No visitors &
    No children
under 10          2.0                2              5.0 Knoxville\n Amazing Large
     Sunny Studio in Greenwich Village 144.0 Entire home/apt             No
    parties & No
smoking           30.0                3              4.0 Knoxville\n    Private 1
    Bdrm Suite in Historic Brownstone 479.0 Private room
                                        No
visitors           2.0                2              2.0 Knoxville\n Charming
    bedroom with huge terrace in Greenpoint 712.0 Private room No parties & No
    children under 10 & No
visitors           2.0                1              3.0 Knoxville\n
                        Huge quiet modern oasis 1165.0 Entire home/apt
                    No smoking & No
parties           2.0                8              2.0 Knoxville\nPrivate large
    room near LGA airport with queen bed 552.0 Private room
No pets           1.0                1              2.0 Knoxville\n   12 east 86
    th Steet Upper East Side Building 632.0 Entire home/apt
                                        No
smoking           30.0                3              2.0 Knoxville\n   Spacious
    Garden Apt in Clinton Hill/Ft Greene 1006.0 Entire home/apt No children under
    10 & No visitors & No
smoking           3.0                3              3.0 Knoxville\n
                                LES Penthouse 993.0 Entire home/apt        No
    children under 10 & No
parties           2.0                8              4.0 Knoxville'}, {'Description
    ': 'Attractions in Chattanooga', 'Content': "
Name Latitude Longitude                                      Address
    Phone
Website      City\n                          The Chattanooga Zoo at Warner Park
    35.042632 -85.282099   301 N Holtzclaw Ave, Chattanooga, TN 37404, USA (423)
    697-1322
```

http://www.chattzoo.org/ Chattanooga\n                                                    Rock
     City Gardens 34.973392 -85.350177     1400 Patten Rd, Lookout Mountain, GA
     30750, USA (706)
820-2531                                                     http://seerockcity.
     com/?utm_source=gmb&utm_medium=organic Chattanooga\n
Tennessee Aquarium 35.055823 -85.311065                1 Broad St, Chattanooga, TN
     37402, USA (423) 265-0695
http://www.tnaqua.org/ Chattanooga\n                               Lookout Mountain
     Incline Railway 35.009546 -85.328564     3917 St Elmo Ave, Chattanooga, TN
     37409, USA (423)
821-4224

     http://www.ridetheincline.com/ Chattanooga\n
Coolidge Park 35.060506 -85.306857             150 River St, Chattanooga, TN 37405,
     USA (423) 643-6311
http://www.chattanooga.gov/public-works/parks/directory-of-park-facilities
     Chattanooga\n                               Creative Discovery Museum 35.052700
     -85.312280
321 Chestnut St, Chattanooga, TN 37402, USA (423) 756-2738

     http://www.cdmfun.org/
Chattanooga\n                                  Chattanooga Ghost Tours Inc
     35.051231 -85.309057          57 E 5th St, Chattanooga, TN 37402, USA (423)
     800-5998
https://chattanoogaghosttours.com/ Chattanooga\n
                                         Ross's Landing 35.056741 -85.310302
      201 Riverfront Pkwy, Chattanooga, TN 37402,
USA (423) 643-6311                                   http://www.chattanooga.gov
     /public-works/parks/directory-of-park-facilities Chattanooga\n
Hunter Museum of American Art 35.055894 -85.306405     10 Bluff View Ave,
     Chattanooga, TN 37403, USA (423) 267-0968
http://www.huntermuseum.org/ Chattanooga\n
     Walnut Street Bridge 35.058330 -85.307296          1 Walnut St, Chattanooga,
     TN 37403, USA
(423) 643-6096                                    http://www.chattanooga.gov/
     public-works/parks/directory-of-park-facilities Chattanooga\n
Tennessee Riverpark 35.095938 -85.245311     4301 Amnicola Hwy, Chattanooga, TN
     37402, USA (423) 209-5370
http://parks.hamiltontn.gov/ Chattanooga\n                 Moccasin Bend National
     Archeological District 35.053024 -85.328656 301 Manufacturers Rd, Chattanooga,
     TN 37405, USA
(423) 648-5623                       https://www.nps.gov/chch/learn/historyculture/
     moccasin-bend-national-archeological-district.htm Chattanooga\n
Ruby Falls 35.019084 -85.339355          1720 Scenic Hwy, Chattanooga, TN 37409, USA
     (423) 821-2544
http://www.rubyfalls.com/ Chattanooga\n
                                         Cravens House 35.013782 -85.341495
     Point Park Visitor Ctr, Lookout Mountain, TN 37350, USA
(423) 821-7786
     https://www.nps.gov/places/cravenshouselom.htm Chattanooga\n
Williams Echo Dome 35.054215 -85.304836             430 E 2nd St, Chattanooga, TN
     37403, USA     Unknown
Unknown Chattanooga\n                     Reflection Riding Arboretum & Nature
     Center 35.010130 -85.364585       400 Garden Rd, Chattanooga, TN 37419, USA
     (423) 821-1160
https://www.reflectionriding.org/ Chattanooga\n
                                         Audubon Acres 34.999350 -85.180340
      900 N Sanctuary Rd, Chattanooga, TN 37421,
USA (423) 892-1499
                                                                              http
     ://www.chattanoogaaudubon.org/ Chattanooga\n
Sculpture Fields at Montague Park 35.026175 -85.293624          1800 Polk St,
     Chattanooga, TN 37408, USA (423) 266-7288

http://www.sculpturefields.org/ Chattanooga\nBessie Smith Cultural Center and
    Chattanooga African American Museum 35.044634 -85.306467 200 E M L King Blvd,
    Chattanooga, TN 37403,
USA (423) 266-8658 http://www.bessiesmithcc.org/?utm_source=google&utm_medium=
    wix_google_business_profile&utm_campaign=3948799342172393035 Chattanooga\n
Wooden Horse 35.055836 -85.306713        10 Bluff View Ave, Chattanooga, TN 37403,
    USA    Unknown
Unknown Chattanooga"}, {'Description': 'Restaurants in Chattanooga', 'Content': "
                Name Average Cost                                        Cuisines
    Aggregate
Rating    City\n              P.F. Chang's          33
    Cafe, Bakery, BBQ, Seafood            4.2 Chattanooga\n               L'amandier
28                    Cafe, BBQ, Mediterranean, Seafood          4.3 Chattanooga\n
            Anupam Sweet        82                             Desserts,
    American, BBQ
3.4 Chattanooga\n            Nirula's          64
    French, Desserts, Seafood          2.6 Chattanooga\n Anupam Sweets &
    Restaurant      62
Tea, Cafe, Desserts, Seafood          3.0 Chattanooga\n          Bisque Bakery
        90                          French, Desserts, Seafood          3.9
    Chattanooga\n
Nooba          93                    Tea, Mexican, Bakery, Seafood          3.8
    Chattanooga\n          The Royal          39                    French,
    Pizza,
Bakery, Fast Food          3.3 Chattanooga\n          Curry n Phulka          40
                    Tea, Chinese, Fast Food          2.8 Chattanooga\n
    DCK-
Dana Choga's Kitchen          67                    Cafe, Pizza, Mediterranean
        3.5 Chattanooga\n          Chaayos          54
Desserts, Cafe, BBQ, Chinese, Seafood          3.3 Chattanooga\n
    Truffles          53          Cafe, Bakery, BBQ, Fast Food, Chinese, American
        3.2
Chattanooga\n              Liquid          15          Tea, Pizza, Bakery,
    Mediterranean, Seafood          4.0 Chattanooga\n              Tpot
        14
Desserts, Tea, Pizza, Mexican, Cafe, Indian          0.0 Chattanooga\n   Sardar A
    Pure Meat Shop          24                    Bakery, Pizza, American
3.4 Chattanooga\n          Warehouse Cafe          90                    Fast
    Food, Pizza, Seafood          3.7 Chattanooga\n          The Beer Cafe
        62
Tea, Pizza, Fast Food          3.8 Chattanooga\n              Subway
    66                    Tea, Cafe, Bakery, Desserts          2.4 Chattanooga
    \n
Indi-QUE          57              Tea, Italian, BBQ, Fast Food, Cafe
    3.3 Chattanooga\nEssex Collections Patisserie          48
Chinese, BBQ, Fast Food          3.4 Chattanooga\n          Food Adda
        100   Desserts, Pizza, Fast Food, Mediterranean, Seafood          0.0
    Chattanooga\n
Bikanervala          65                                    Cafe, Bakery
        3.2 Chattanooga\n          Havmor Ice Cream          75
    Pizza, Bakery, Fast
Food, Chinese, Seafood          3.6 Chattanooga\n          Moti Mahal Delux
        52              Tea, Pizza, BBQ, Cafe, Mediterranean          2.7
    Chattanooga\n
Aggarwal Sweet India          30                          Tea, Desserts
        2.8 Chattanooga\n          Jack Po!tato's          63
                Tea,
Bakery, Mediterranean, Seafood          3.3 Chattanooga\n          Muradabadi
        16                    Cafe, Pizza, Bakery, BBQ          3.0
Chattanooga\n      Vishal Restaurant          27                    Tea,
    Pizza, BBQ, Seafood          3.1 Chattanooga\n          Raj Sweets
        69
Desserts, Tea, BBQ, Cafe, Mediterranean          2.8 Chattanooga\n          Punjabee's
    Darbar          41          Tea, French, Bakery, Fast Food, Chinese
        3.0

```
Chattanooga\n                   Basil Tree          20                        Bakery, Indian,
    American, Desserts        2.4 Chattanooga\n          Domino's Pizza
            77
Bakery, Pizza, American, BBQ          2.8 Chattanooga\n        Pizza Hut Delivery
            25                        Tea, Cafe, American, Indian          3.9
    Chattanooga\n
Ichiban          87                        Mexican, Bakery, Desserts
    3.9 Chattanooga\n              Habibi          95
Cafe, BBQ, Seafood              3.7 Chattanooga\n          Cook Du Kdu          69
    Pizza, Fast Food, Cafe, Indian, Mediterranean, Seafood        3.1 Chattanooga\n
Go Krazy          64                        French, BBQ, Fast Food
    3.5 Chattanooga\n              Bentoya          41                        Tea,
    Fast
Food, Bakery, Seafood          3.8 Chattanooga\n          Burger's King
    16                        Tea, Bakery, Desserts          4.1 Chattanooga
    "},
{'Description': 'Accommodations in Chattanooga', 'Content': '
                                NAME price      room type
                            house_rules minimum
nights maximum occupancy review rate number     city\n     Affordable Private
    Spacious Room in Brooklyn 790.0 Private room                        No
    parties
2.0              2              3.0 Chattanooga\n
    Sunny One Bedroom 722.0 Entire home/apt        No pets & No children under 10
            1.0
5              3.0 Chattanooga\nUpper West / Morningside Heights Apt, Near Subway
    290.0 Entire home/apt                        No visitors          3.0
                3
3.0 Chattanooga\n              Sunny room+Pvte office in huge loft 728.0 Private room
                    No parties          4.0          2
                5.0
Chattanooga\n     Extra Cozy Room in Center of Williamsburg 1033.0 Private room
                    No pets          1.0          1
                1.0
Chattanooga\n   Luxury & Charm. Steps from Christopher Park! 301.0 Entire home/apt
        No pets & No visitors & No parties          5.0          2
                4.0
Chattanooga\n                        Fort Greene Room 722.0  Private room
        No visitors & No children under 10          2.0          1
                2.0
Chattanooga\n                                                973.0
    Entire home/apt                        No pets          10.0
                6
4.0 Chattanooga\n     4 bdrm/2 bath apt. Central Pk, Columbia U. 564.0 Entire home/
    apt              No visitors & No pets          30.0          4
                3.0
Chattanooga\n     Cozy room in Bushwick- 15 min to the city 933.0 Private room
                No parties & No pets          1.0          2
                2.0
Chattanooga\n              Modern apartment w/ gorgeous view 921.0 Private room
                    No pets          3.0          2
                1.0
Chattanooga\n     Artsy Private BR in Fort Greene Cumberland 727.0 Private room
                    No visitors          30.0          1
                3.0
Chattanooga\n              Studio Deluxe 1 - Wyndham Midtown 45 848.0 Private room No
    pets & No smoking & No children under 10          3.0          1
                2.0
Chattanooga\n     Trendy and Stylish Downtown apartment !!! 110.0 Entire home/apt
                    No pets          3.0          2
                2.0
Chattanooga\n     Sonder | 180 Water | Luxurious 1BR + Rooftop 676.0 Entire home/apt
                No pets & No smoking          29.0          5
    3.0
```

Chattanooga'}, {'Description': 'Flight from Houston to Nashville on 2022-03-21', '
Content': 'Flight Number Price DepTime ArrTime ActualElapsedTime FlightDate
OriginCityName DestCityName
Distance\n   F3827247   241   14:22   16:10 1 hours 48 minutes 2022-03-21    Houston
    Nashville    657.0\n    F3827724   147   18:35   20:33 1 hours 58 minutes
    2022-03-21
Houston   Nashville    657.0\n    F3827820   157   12:07   14:02 1 hours 55 minutes
    2022-03-21    Houston    Nashville    657.0\n    F3827930   305   07:36   09:41 2
    hours 5
minutes 2022-03-21    Houston    Nashville    657.0\n    F3956407   264   11:38
    13:15 1 hours 37 minutes 2022-03-21    Houston    Nashville    670.0\n
    F3956408   246
08:08   09:47 1 hours 39 minutes 2022-03-21    Houston    Nashville    670.0\n
    F3956409   145   17:36   19:12 1 hours 36 minutes 2022-03-21    Houston
    Nashville    670.0\n
F3956532   225   07:14   08:52 1 hours 38 minutes 2022-03-21    Houston    Nashville
    657.0\n    F3956533   292   11:42   13:19 1 hours 37 minutes 2022-03-21
    Houston
Nashville    657.0\n    F3956534   248   18:20   20:06 1 hours 46 minutes 2022-03-21
    Houston    Nashville    657.0\n    F4038756   272   10:11   11:52 1 hours 41
    minutes
2022-03-21    Houston    Nashville    657.0\n    F4038966   149   16:33   18:13 1
    hours 40 minutes 2022-03-21    Houston    Nashville    657.0'}, {'Description': '
    Self-driving
from Houston to Nashville', 'Content': 'self-driving, from Houston to Nashville,
    duration: 11 hours 50 mins, distance: 1,253 km, cost: 62'}, {'Description': '
    Taxi from Houston to
Nashville', 'Content': 'taxi, from Houston to Nashville, duration: 11 hours 50 mins,
    distance: 1,253 km, cost: 1253'}, {'Description': 'Flight from Nashville to
    Knoxville on 2022-03-23',
'Content': 'There is no flight from Nashville to Knoxville on 2022-03-23.'}, {'
    Description': 'Self-driving from Nashville to Knoxville', 'Content': 'self-
    driving, from Nashville to
Knoxville, duration: 2 hours 42 mins, distance: 290 km, cost: 14'}, {'Description':
    'Taxi from Nashville to Knoxville', 'Content': 'taxi, from Nashville to
    Knoxville, duration: 2 hours 42
mins, distance: 290 km, cost: 290'}, {'Description': 'Flight from Knoxville to
    Chattanooga on 2022-03-25', 'Content': 'There is no flight from Knoxville to
    Chattanooga on 2022-03-25.'},
{'Description': 'Self-driving from Knoxville to Chattanooga', 'Content': 'self-
    driving, from Knoxville to Chattanooga, duration: 1 hour 41 mins, distance: 180
    km, cost: 9'}, {'Description':
'Taxi from Knoxville to Chattanooga', 'Content': 'taxi, from Knoxville to
    Chattanooga, duration: 1 hour 41 mins, distance: 180 km, cost: 180'}, {'
    Description': 'Flight from Chattanooga to
Houston on 2022-03-27', 'Content': 'There is no flight from Chattanooga to Houston
    on 2022-03-27.'}, {'Description': 'Self-driving from Chattanooga to Houston', '
    Content': 'self-driving,
from Chattanooga to Houston, duration: 11 hours 47 mins, distance: 1,309 km, cost:
    65'}, {'Description': 'Taxi from Chattanooga to Houston', 'Content': 'taxi,
    from Chattanooga to Houston,
duration: 11 hours 47 mins, distance: 1,309 km, cost: 1309'}]
Query: Could you design a one-week travel itinerary for two, departing from Houston
    and touring three cities in Tennessee from March 21st to March 27th, 2022? Our
    budget is now $8,200. We
require accommodations that allow smoking and should ideally be private rooms. As
    for transportation, we would prefer not to self-drive.
Travel Plan:
[
    {
        "day": 1,
        "people_number": 2,
        "current_city": "from Houston to Nashville",
        "transportation": "Flight Number: F3827820, from Houston to Nashville,
            Departure Time: 12:07, Arrival Time: 14:02",
```

```
        "breakfast": "-",
        "attraction": "Country Music Hall of Fame and Museum, Nashville",
        "lunch": "Cafe Arabelle, Nashville",
        "dinner": "Les 3 Brasseurs, Nashville",
        "accommodation": "FiDi Cozy room overlooking East River, Nashville"
    },
    {
        "day": 2,
        "people_number": 2,
        "current_city": "Nashville",
        "transportation": "-",
        "breakfast": "Burger Planet, Nashville",
        "attraction": "Johnny Cash Museum; Centennial Park, Nashville",
        "lunch": "Mamagoto, Nashville",
        "dinner": "Oh! Calcutta, Nashville",
        "accommodation": "FiDi Cozy room overlooking East River, Nashville"
    },
    {
        "day": 3,
        "people_number": 2,
        "current_city": "from Nashville to Knoxville",
        "transportation": "Taxi, from Nashville to Knoxville, Departure Time: 09:00,
            Arrival Time: 11:42, Duration: 2 hours 42 minutes, Cost: $290",
        "breakfast": "Cafe El Chico, Nashville",
        "attraction": "World's Fair Park, Knoxville",
        "lunch": "Ali Baba & 41 Dishes, Knoxville",
        "dinner": "Sky On 57, Knoxville",
        "accommodation": "Light-filled Room in Renovated Apt, Knoxville"
    },
    {
        "day": 4,
        "people_number": 2,
        "current_city": "Knoxville",
        "transportation": "-",
        "breakfast": "Chaat Corner, Knoxville",
        "attraction": "Ijams Nature Center; Sunsphere, Knoxville",
        "lunch": "Anand Sweets, Knoxville",
        "dinner": "La-Nawaab, Knoxville",
        "accommodation": "Light-filled Room in Renovated Apt, Knoxville"
    },
    {
        "day": 5,
        "people_number": 2,
        "current_city": "from Knoxville to Chattanooga",
        "transportation": "Taxi, from Knoxville to Chattanooga, Departure Time:
            09:00, Arrival Time: 10:41, Duration: 1 hour 41 minutes, Cost: $180",
        "breakfast": "Anand Sweets, Knoxville",
        "attraction": "Tennessee Aquarium, Chattanooga",
        "lunch": "Anupam Sweet, Chattanooga",
        "dinner": "Bisque Bakery, Chattanooga",
        "accommodation": "Trendy and Stylish Downtown apartment !!!, Chattanooga"
    },
    {
        "day": 6,
        "people_number": 2,
        "current_city": "Chattanooga",
        "transportation": "-",
        "breakfast": "Anupam Sweets & Restaurant, Chattanooga",
        "attraction": "Rock City Gardens; Ruby Falls, Chattanooga",
        "lunch": "Chaayos, Chattanooga",
        "dinner": "L'amandier, Chattanooga",
        "accommodation": "Trendy and Stylish Downtown apartment !!!, Chattanooga"
    },
    {
        "day": 7,
```

```
3186              "people_number": 2,
3187              "current_city": "from Chattanooga to Houston",
3188              "transportation": "Taxi, from Chattanooga to Houston, Departure Time: 08:00,
3189                  Arrival Time: 19:47, Duration: 11 hours 47 minutes, Cost: $1309",
3190              "breakfast": "Bisque Bakery, Chattanooga",
3191              "attraction": "Coolidge Park, Chattanooga",
3192              "lunch": "Nooba, Chattanooga",
3193              "dinner": "-",
3194              "accommodation": "-"
3195          }
3196      ]
```

By inspecting your plan we find the following issue. Please refine your plan
    according to the feedback below:
1. The room type should be private room :-> Trendy and Stylish Downtown apartment
    !!!
2. The restaurant in day 5 breakfast is repeated.
3. The accommodation Trendy and Stylish Downtown apartment !!!, Chattanooga do not
    obey the minumum nights rule.
4. The lunch in day 1 is invalid or not in the data provided.

Travel Plan (please only output the JSON string without explanatory information):

==============RESPONSE OF o1-preview=================

```
[
    {
        "day": 1,
        "people_number": 2,
        "current_city": "from Houston to Nashville",
        "transportation": "Flight Number: F3827820, from Houston to Nashville,
            Departure Time: 12:07, Arrival Time: 14:02",
        "breakfast": "-",
        "attraction": "Country Music Hall of Fame and Museum, Nashville",
        "lunch": "Smoke House Deli, Nashville",
        "dinner": "Les 3 Brasseurs, Nashville",
        "accommodation": "FiDi Cozy room overlooking East River, Nashville"
    },
    {
        "day": 2,
        "people_number": 2,
        "current_city": "Nashville",
        "transportation": "-",
        "breakfast": "Burger Planet, Nashville",
        "attraction": "Johnny Cash Museum; Centennial Park, Nashville",
        "lunch": "Mamagoto, Nashville",
        "dinner": "Oh! Calcutta, Nashville",
        "accommodation": "FiDi Cozy room overlooking East River, Nashville"
    },
    {
        "day": 3,
        "people_number": 2,
        "current_city": "from Nashville to Knoxville",
        "transportation": "Taxi, from Nashville to Knoxville, Departure Time: 09:00,
            Arrival Time: 11:42, Duration: 2 hours 42 minutes, Cost: $290",
        "breakfast": "Cafe El Chico, Nashville",
        "attraction": "World's Fair Park, Knoxville",
        "lunch": "Ali Baba & 41 Dishes, Knoxville",
        "dinner": "Sky On 57, Knoxville",
        "accommodation": "Light-filled Room in Renovated Apt, Knoxville"
    },
    {
        "day": 4,
        "people_number": 2,
        "current_city": "Knoxville",
```

```
3240            "transportation": "-",
3241            "breakfast": "Chaat Corner, Knoxville",
3242            "attraction": "Ijams Nature Center; Sunsphere, Knoxville",
3243            "lunch": "Anand Sweets, Knoxville",
3244            "dinner": "La-Nawaab, Knoxville",
3245            "accommodation": "Light-filled Room in Renovated Apt, Knoxville"
3246        },
             {
3247            "day": 5,
3248            "people_number": 2,
3249            "current_city": "from Knoxville to Chattanooga",
3250            "transportation": "Taxi, from Knoxville to Chattanooga, Departure Time:
                   09:00, Arrival Time: 10:41, Duration: 1 hour 41 minutes, Cost: $180",
3251            "breakfast": "Burger Planet, Knoxville",
3252            "attraction": "Tennessee Aquarium, Chattanooga",
3253            "lunch": "Anupam Sweet, Chattanooga",
3254            "dinner": "Bisque Bakery, Chattanooga",
3255            "accommodation": "Affordable Private Spacious Room in Brooklyn, Chattanooga"
3256        },
             {
3257            "day": 6,
3258            "people_number": 2,
3259            "current_city": "Chattanooga",
3260            "transportation": "-",
3261            "breakfast": "Anupam Sweets & Restaurant, Chattanooga",
3262            "attraction": "Rock City Gardens; Ruby Falls, Chattanooga",
3263            "lunch": "Chaayos, Chattanooga",
3264            "dinner": "L'amandier, Chattanooga",
               "accommodation": "Affordable Private Spacious Room in Brooklyn, Chattanooga"
3265        },
             {
3266            "day": 7,
3267            "people_number": 2,
3268            "current_city": "from Chattanooga to Houston",
3269            "transportation": "Taxi, from Chattanooga to Houston, Departure Time: 08:00,
                   Arrival Time: 19:47, Duration: 11 hours 47 minutes, Cost: $1309",
3270            "breakfast": "Bisque Bakery, Chattanooga",
3271            "attraction": "Coolidge Park, Chattanooga",
3272            "lunch": "Nooba, Chattanooga",
3273            "dinner": "-",
3274            "accommodation": "-"
3275        }
3276    ]
```

# P    Prompts for Trip Planning

## P.1    First Iteration

You are an expert at planning trips. You are given a few constraints regarding the
    cities to visit and the durations of staying at each city. You are also given
    the flight information between the cities('and' suggests flights are possible
    both ways and 'from - to -' suggests
one way flights). You can travel to any city only once.
Produce only one plan, strictly adhering to the format shown in the examples below,
    without any extra information. Your answer shold start with 'SOLUTION:'.
Here are a few example tasks and solutions:

TASK: You plan to visit 10 European cities for 21 days in total. You only take
    direct flights to commute between cities. You plan to stay in Vienna for 3 days.
     You want to spend 5 days in Frankfurt. You want to spend 2 days in Oslo. You
    are going to attend a wedding in Oslo
between day 20 and day 21. You want to spend 3 days in Prague. You would like to
    visit Valencia for 2 days. You want to meet a friend in Valencia between day 17

and day 18. You plan to stay in Dubrovnik for 2 days. You would like to visit Edinburgh for 5 days. From day 1 to
day 5, there is a annual show you want to attend in Edinburgh. You plan to stay in London for 2 days. You plan to visit relatives in London between day 12 and day 13. You would like to visit Munich for 3 days. You would like to meet your friends at Munich between day 18 and
day 20 to tour together. You would like to visit Budapest for 3 days.

Here are the cities that have direct flights:
Valencia and Munich, Vienna and Munich, Vienna and Valencia, London and Budapest, London and Oslo, Edinburgh and Budapest, Frankfurt and Budapest, Frankfurt and London, Prague and Oslo, Edinburgh and Oslo, Edinburgh and Munich, Prague and Munich, London and Prague, Edinburgh
and London, Edinburgh and Frankfurt, Dubrovnik and Munich, Dubrovnik and Vienna, Munich and Oslo, Dubrovnik and Oslo, Budapest and Munich, Frankfurt and Prague, Vienna and London, Frankfurt and Vienna, Frankfurt and Oslo, Frankfurt and Munich, Vienna and Oslo, Vienna and
Prague, Budapest and Oslo, Budapest and Prague, London and Valencia, London and Munich, Frankfurt and Dubrovnik, Prague and Valencia, Frankfurt and Valencia, Edinburgh and Prague, Vienna and Budapest.

Find a trip plan of visiting the cities for 21 days by taking direct flights to commute between them.
SOLUTION: Here is the trip plan for visiting the 10 European cities for 21 days:

**Day 1-5:** Arriving in Edinburgh and visit Edinburgh for 5 days.
**Day 5:** Fly from Edinburgh to Frankfurt.
**Day 5-9:** Visit Frankfurt for 5 days.
**Day 9:** Fly from Frankfurt to Dubrovnik.
**Day 9-10:** Visit Dubrovnik for 2 days.
**Day 10:** Fly from Dubrovnik to Vienna.
**Day 10-12:** Visit Vienna for 3 days.
**Day 12:** Fly from Vienna to London.
**Day 12-13:** Visit London for 2 days.
**Day 13:** Fly from London to Budapest.
**Day 13-15:** Visit Budapest for 3 days.
**Day 15:** Fly from Budapest to Prague.
**Day 15-17:** Visit Prague for 3 days.
**Day 17:** Fly from Prague to Valencia.
**Day 17-18:** Visit Valencia for 2 days.
**Day 18:** Fly from Valencia to Munich.
**Day 18-20:** Visit Munich for 3 days.
**Day 20:** Fly from Munich to Oslo.
**Day 20-21:** Visit Oslo for 2 days.

TASK: You plan to visit 10 European cities for 22 days in total. You only take direct flights to commute between cities. You want to spend 4 days in Helsinki. You plan to stay in Vilnius for 2 days. You plan to stay in Istanbul for 5 days. From day 17 to day 21, there is a
annual show you want to attend in Istanbul. You want to spend 2 days in Hamburg. You plan to stay in Porto for 2 days. During day 11 and day 12, you have to attend a conference in Porto. You want to spend 4 days in Athens. You would like to visit Prague for 2 days. You plan
to visit relatives in Prague between day 13 and day 14. You want to spend 2 days in Frankfurt. You want to spend 3 days in Krakow. You want to spend 5 days in Munich. You would like to meet your friends at Munich between day 1 and day 5 to tour together.

Here are the cities that have direct flights:
from Krakow to Vilnius, Helsinki and Hamburg, Hamburg and Athens, Munich and Frankfurt, Hamburg and Porto, Munich and Istanbul, Prague and Athens, Frankfurt and Athens, Munich and Athens, Munich and Prague, from Vilnius to Munich, Hamburg and Istanbul, Frankfurt and
Istanbul, Munich and Krakow, Munich and Hamburg, Munich and Helsinki, Prague and Istanbul, Frankfurt and Vilnius, Helsinki and Istanbul, Athens and Vilnius,

Krakow and Frankfurt, Helsinki and Frankfurt, Porto and Frankfurt, Frankfurt
    and Prague, Istanbul and Vilnius, Krakow
and Istanbul, Krakow and Prague, Munich and Porto, Helsinki and Vilnius, Helsinki
    and Prague, Porto and Istanbul, Hamburg and Frankfurt, Krakow and Helsinki,
    Athens and Istanbul.

Find a trip plan of visiting the cities for 22 days by taking direct flights to
    commute between them.
SOLUTION: Here is the trip plan for visiting the 10 European cities for 22 days:

**Day 1-5:** Arriving in Munich and visit Munich for 5 days.
**Day 5:** Fly from Munich to Krakow.
**Day 5-7:** Visit Krakow for 3 days.
**Day 7:** Fly from Krakow to Helsinki.
**Day 7-10:** Visit Helsinki for 4 days.
**Day 10:** Fly from Helsinki to Hamburg.
**Day 10-11:** Visit Hamburg for 2 days.
**Day 11:** Fly from Hamburg to Porto.
**Day 11-12:** Visit Porto for 2 days.
**Day 12:** Fly from Porto to Frankfurt.
**Day 12-13:** Visit Frankfurt for 2 days.
**Day 13:** Fly from Frankfurt to Prague.
**Day 13-14:** Visit Prague for 2 days.
**Day 14:** Fly from Prague to Athens.
**Day 14-17:** Visit Athens for 4 days.
**Day 17:** Fly from Athens to Istanbul.
**Day 17-21:** Visit Istanbul for 5 days.
**Day 21:** Fly from Istanbul to Vilnius.
**Day 21-22:** Visit Vilnius for 2 days.

TASK: You plan to visit 10 European cities for 23 days in total. You only take
    direct flights to commute between cities. You would like to visit Stuttgart for
     2 days. You would like to visit Split for 2 days. You are going to attend a
    wedding in Split between day 22 and day
23. You would like to visit Vienna for 5 days. You want to spend 4 days in Madrid.
    You plan to stay in Athens for 2 days. You would like to visit London for 3
    days. During day 8 and day 10, you have to attend a conference in London. You
    plan to stay in Paris for 3 days. You
want to meet a friend in Paris between day 10 and day 12. You plan to stay in
    Reykjavik for 2 days. You have to attend a workshop in Reykjavik between day 16
     and day 17. You want to spend 4 days in Seville. You want to spend 5 days in
    Milan. You would like to meet your
friends at Milan between day 17 and day 21 to tour together.

Here are the cities that have direct flights:
Athens and Paris, Athens and Vienna, Madrid and Vienna, Madrid and Split, Vienna and
     Stuttgart, Paris and Milan, London and Vienna, London and Milan, Paris and
    Reykjavik, Athens and London, from Reykjavik to Stuttgart, Seville and Milan,
    from Reykjavik to Madrid, London and
Stuttgart, Milan and Stuttgart, Vienna and Reykjavik, Athens and Split, Athens and
    Milan, Madrid and Athens, Madrid and London, Paris and Split, London and Paris,
     Seville and Vienna, Vienna and Milan, Athens and Stuttgart, Madrid and Paris,
    Seville and Madrid, from Reykjavik
to Athens, Vienna and Split, London and Split, Stuttgart and Split, Seville and
    Paris, Paris and Stuttgart, Reykjavik and Milan, London and Reykjavik, Madrid
    and Milan, Paris and Vienna, Milan and Split.

Find a trip plan of visiting the cities for 23 days by taking direct flights to
    commute between them.
SOLUTION: Here is the trip plan for visiting the 10 European cities for 23 days:

**Day 1-4:** Arriving in Seville and visit Seville for 4 days.
**Day 4:** Fly from Seville to Madrid.
**Day 4-7:** Visit Madrid for 4 days.
**Day 7:** Fly from Madrid to Athens.

```
3402    **Day 7-8:** Visit Athens for 2 days.
3403    **Day 8:** Fly from Athens to London.
3404    **Day 8-10:** Visit London for 3 days.
3405    **Day 10:** Fly from London to Paris.
3406    **Day 10-12:** Visit Paris for 3 days.
3407    **Day 12:** Fly from Paris to Vienna.
3408    **Day 12-16:** Visit Vienna for 5 days.
3409    **Day 16:** Fly from Vienna to Reykjavik.
3410    **Day 16-17:** Visit Reykjavik for 2 days.
3411    **Day 17:** Fly from Reykjavik to Milan.
3412    **Day 17-21:** Visit Milan for 5 days.
3413    **Day 21:** Fly from Milan to Stuttgart.
3414    **Day 21-22:** Visit Stuttgart for 2 days.
3415    **Day 22:** Fly from Stuttgart to Split.
3416    **Day 22-23:** Visit Split for 2 days.
```

TASK: You plan to visit 10 European cities for 25 days in total. You only take direct flights to commute between cities. You would like to visit Berlin for 2 days. You would like to visit Riga for 2 days. During day 5 and day 6, you have to attend a conference in Riga. You want to spend 3 days in Barcelona. You would like to visit Lyon for 4 days. You would like to meet your friends at Lyon between day 8 and day 11 to tour together. You plan to stay in Naples for 2 days. You plan to stay in Venice for 5 days. You want to spend 5 days in Helsinki. You plan to visit relatives in Helsinki between day 21 and day 25. You plan to stay in Rome for 5 days. You want to spend 3 days in Vilnius. You want to spend 3 days in Amsterdam. You are going to attend a wedding in Amsterdam between day 19 and day 21.

Here are the cities that have direct flights:
Berlin and Amsterdam, Rome and Helsinki, Rome and Lyon, Naples and Amsterdam, Riga and Barcelona, Rome and Venice, Riga and Amsterdam, from Riga to Vilnius, Barcelona and Berlin, Rome and Barcelona, Rome and Amsterdam, Barcelona and Venice, Berlin and Helsinki, Amsterdam and Helsinki, Vilnius and Helsinki, Rome and Berlin, from Rome to Riga, Barcelona and Amsterdam, Venice and Naples, Barcelona and Lyon, Naples and Berlin, Barcelona and Helsinki, Venice and Helsinki, Barcelona and Naples, Vilnius and Amsterdam, Venice and Amsterdam, Lyon and Venice, Naples and Helsinki, Riga and Berlin, Rome and Naples, Venice and Berlin, Riga and Helsinki, Berlin and Vilnius, Lyon and Amsterdam.

Find a trip plan of visiting the cities for 25 days by taking direct flights to commute between them.
SOLUTION: Here is the trip plan for visiting the 10 European cities for 25 days:

**Day 1-5:** Arriving in Rome and visit Rome for 5 days.
**Day 5:** Fly from Rome to Riga.
**Day 5-6:** Visit Riga for 2 days.
**Day 6:** Fly from Riga to Barcelona.
**Day 6-8:** Visit Barcelona for 3 days.
**Day 8:** Fly from Barcelona to Lyon.
**Day 8-11:** Visit Lyon for 4 days.
**Day 11:** Fly from Lyon to Venice.
**Day 11-15:** Visit Venice for 5 days.
**Day 15:** Fly from Venice to Naples.
**Day 15-16:** Visit Naples for 2 days.
**Day 16:** Fly from Naples to Berlin.
**Day 16-17:** Visit Berlin for 2 days.
**Day 17:** Fly from Berlin to Vilnius.
**Day 17-19:** Visit Vilnius for 3 days.
**Day 19:** Fly from Vilnius to Amsterdam.
**Day 19-21:** Visit Amsterdam for 3 days.
**Day 21:** Fly from Amsterdam to Helsinki.
**Day 21-25:** Visit Helsinki for 5 days.

TASK: You plan to visit 10 European cities for 27 days in total. You only take
    direct flights to commute between cities. You would like to visit Prague for 5
    days. You have to attend a workshop in Prague between day 7 and day 11. You
    would like to visit Helsinki for 3 days.
You are going to attend a wedding in Helsinki between day 15 and day 17. You plan to
    stay in Tallinn for 2 days. You want to meet a friend in Tallinn between day 6
    and day 7. You want to spend 4 days in Edinburgh. You want to spend 5 days in
    Paris. You want to spend 4 days
in Vienna. You plan to stay in Lisbon for 5 days. From day 11 to day 15, there is a
    annual show you want to attend in Lisbon. You want to spend 4 days in Budapest.
    You plan to stay in Lyon for 2 days. You plan to stay in Brussels for 2 days.
    You would like to meet your
friends at Brussels between day 1 and day 2 to tour together.

Here are the cities that have direct flights:
Prague and Lyon, Brussels and Lisbon, Helsinki and Budapest, Vienna and Lyon, Paris
    and Tallinn, Brussels and Prague, Brussels and Helsinki, Prague and Helsinki,
    Brussels and Vienna, Brussels and Budapest, Lisbon and Budapest, Tallinn and
    Helsinki, Brussels and Paris,
Brussels and Tallinn, Lisbon and Lyon, Prague and Lisbon, Paris and Prague, Helsinki
    and Edinburgh, Prague and Edinburgh, Tallinn and Prague, Brussels and Lyon,
    Paris and Lisbon, Helsinki and Vienna, Paris and Helsinki, Paris and Budapest,
    Edinburgh and Budapest, Brussels
and Edinburgh, Lisbon and Vienna, Paris and Lyon, Lisbon and Helsinki, Prague and
    Vienna, Paris and Vienna, Prague and Budapest, Paris and Edinburgh, Budapest
    and Vienna.

Find a trip plan of visiting the cities for 27 days by taking direct flights to
    commute between them.
SOLUTION: Here is the trip plan for visiting the 10 European cities for 27 days:

**Day 1-2:** Arriving in Brussels and visit Brussels for 2 days.
**Day 2:** Fly from Brussels to Paris.
**Day 2-6:** Visit Paris for 5 days.
**Day 6:** Fly from Paris to Tallinn.
**Day 6-7:** Visit Tallinn for 2 days.
**Day 7:** Fly from Tallinn to Prague.
**Day 7-11:** Visit Prague for 5 days.
**Day 11:** Fly from Prague to Lisbon.
**Day 11-15:** Visit Lisbon for 5 days.
**Day 15:** Fly from Lisbon to Helsinki.
**Day 15-17:** Visit Helsinki for 3 days.
**Day 17:** Fly from Helsinki to Edinburgh.
**Day 17-20:** Visit Edinburgh for 4 days.
**Day 20:** Fly from Edinburgh to Budapest.
**Day 20-23:** Visit Budapest for 4 days.
**Day 23:** Fly from Budapest to Vienna.
**Day 23-26:** Visit Vienna for 4 days.
**Day 26:** Fly from Vienna to Lyon.
**Day 26-27:** Visit Lyon for 2 days.

Query:
You plan to visit 10 European cities for 25 days in total. You only take direct
    flights to commute between cities. You plan to stay in Berlin for 4 days. You
    have to attend a workshop in Berlin between day 12 and day 15. You would like
    to visit Prague for 2 days. You plan to
stay in Stuttgart for 5 days. You would like to visit Manchester for 3 days. You
    want to spend 2 days in Nice. You would like to visit Reykjavik for 2 days. You
    would like to visit Florence for 3 days. You plan to stay in Vilnius for 5
    days. You would like to meet your
friends at Vilnius between day 15 and day 19 to tour together. You plan to stay in
    Oslo for 4 days. You would like to visit Dubrovnik for 4 days. You plan to
    visit relatives in Dubrovnik between day 1 and day 4.

Here are the cities that have direct flights:
from Reykjavik to Stuttgart, Manchester and Stuttgart, Nice and Berlin, Oslo and
    Prague, Stuttgart and Berlin, Manchester and Nice, Reykjavik and Oslo,
    Reykjavik and Prague, Manchester and Prague, Reykjavik and Berlin, Dubrovnik
    and Manchester, Manchester and Oslo,
Manchester and Berlin, Prague and Florence, Berlin and Vilnius, Dubrovnik and Oslo,
    Nice and Oslo, Berlin and Oslo, Nice and Reykjavik, Vilnius and Oslo.

Find a trip plan of visiting the cities for 25 days by taking direct flights to
    commute between them.

==============RESPONSE OF o1-preview=================
SOLUTION: Here is the trip plan for visiting the 10 European cities for 25 days:

**Day 1-4:** Arriving in Dubrovnik and visit Dubrovnik for 4 days.

**Day 4:** Fly from Dubrovnik to Manchester.

**Day 4-7:** Visit Manchester for 3 days.

**Day 7:** Fly from Manchester to Stuttgart.

**Day 7-12:** Visit Stuttgart for 5 days.

**Day 12:** Fly from Stuttgart to Berlin.

**Day 12-15:** Visit Berlin for 4 days. (Attend workshop between day 12 and day 15)

**Day 15:** Fly from Berlin to Vilnius.

**Day 15-19:** Visit Vilnius for 5 days. (Meet friends between day 15 and day 19)

**Day 19:** Fly from Vilnius to Oslo via Berlin.

**Day 19-23:** Visit Oslo for 4 days.

**Day 23:** Fly from Oslo to Reykjavik.

**Day 23-25:** Visit Reykjavik for 2 days.

==============FEEDBACK FROM METACONTROLLER=================
Number of cities in plan is 7, expected 10

P.2   Back Prompt

You are an expert at planning trips. Fix the below given trip schedule such that it
    meets as many specified constraints as possible. You are given a few
    constraints regarding the cities to visit and the durations of staying at each
    city. You are also given the flight
information between the cities.
Produce only one plan, strictly adhering to the format shown in the examples below,
    without any extra information.

The produced plan will be parsed using the following python code:

```python
def parse_response(response: str):
  """Parse the response.

  Returns a parsed plan in a list of (city, stay_days) tuples.

  Args:
    response: Raw response from the model.
```

```
   Returns:
     Structured plan after parsing.
   """
   pattern_visit = r'\d+-\d+'
   pattern_flight = r'.*Day (\d+).*from (\w+) to (\w+)'
   pattern_days = r'European cities for (\d+) days'

   days, flights, flight_days = [], [], []
   total_days = None
   for piece in response.split('\n'):
     days_match = re.findall(pattern_days, piece)
     if days_match:
       total_days = int(days_match[0])

     visit_match = re.findall(pattern_visit, piece)
     if visit_match:
       days.append(visit_match[0])
       end_day = int(visit_match[0].split('-')[1])
       # Reach the end of the plan, stop to avoid parsing alternative plans.
       if end_day == total_days:
         break
     flight_match = re.findall(pattern_flight, piece)
     if flight_match:
       flights.append(flight_match[0])

   visit_cities, parsed_plan = [], []
   for flight_day, begin_city, end_city in flights:
     flight_days.append(int(flight_day))
     if not visit_cities:
       visit_cities.append(begin_city)
       visit_cities.append(end_city)
     else:
       visit_cities.append(end_city)

   if not days or not flights or not visit_cities:
     return []
   last_day = int(days[-1].split('-')[1])
   flight_days = [1] + flight_days +
   for i, visit_city in enumerate(visit_cities):
     city_stay = flight_days - flight_days + 1
     parsed_plan.append((visit_city, city_stay))

   return parsed_plan
'''

Here are a few example tasks and solutions:

TASK: You plan to visit 10 European cities for 21 days in total. You only take
    direct flights to commute between cities. You plan to stay in Vienna for 3 days.
     You want to spend 5 days in Frankfurt. You want to spend 2 days in Oslo. You
    are going to attend a wedding in Oslo
between day 20 and day 21. You want to spend 3 days in Prague. You would like to
    visit Valencia for 2 days. You want to meet a friend in Valencia between day 17
     and day 18. You plan to stay in Dubrovnik for 2 days. You would like to visit
    Edinburgh for 5 days. From day 1 to
day 5, there is a annual show you want to attend in Edinburgh. You plan to stay in
    London for 2 days. You plan to visit relatives in London between day 12 and day
     13. You would like to visit Munich for 3 days. You would like to meet your
    friends at Munich between day 18 and
day 20 to tour together. You would like to visit Budapest for 3 days.

Here are the cities that have direct flights:
Valencia and Munich, Vienna and Munich, Vienna and Valencia, London and Budapest,
    London and Oslo, Edinburgh and Budapest, Frankfurt and Budapest, Frankfurt and
```

London, Prague and Oslo, Edinburgh and Oslo, Edinburgh and Munich, Prague and Munich, London and Prague, Edinburgh
and London, Edinburgh and Frankfurt, Dubrovnik and Munich, Dubrovnik and Vienna, Munich and Oslo, Dubrovnik and Oslo, Budapest and Munich, Frankfurt and Prague, Vienna and London, Frankfurt and Vienna, Frankfurt and Oslo, Frankfurt and Munich, Vienna and Oslo, Vienna and
Prague, Budapest and Oslo, Budapest and Prague, London and Valencia, London and Munich, Frankfurt and Dubrovnik, Prague and Valencia, Frankfurt and Valencia, Edinburgh and Prague, Vienna and Budapest.

Find a trip plan of visiting the cities for 21 days by taking direct flights to commute between them.
SOLUTION: Here is the trip plan for visiting the 10 European cities for 21 days:

**Day 1-5:** Arriving in Edinburgh and visit Edinburgh for 5 days.
**Day 5:** Fly from Edinburgh to Frankfurt.
**Day 5-9:** Visit Frankfurt for 5 days.
**Day 9:** Fly from Frankfurt to Dubrovnik.
**Day 9-10:** Visit Dubrovnik for 2 days.
**Day 10:** Fly from Dubrovnik to Vienna.
**Day 10-12:** Visit Vienna for 3 days.
**Day 12:** Fly from Vienna to London.
**Day 12-13:** Visit London for 2 days.
**Day 13:** Fly from London to Budapest.
**Day 13-15:** Visit Budapest for 3 days.
**Day 15:** Fly from Budapest to Prague.
**Day 15-17:** Visit Prague for 3 days.
**Day 17:** Fly from Prague to Valencia.
**Day 17-18:** Visit Valencia for 2 days.
**Day 18:** Fly from Valencia to Munich.
**Day 18-20:** Visit Munich for 3 days.
**Day 20:** Fly from Munich to Oslo.
**Day 20-21:** Visit Oslo for 2 days.

TASK: You plan to visit 10 European cities for 22 days in total. You only take direct flights to commute between cities. You want to spend 4 days in Helsinki. You plan to stay in Vilnius for 2 days. You plan to stay in Istanbul for 5 days. From day 17 to day 21, there is a
annual show you want to attend in Istanbul. You want to spend 2 days in Hamburg. You plan to stay in Porto for 2 days. During day 11 and day 12, you have to attend a conference in Porto. You want to spend 4 days in Athens. You would like to visit Prague for 2 days. You plan
to visit relatives in Prague between day 13 and day 14. You want to spend 2 days in Frankfurt. You want to spend 3 days in Krakow. You want to spend 5 days in Munich. You would like to meet your friends at Munich between day 1 and day 5 to tour together.

Here are the cities that have direct flights:
from Krakow to Vilnius, Helsinki and Hamburg, Hamburg and Athens, Munich and Frankfurt, Hamburg and Porto, Munich and Istanbul, Prague and Athens, Frankfurt and Athens, Munich and Athens, Munich and Prague, from Vilnius to Munich, Hamburg and Istanbul, Frankfurt and
Istanbul, Munich and Krakow, Munich and Hamburg, Munich and Helsinki, Prague and Istanbul, Frankfurt and Vilnius, Helsinki and Istanbul, Athens and Vilnius, Krakow and Frankfurt, Helsinki and Frankfurt, Porto and Frankfurt, Frankfurt and Prague, Istanbul and Vilnius, Krakow
and Istanbul, Krakow and Prague, Munich and Porto, Helsinki and Vilnius, Helsinki and Prague, Porto and Istanbul, Hamburg and Frankfurt, Krakow and Helsinki, Athens and Istanbul.

Find a trip plan of visiting the cities for 22 days by taking direct flights to commute between them.
SOLUTION: Here is the trip plan for visiting the 10 European cities for 22 days:

**Day 1-5:** Arriving in Munich and visit Munich for 5 days.

**Day 5:** Fly from Munich to Krakow.
**Day 5-7:** Visit Krakow for 3 days.
**Day 7:** Fly from Krakow to Helsinki.
**Day 7-10:** Visit Helsinki for 4 days.
**Day 10:** Fly from Helsinki to Hamburg.
**Day 10-11:** Visit Hamburg for 2 days.
**Day 11:** Fly from Hamburg to Porto.
**Day 11-12:** Visit Porto for 2 days.
**Day 12:** Fly from Porto to Frankfurt.
**Day 12-13:** Visit Frankfurt for 2 days.
**Day 13:** Fly from Frankfurt to Prague.
**Day 13-14:** Visit Prague for 2 days.
**Day 14:** Fly from Prague to Athens.
**Day 14-17:** Visit Athens for 4 days.
**Day 17:** Fly from Athens to Istanbul.
**Day 17-21:** Visit Istanbul for 5 days.
**Day 21:** Fly from Istanbul to Vilnius.
**Day 21-22:** Visit Vilnius for 2 days.

TASK: You plan to visit 10 European cities for 23 days in total. You only take
    direct flights to commute between cities. You would like to visit Stuttgart for
     2 days. You would like to visit Split for 2 days. You are going to attend a
    wedding in Split between day 22 and day
23. You would like to visit Vienna for 5 days. You want to spend 4 days in Madrid.
    You plan to stay in Athens for 2 days. You would like to visit London for 3
    days. During day 8 and day 10, you have to attend a conference in London. You
    plan to stay in Paris for 3 days. You
want to meet a friend in Paris between day 10 and day 12. You plan to stay in
    Reykjavik for 2 days. You have to attend a workshop in Reykjavik between day 16
     and day 17. You want to spend 4 days in Seville. You want to spend 5 days in
    Milan. You would like to meet your
friends at Milan between day 17 and day 21 to tour together.

Here are the cities that have direct flights:
Athens and Paris, Athens and Vienna, Madrid and Vienna, Madrid and Split, Vienna and
     Stuttgart, Paris and Milan, London and Vienna, London and Milan, Paris and
    Reykjavik, Athens and London, from Reykjavik to Stuttgart, Seville and Milan,
    from Reykjavik to Madrid, London and
Stuttgart, Milan and Stuttgart, Vienna and Reykjavik, Athens and Split, Athens and
    Milan, Madrid and Athens, Madrid and London, Paris and Split, London and Paris,
     Seville and Vienna, Vienna and Milan, Athens and Stuttgart, Madrid and Paris,
    Seville and Madrid, from Reykjavik
to Athens, Vienna and Split, London and Split, Stuttgart and Split, Seville and
    Paris, Paris and Stuttgart, Reykjavik and Milan, London and Reykjavik, Madrid
    and Milan, Paris and Vienna, Milan and Split.

Find a trip plan of visiting the cities for 23 days by taking direct flights to
    commute between them.
SOLUTION: Here is the trip plan for visiting the 10 European cities for 23 days:

**Day 1-4:** Arriving in Seville and visit Seville for 4 days.
**Day 4:** Fly from Seville to Madrid.
**Day 4-7:** Visit Madrid for 4 days.
**Day 7:** Fly from Madrid to Athens.
**Day 7-8:** Visit Athens for 2 days.
**Day 8:** Fly from Athens to London.
**Day 8-10:** Visit London for 3 days.
**Day 10:** Fly from London to Paris.
**Day 10-12:** Visit Paris for 3 days.
**Day 12:** Fly from Paris to Vienna.
**Day 12-16:** Visit Vienna for 5 days.
**Day 16:** Fly from Vienna to Reykjavik.
**Day 16-17:** Visit Reykjavik for 2 days.
**Day 17:** Fly from Reykjavik to Milan.
**Day 17-21:** Visit Milan for 5 days.

**Day 21:** Fly from Milan to Stuttgart.
**Day 21-22:** Visit Stuttgart for 2 days.
**Day 22:** Fly from Stuttgart to Split.
**Day 22-23:** Visit Split for 2 days.

TASK: You plan to visit 10 European cities for 25 days in total. You only take direct flights to commute between cities. You would like to visit Berlin for 2 days. You would like to visit Riga for 2 days. During day 5 and day 6, you have to attend a conference in Riga. You
want to spend 3 days in Barcelona. You would like to visit Lyon for 4 days. You would like to meet your friends at Lyon between day 8 and day 11 to tour together. You plan to stay in Naples for 2 days. You plan to stay in Venice for 5 days. You want to spend 5 days in
Helsinki. You plan to visit relatives in Helsinki between day 21 and day 25. You plan to stay in Rome for 5 days. You want to spend 3 days in Vilnius. You want to spend 3 days in Amsterdam. You are going to attend a wedding in Amsterdam between day 19 and day 21.

Here are the cities that have direct flights:
Berlin and Amsterdam, Rome and Helsinki, Rome and Lyon, Naples and Amsterdam, Riga and Barcelona, Rome and Venice, Riga and Amsterdam, from Riga to Vilnius, Barcelona and Berlin, Rome and Barcelona, Rome and Amsterdam, Barcelona and Venice, Berlin and Helsinki, Amsterdam and
Helsinki, Vilnius and Helsinki, Rome and Berlin, from Rome to Riga, Barcelona and Amsterdam, Venice and Naples, Barcelona and Lyon, Naples and Berlin, Barcelona and Helsinki, Venice and Helsinki, Barcelona and Naples, Vilnius and Amsterdam, Venice and Amsterdam, Lyon and
Venice, Naples and Helsinki, Riga and Berlin, Rome and Naples, Venice and Berlin, Riga and Helsinki, Berlin and Vilnius, Lyon and Amsterdam.

Find a trip plan of visiting the cities for 25 days by taking direct flights to commute between them.
SOLUTION: Here is the trip plan for visiting the 10 European cities for 25 days:

**Day 1-5:** Arriving in Rome and visit Rome for 5 days.
**Day 5:** Fly from Rome to Riga.
**Day 5-6:** Visit Riga for 2 days.
**Day 6:** Fly from Riga to Barcelona.
**Day 6-8:** Visit Barcelona for 3 days.
**Day 8:** Fly from Barcelona to Lyon.
**Day 8-11:** Visit Lyon for 4 days.
**Day 11:** Fly from Lyon to Venice.
**Day 11-15:** Visit Venice for 5 days.
**Day 15:** Fly from Venice to Naples.
**Day 15-16:** Visit Naples for 2 days.
**Day 16:** Fly from Naples to Berlin.
**Day 16-17:** Visit Berlin for 2 days.
**Day 17:** Fly from Berlin to Vilnius.
**Day 17-19:** Visit Vilnius for 3 days.
**Day 19:** Fly from Vilnius to Amsterdam.
**Day 19-21:** Visit Amsterdam for 3 days.
**Day 21:** Fly from Amsterdam to Helsinki.
**Day 21-25:** Visit Helsinki for 5 days.

TASK: You plan to visit 10 European cities for 27 days in total. You only take direct flights to commute between cities. You would like to visit Prague for 5 days. You have to attend a workshop in Prague between day 7 and day 11. You would like to visit Helsinki for 3 days.
You are going to attend a wedding in Helsinki between day 15 and day 17. You plan to stay in Tallinn for 2 days. You want to meet a friend in Tallinn between day 6 and day 7. You want to spend 4 days in Edinburgh. You want to spend 5 days in Paris. You want to spend 4 days
in Vienna. You plan to stay in Lisbon for 5 days. From day 11 to day 15, there is a annual show you want to attend in Lisbon. You want to spend 4 days in Budapest.

You plan to stay in Lyon for 2 days. You plan to stay in Brussels for 2 days. You would like to meet your
friends at Brussels between day 1 and day 2 to tour together.

Here are the cities that have direct flights:
Prague and Lyon, Brussels and Lisbon, Helsinki and Budapest, Vienna and Lyon, Paris and Tallinn, Brussels and Prague, Brussels and Helsinki, Prague and Helsinki, Brussels and Vienna, Brussels and Budapest, Lisbon and Budapest, Tallinn and Helsinki, Brussels and Paris,
Brussels and Tallinn, Lisbon and Lyon, Prague and Lisbon, Paris and Prague, Helsinki and Edinburgh, Prague and Edinburgh, Tallinn and Prague, Brussels and Lyon, Paris and Lisbon, Helsinki and Vienna, Paris and Helsinki, Paris and Budapest, Edinburgh and Budapest, Brussels
and Edinburgh, Lisbon and Vienna, Paris and Lyon, Lisbon and Helsinki, Prague and Vienna, Paris and Vienna, Prague and Budapest, Paris and Edinburgh, Budapest and Vienna.

Find a trip plan of visiting the cities for 27 days by taking direct flights to commute between them.
SOLUTION: Here is the trip plan for visiting the 10 European cities for 27 days:

**Day 1-2:** Arriving in Brussels and visit Brussels for 2 days.
**Day 2:** Fly from Brussels to Paris.
**Day 2-6:** Visit Paris for 5 days.
**Day 6:** Fly from Paris to Tallinn.
**Day 6-7:** Visit Tallinn for 2 days.
**Day 7:** Fly from Tallinn to Prague.
**Day 7-11:** Visit Prague for 5 days.
**Day 11:** Fly from Prague to Lisbon.
**Day 11-15:** Visit Lisbon for 5 days.
**Day 15:** Fly from Lisbon to Helsinki.
**Day 15-17:** Visit Helsinki for 3 days.
**Day 17:** Fly from Helsinki to Edinburgh.
**Day 17-20:** Visit Edinburgh for 4 days.
**Day 20:** Fly from Edinburgh to Budapest.
**Day 20-23:** Visit Budapest for 4 days.
**Day 23:** Fly from Budapest to Vienna.
**Day 23-26:** Visit Vienna for 4 days.
**Day 26:** Fly from Vienna to Lyon.
**Day 26-27:** Visit Lyon for 2 days.

Query:
You plan to visit 10 European cities for 25 days in total. You only take direct flights to commute between cities. You plan to stay in Berlin for 4 days. You have to attend a workshop in Berlin between day 12 and day 15. You would like to visit Prague for 2 days. You plan to
stay in Stuttgart for 5 days. You would like to visit Manchester for 3 days. You want to spend 2 days in Nice. You would like to visit Reykjavik for 2 days. You would like to visit Florence for 3 days. You plan to stay in Vilnius for 5 days. You would like to meet your
friends at Vilnius between day 15 and day 19 to tour together. You plan to stay in Oslo for 4 days. You would like to visit Dubrovnik for 4 days. You plan to visit relatives in Dubrovnik between day 1 and day 4.

Here are the cities that have direct flights:
from Reykjavik to Stuttgart, Manchester and Stuttgart, Nice and Berlin, Oslo and Prague, Stuttgart and Berlin, Manchester and Nice, Reykjavik and Oslo, Reykjavik and Prague, Manchester and Prague, Reykjavik and Berlin, Dubrovnik and Manchester, Manchester and Oslo,
Manchester and Berlin, Prague and Florence, Berlin and Vilnius, Dubrovnik and Oslo, Nice and Oslo, Berlin and Oslo, Nice and Reykjavik, Vilnius and Oslo.

Find a trip plan of visiting the cities for 25 days by taking direct flights to commute between them.

Incorrect plan in natural language:
SOLUTION: Here is the trip plan for visiting the 10 European cities for 25 days:

**Day 1-4:** Arriving in Dubrovnik and visit Dubrovnik for 4 days.

**Day 4:** Fly from Dubrovnik to Manchester.

**Day 4-7:** Visit Manchester for 3 days.

**Day 7:** Fly from Manchester to Stuttgart.

**Day 7-12:** Visit Stuttgart for 5 days.

**Day 12:** Fly from Stuttgart to Berlin.

**Day 12-15:** Visit Berlin for 4 days. (Attend workshop between day 12 and day 15)

**Day 15:** Fly from Berlin to Vilnius.

**Day 15-19:** Visit Vilnius for 5 days. (Meet friends between day 15 and day 19)

**Day 19:** Fly from Vilnius to Oslo via Berlin.

**Day 19-23:** Visit Oslo for 4 days.

**Day 23:** Fly from Oslo to Reykjavik.

**Day 23-25:** Visit Reykjavik for 2 days.

Incorrect plan in parsed format:
[['Dubrovnik', 4], ['Manchester', 4], ['Stuttgart', 6], ['Berlin', 4], ['Vilnius',
    5], ['Oslo', 5], ['Reykjavik', 3]]

Errors with the above plan:
Number of cities in plan is 7, expected 10

Give the corrected plan in natural language such that it can be parsed by the above
    python code.
Strictly adhere to the format shown in the examples below, without any extra
    information.

Corrected plan:

===============RESPONSE OF o1-preview==================
SOLUTION: Here is the trip plan for visiting the 10 European cities for 25 days:

**Day 1-4:** Arriving in Dubrovnik and visit Dubrovnik for 4 days. (Visit relatives
    between day 1 and day 4)

**Day 4:** Fly from Dubrovnik to Manchester.

**Day 4-6:** Visit Manchester for 3 days.

**Day 6:** Fly from Manchester to Nice.

**Day 6-7:** Visit Nice for 2 days.

**Day 7:** Fly from Nice to Reykjavik.

**Day 7-8:** Visit Reykjavik for 2 days.

**Day 8:** Fly from Reykjavik to Stuttgart.

**Day 8-10:** Visit Stuttgart for 3 days.

**Day 10:** Fly from Stuttgart to Prague.

**Day 10-11:** Visit Prague for 2 days.

**Day 11:** Fly from Prague to Berlin.

**Day 11-15:** Visit Berlin for 4 days. (Attend workshop between day 12 and day 15)

**Day 15:** Fly from Berlin to Vilnius.

**Day 15-19:** Visit Vilnius for 5 days. (Meet friends between day 15 and day 19)

**Day 19:** Fly from Vilnius to Oslo.

**Day 19-22:** Visit Oslo for 4 days.

**Day 22:** Fly from Oslo to Florence via Prague.

**Day 22-25:** Visit Florence for 3 days.

## Q  PROMPTS FOR CALENDAR SCHEDULING

### Q.1  FIRST ITERATION

You are an expert at scheduling meetings. You are given a few constraints on the
    existing schedule of each participant, the meeting duration, and possibly some
    preferences on the meeting time. Note there exists a solution that works with
    existing schedule of every
participant. Produce only one meeting time, strictly adhering to the format shown in
     the examples below, without providing any extra information. Here are a few
    example tasks and solutions:

TASK: You need to schedule a meeting for George, Steven, Aaron, Patrick and Cynthia
    for half an hour between the work hours of 9:00 to 17:00 on Monday.

Here are the existing schedules for everyone during the day:
Georgehas no meetings the whole day.
Steven is free the entire day.
Aaron has blocked their calendar on Monday during 9:00 to 10:00, 11:30 to 12:00,
    15:30 to 17:00;
Patrick has blocked their calendar on Monday during 9:00 to 9:30, 10:00 to 11:00,
    11:30 to 12:00, 12:30 to 14:00, 15:00 to 15:30;
Cynthia is busy on Monday during 9:00 to 9:30, 10:30 to 11:30, 12:30 to 14:30, 15:00
     to 16:30;

Aaron can not meet on Monday after 12:30. Find a time that works for everyone's
    schedule and constraints.
SOLUTION: Here is the proposed time: Monday, 12:00 - 12:30

TASK: You need to schedule a meeting for Elizabeth, Eugene, Nancy, Justin and Roy
    for half an hour between the work hours of 9:00 to 17:00 on Monday.

Here are the existing schedules for everyone during the day:
Elizabeth's calendar is wide open the entire day.
Eugene has blocked their calendar on Monday during 12:00 to 12:30, 13:30 to 14:00,
    15:00 to 16:00;
Nancy has meetings on Monday during 10:30 to 11:00, 12:00 to 13:00, 14:00 to 14:30,
    15:00 to 16:00;
Justin has meetings on Monday during 9:30 to 10:00, 10:30 to 11:30, 12:00 to 12:30,
    13:00 to 13:30, 14:30 to 17:00;
Roy is busy on Monday during 10:00 to 11:00, 11:30 to 12:30, 13:30 to 14:00, 14:30
    to 16:00;

Find a time that works for everyone's schedule and constraints.
SOLUTION: Here is the proposed time: Monday, 9:00 - 9:30

TASK: You need to schedule a meeting for Mason, Linda, Barbara, Roger and Lisa for
    half an hour between the work hours of 9:00 to 17:00 on Monday.

Here are the existing schedules for everyone during the day:
Masonhas no meetings the whole day.
Linda is busy on Monday during 9:30 to 11:00, 12:00 to 12:30, 13:00 to 13:30, 16:30
    to 17:00;
Barbara has blocked their calendar on Monday during 9:00 to 9:30, 12:00 to 13:00,
    15:30 to 16:00;
Roger has meetings on Monday during 9:00 to 9:30, 13:00 to 14:30, 15:00 to 17:00;
Lisa has blocked their calendar on Monday during 9:00 to 10:00, 10:30 to 11:30,
    12:00 to 12:30, 13:00 to 14:30, 15:00 to 17:00;

Mason would rather not meet on Monday after 13:30. Find a time that works for
    everyone's schedule and constraints.
SOLUTION: Here is the proposed time: Monday, 11:30 - 12:00

TASK: You need to schedule a meeting for Thomas, Olivia, Doris, Amber and Ronald for
     half an hour between the work hours of 9:00 to 17:00 on Monday.

Here are the existing schedules for everyone during the day:
Thomas has meetings on Monday during 12:00 to 12:30, 16:30 to 17:00;
Olivia has blocked their calendar on Monday during 10:30 to 11:30, 13:00 to 14:00,
    16:00 to 17:00;
Doris is busy on Monday during 13:30 to 14:00, 16:30 to 17:00;
Amber is busy on Monday during 10:00 to 10:30, 12:00 to 13:00, 14:00 to 15:00, 15:30
     to 17:00;
Ronald has meetings on Monday during 9:30 to 11:30, 12:00 to 13:00, 13:30 to 14:00,
    15:00 to 15:30;

Ronald would rather not meet on Monday before 11:30. Find a time that works for
    everyone's schedule and constraints.
SOLUTION: Here is the proposed time: Monday, 11:30 - 12:00

TASK: You need to schedule a meeting for Patrick, Kathleen, Frank, Patricia and
    Carolyn for half an hour between the work hours of 9:00 to 17:00 on Monday.

Here are the existing schedules for everyone during the day:
Patrick is free the entire day.
Kathleen has meetings on Monday during 9:30 to 10:00, 15:00 to 15:30;
Frank has blocked their calendar on Monday during 11:00 to 11:30, 15:30 to 17:00;
Patricia is busy on Monday during 9:00 to 12:00, 14:30 to 15:00, 16:00 to 16:30;
Carolyn has meetings on Monday during 9:00 to 9:30, 11:00 to 12:30, 13:00 to 17:00;

Find a time that works for everyone's schedule and constraints.
SOLUTION: Here is the proposed time: Monday, 12:30 - 13:00

Query:
TASK: You need to schedule a meeting for Stephen, Elijah, William, Jeremy and
    Timothy for half an hour between the work hours of 9:00 to 17:00 on Monday.

Here are the existing schedules for everyone during the day:
Stephen is free the entire day.
Elijah is busy on Monday during 9:00 to 9:30, 12:30 to 13:00, 14:30 to 15:00, 16:00
    to 16:30;
William is busy on Monday during 9:30 to 10:00, 15:30 to 16:00;
Jeremy is busy on Monday during 9:00 to 9:30, 10:00 to 12:00, 13:00 to 15:00, 15:30
    to 17:00;
Timothy is busy on Monday during 10:00 to 10:30, 11:30 to 14:30, 15:30 to 16:00;

Find a time that works for everyone's schedule and constraints.
SOLUTION:

```
==============RESPONSE OF o1-mini=================
SOLUTION: Here is the proposed time: Monday, 15:00 - 15:30
```

