# OpenReview forum: "Planning in Strawberry Fields: Evaluating and Improving the Planning and Scheduling Capabilities of LRM o1"
_ICLR.cc/2025/Conference — ICLR 2025 Conference Withdrawn Submission_

### Official Review · Reviewer_Yuco · 2024-11-01

**Soundness:** 3
**Presentation:** 3
**Contribution:** 1
**Rating:** 3
**Confidence:** 3

**Summary:**

In this paper, the authors benchmarked the performance of the newly released o1-preview and o1-mini on a few planning and scheduling datasets, and further evaluate how o1 models can be combined with LLM-Modulo system to enhance the guarantees of the correctness.

**Strengths:**

- The paper is overall well-written and easy to follow.
- The paper tests the state-of-the-art model, the o1 series, and studies their performance on complex reasoning tasks, including planning and scheduling problems.

**Weaknesses:**

- While it is definitely important to understand the actual performance of the newly released o1 models not from the OpenAI team as an advertisement, the contribution seems limited. This paper does not propose a new dataset or a new method. The conclusion is also not surprising given the extremely large amount of work on LLM for planning that has already been published and even cited in the paper  [1, 2, 3]. And the new argument on the cost side seems obvious given the price already released by OpenAI.
- While the overall writing is easy to follow, the style does not follow the typical academic paper but is more like a blog post. Many non-declarative sentences are used.
- The paper does not follow the ICLR template format right now.

[1]. Valmeekam K, Marquez M, Olmo A, et al. Planbench: An extensible benchmark for evaluating large language models on planning and reasoning about change[J]. Advances in Neural Information Processing Systems, 2024, 36.

[2]. Kambhampati S, Valmeekam K, Guan L, et al. LLMs can't plan, but can help planning in LLM-modulo frameworks[J]. arXiv preprint arXiv:2402.01817, 2024.

[3]. Kambhampati S. Can large language models reason and plan?[J]. Annals of the New York Academy of Sciences, 2024, 1534(1): 15-18.

**Questions:**

Can you explain the significance of this paper? What is the main contribution besides evaluating a new series of models?

---

> ### Author Response · Authors · 2024-11-25
>
> > **Can you explain the significance of this paper? What is the main contribution besides evaluating a new series of models?**
>
> The significance of the paper is in terms of understanding the cost-benefit tradeoffs of LRMs like o1. By all accounts, o1 is not just a new model–but one that attempts to add significant reasoning components on top of a normal autoregressive LLM substrate. (We note that we have already included data about the performance of most newer autoregressive LLMs that came after the evaluation presented in [1] above–as *background*).
>
>
> Saying that the OpenAI’s official post makes clear the inference cost considerations maybe a little too simplistic. We don’t think anyone realized that the increase would be as severe as what we actually found in our tests. Making the actual cost increase clear–especially when set in juxtaposition to the cost of solving with guarantees with external solvers–is crucial for us to get a proper understanding of what o1-style approaches offer.
> While it may seem “obvious” in retrospect, it was certainly not clear at the outset that LRM-Modulo will be an effective way to further improve the performance of LRM’s like o1. There really is no reason to believe that o1 will take the critiques from external verifiers, given that OpenAI claims that o1 makes up its own internal chains of thought.
>
>
> Overall, we believe that the reviewer may be unreasonably harsh on our very systematic external study on o1. Afterall, most “LLMs are Zero Shot <XXX>” papers essentially present problems to various LLMs and report their findings. In other words, studies like these are already known to be useful for the research community.
>
>
> Please see also the global comment.

---

> > ### Comment · Reviewer_Yuco · 2024-11-25
> >
> > Thank you for your response and clarification.
> >
> > My sincere apologies if the authors are from OpenAI, in which case, this work may provide valuable insights for improving o1-model training, particularly for preserving planning capabilities and integration with the LLM-modulo framework, its broader applicability appears limited for most researchers in the ICLR community who can only use these models rather than develop them. For the cost part, while it is good to see the actual cost on the current dataset, it is still very case-by-case and researchers still need to test on their own scenario first rather than directly use the conclusion here.
> > Overall, after reading other reviewers' comments, I think these perspectives seem to align with the concerns raised by all other reviewers, and I have updated my score accordingly.

---

### Official Review · Reviewer_SF4P · 2024-11-02

**Soundness:** 2
**Presentation:** 1
**Contribution:** 2
**Rating:** 3
**Confidence:** 4

**Summary:**

The paper examines the capabilities of large reasoning models (LRMs) for planning and scheduling tasks.  It evaluates two LRMs, o1-preview and o1-mini, on benchmarks (for example PlanBench, Natural Plan benchmark and Travel Planning benchmark) and compares their performance to previous results.  The paper also notes the steep inference cost of LRMs and discusses the tradeoffs between using LLMs versus LRMs, arguing that in some cases an LLM-Modulo approach may be significantly cheaper than o1 models for comparative performance, and with guarantees.

**Strengths:**

+ The paper demonstrates strong integration with existing framework (LLM-Modulo), leading to cost reductions in the evaluation process.
+ The authors provide thorough documentation of the prompts and their design.
+ The research contributes to our understanding of model capabilities and limitations, which is important for the field’s development and for improving model robustness.

**Weaknesses:**

The paper’s introduction could benefit from a clearer motivation. The decision to produce a new benchmark evaluation for each model release raises a question: does this paper bring a unique perspective to the evaluation landscape, or could the model creators be responsible for performance assessments of "established" benchmarks? Given that no new methodology or dataset is introduced, the contribution could appear limited to evaluating 1-2 additional models.

- Structural and Organizational Observations

The structure of the paper could be improved for clarity. For example, the results are presented as early as the related work section and continue sporadically throughout, which may make it challenging for readers to follow the flow of the argument. Additionally, the section on related work doesn’t provide a comprehensive view of the current state of the field, which might leave readers without a solid foundation in the broader research context.

Some terms, like "IPC," are introduced in the introduction but explained only in later sections. Defining these terms earlier could help readers understand the content more easily. Additionally, while the appendix includes sample prompts, adding a detailed example or task illustration in the main text could make the methodology and objectives clearer to readers.

- Evaluation and Methodological Details

The evaluation process might benefit from further clarification, especially concerning the assessment of LL(R)M solutions. For instance, it’s not entirely clear whether the evaluation is based on finding an optimal solution path or whether any correct path is acceptable, and if only one solution path is considered correct per example. Further qualitative analysis of the model’s errors would also be helpful—such as identifying if mistakes are common at the start of a task, or more likely toward the end. Discussing typical solution lengths would add another informative layer to the evaluation.

The evaluation protocols could be explained in more detail. The authors mention using a fixed temperature of 1 (this is fixed temperature for o1 models), but it would still help to run each instance multiple times (e.g., 3-5 runs) to check for any variation in results. This approach would make the findings more reliable. Given the focus on evaluation costs, applying this method to a smaller subset could balance accuracy with resource use.

- Contributions and Added Insights

The paper’s introduction suggests new metrics will be introduced, but no specific metrics are proposed. While the authors argue that “new approaches to measuring LRM reasoning capabilities must consider efficiency, cost, and guarantees,” this point is broadly applicable to all LLMs rather than being specific to LRMs

In the section evaluating Sokoban, some additional details could improve transparency. For instance, providing information about the board distributions (e.g., sizes and number of boxes) and specific results across different board configurations could give readers a clearer picture of the model’s performance. The authors report that o1-mini achieved a 10.9% success rate on Sokoban, while Llama3.1-405B achieved 0%; however, further analysis on this performance gap would be helpful.

- Style and Presentation Suggestions

The paper’s tone in certain sections could be more formal. Phrasing that is occasionally colloquial, along with exclamation marks, can detract from the overall scientific rigor expected in research papers. Additionally, consistency in the figures could improve readability—specifically, in Figure 1, the lines on the left plot are hard to see, and the color scheme is inconsistent between plots, which may make interpretation challenging.

Currently, the absence of code limits the ability of readers to reproduce the findings. Additionally, a discussion on limitations and broader impacts would provide helpful context regarding the scope and implications of the results.

- Cost Analysis Considerations

The cost analysis presented in Table 4 could be better suited to the appendix. Since model costs can vary over time, the information may quickly become outdated.

**Questions:**

Q1. How is the solution generated by the LLM evaluated? Is it based on finding the optimal path, or is any correct path acceptable? Also, is there only one correct solution path for each example?

Q2. Is there any qualitative analysis of errors? For instance, when does the model typically make mistakes? Does it struggle to begin tasks correctly, or do errors occur more frequently toward the end of the solution? Additionally, what is the average length of the solutions?

Q3. Could you provide more detail on the evaluation protocols? The paper mentions a temperature setting of 1, but was any sampling used, given the non-deterministic nature of results? For example, was each instance evaluated multiple times (e.g., 3-5 runs) to capture potential variance?

Q4. Could you clarify the distribution details for the Sokoban tasks? Specifically, what are the board sizes and the number of boxes used, and are there specific results for each configuration?

Q5. The paper reports that on Sokoban, o1-mini achieved a 10.9% success rate, while Llama3.1 (405B) achieved 0%. Could you provide some analysis or discussion on why there is such a difference in performance?

Q6. For Gemini 1.5 Pro, were the parameters regarding safety settings configured correctly to produce reasonable answers?

Q7. Could you define "gaslighting" in this context? Are there relevant citations or sources that you could provide on this topic?

---

> ### Author Response · Authors · 2024-11-25
>
> > **does this paper bring a unique perspective to the evaluation landscape**
>
> The significance of the paper is in terms of understanding the cost-benefit tradeoffs of LRMs like o1. By all accounts, o1 is not just a new model–but one that attempts to add significant reasoning components on top of a normal autoregressive LLM substrate. We note that we have already included data about the performance of most newer autoregressive LLMs that came after the evaluation presented in [1]–as *background*.
>
> Please also see the global comment.
>
>
> > **Q1. How is the solution generated by the LLM evaluated? Is it based on finding the optimal path, or is any correct path acceptable? Also, is there only one correct solution path for each example?**
>
> For planning domains, we use VAL [2] (similar to PlanBench), which essentially can be seen as a simulator where the plan is executed step-by-step from the initial state and if all the goals are satisfied in the resulting state, the plan is deemed correct. We do not check for plan optimality. As long as the plan is satisficing, as in it reaches the goal state, it is considered correct. Further, for all our instances there can be multiple solutions.
>
> > **Q2. Is there any qualitative analysis of errors? For instance, when does the model typically make mistakes? Does it struggle to begin tasks correctly, or do errors occur more frequently toward the end of the solution? Additionally, what is the average length of the solutions?**
>
> For a closer look at the kind of errors the models make, we checked what kinds of invalid plans o1 generates. The invalidity of a plan could be either due to an action being inexecutable or due to the plan not reaching the desired goal state after execution. As shown in the tables below, majority of the invalid plans of o1-preview and o1-mini are inexecutable.
>
> Results of o1-preview (all are zero-shot)
>
> | # of Instances | Domain                            | # of Valid plans | # of Inexecutable plans | # of Non goal reaching plans |
> |----------------|-----------------------------------|------------------|-------------------------|------------------------------|
> | 600            | Blocksworld                       | 587              | 12                      | 1                            |
> | 600            | Mystery Blocksworld               | 317              | 202                     | 81                           |
> | 600            | Randomized Obfuscated Blocksworld | 224              | 249                     | 127                          |
> | 200            | Logistics                         | 188              | 12                      | 0                            |
>
>
> Results of o1-mini (all are zero-shot)
>
> | # of Instances | Domain                            | # of Valid plans | # of Inexecutable plans | # of Non goal reaching plans |
> |----------------|-----------------------------------|------------------|-------------------------|------------------------------|
> | 600            | Blocksworld                       | 340              | 237                     | 23                           |
> | 600            | Mystery Blocksworld               | 115              | 346                     | 140                          |
> | 600            | Randomized Obfuscated Blocksworld | 21               | 114                     | 465                          |
>
>
> The distribution of problems for Blocksworld (the original and obfuscated versions) and Logistics is the same as that of PlanBench. For Blocksworld, the average optimal length of plans is 7.14 while for logistics it is 20.14.
>
> We will update the paper with the above details and discussion.
>
> > **Q3. Could you provide more detail on the evaluation protocols? The paper mentions a temperature setting of 1, but was any sampling used, given the non-deterministic nature of results? For example, was each instance evaluated multiple times (e.g., 3-5 runs) to capture potential variance?**
>
> Due to cost considerations (as described in Section 3.3), we had a single run for all our experiments.
>
> > **Q4. Could you clarify the distribution details for the Sokoban tasks? Specifically, what are the board sizes and the number of boxes used, and are there specific results for each configuration?**
>
> The board sizes range from 4x4 to 10x10 and the number of boxes range from 1 to 4. With respect to the number of boxes, there are 22 1-box instances, 15 2-box, 12 3-box and 6 4-box instances. All of the instances solved by o1 and o1-modulo are either 1-box or 2-box instances. We will add this information and discussion as part of the paper.

---

> ### Author Response · Authors · 2024-11-25
>
> > **Q5. The paper reports that on Sokoban, o1-mini achieved a 10.9% success rate, while Llama3.1 (405B) achieved 0%. Could you provide some analysis or discussion on why there is such a difference in performance?**
>
> As explained in the paper, we believe that the o1 models are fundamentally different from those of vanilla LLMs like LLaMA during both the pre-training phase and at inference time. We provide a detailed account of our speculations on these o1 models in Appendix B. Verifying our speculations is unfortunately currently infeasible given the closed nature of OpenAI and their efforts to not leak anything about the model (including not allowing users to see the reasoning tokens generated by the model).
>
> > **Q6. For Gemini 1.5 Pro, were the parameters regarding safety settings configured correctly to produce reasonable answers?**
>
> Our aim was to evaluate the latest set of LLMs as is, so we used the default safety settings for the Gemini models. It blocks content with medium or high probability of being unsafe across any filter.
>
> > **Q7. Could you define "gaslighting" in this context? Are there relevant citations or sources that you could provide on this topic?**
>
> Gaslighting is a colloquial term that generally refers to manipulating someone in a way that causes them to either doubt themselves or, at the very least, creates confusion in their mind. In the case of o1, when it outputs an incorrect answer it also provides a creative but nonsensical justification. This might create confusion in the user’s mind and possibly even lead to the user doubting their own knowledge of the domain.
>
>
> [1] Valmeekam K, Marquez M, Olmo A, et al. Planbench: An extensible benchmark for evaluating large language models on planning and reasoning about change[J]. Advances in Neural Information Processing Systems, 2024, 36.
>
> [2] Howey, R., Long, D., & Fox, M. (2004, November). VAL: Automatic plan validation, continuous effects and mixed initiative planning using PDDL. In 16th IEEE International Conference on Tools with Artificial Intelligence (pp. 294-301). IEEE.

---

> > ### Comment · Reviewer_SF4P · 2024-11-25
> >
> > Dear Authors,
> >
> > Thank you for your responses. The main takeaways from your research need to be more clearly articulated to help readers understand your contribution. Additionally, I have concerns about the writing style, which reads more like a blog post than an academic paper. As another reviewer also noticed, there are many non-declarative sentences throughout the text, which should be avoided in academic writing. I recommend adopting a more formal academic tone.

---

### Official Review · Reviewer_5DyR · 2024-11-04

**Soundness:** 2
**Presentation:** 2
**Contribution:** 3
**Rating:** 5
**Confidence:** 3

**Summary:**

OpenAI's o1 (Strawberry) represents a shift from traditional LLMs to Large Reasoning Models (LRMs). The paper evaluates the o1 family (including o1-preview and o1-mini) on planning and scheduling capabilities using established benchmarks like PlanBench, TravelPlanner, and graph coloring problems. The study shows that while o1 models outperform previous LLMs, they result in higher inference costs and lack guarantees for output accuracy. The study also explores methods to enhance reliability through integration with RM-Modulo framework as external verification.

**Strengths:**

- The paper focuses on important practical considerations such as reliability and efficiency of LRMs
- The paper utilizes established benchmarks while extending them for more rigorous testing and takes a systematic approach to evaluating both basic and complex planning scenarios. The paper presents a thorough evaluation with performance metrics, such as accuracy and inference time clearly reported and supported with comparative data.
- Proposed the integration of LRM-Modulo framework for enhanced reliability and potentially efficiency when coupled with the mini version

**Weaknesses:**

- The reliability of the proposed LLM-Modulo integration when paired with the o1-mini model is unclear to me. While we can observe improved o1-mini performance, in cases such as Blockworld (hard), Sokoban, Trip Planning, o1-mini still significantly lags behind o1 performance. As the inference efficiency is an important consideration, it is unclear whether the LLM-Modulo can reliably deliver better efficiency while preserving the quality of the response.
- The studied problem is important but the novelty of the paper's proposed solution is limited.

**Questions:**

- Can the authors please explain the applicability and scalability of LLM-Modulo outside of the tested benchmarks, e.g., for practical usage? What is the minimum requirement of a task for the LLM-Modulo integration to be implemented?
- What are the potential ways to make the high inference cost of o1 more manageable in practical settings, besides using o1-mini as a replacement?

---

> ### Author Response · Authors · 2024-11-25
>
> > **Can the authors please explain the applicability and scalability of LLM-Modulo outside of the tested benchmarks, e.g., for practical usage? What is the minimum requirement of a task for the LLM-Modulo integration to be implemented?**
>
> LLM-Modulo seems to be a general framework for using LLMs as smart generators, whose solution guesses are checked/critiqued by external verifiers–either classical/symbolic or learned. The original paper points out that the verifiers/critics can themselves be teased out from LLMs with partial help from humans or other model/code critics (in a sort of recursive LLM-Modulo) framework.
> This general idea of using LLMs as smart generators whose guesses are validated in an iterative loop has also been applied in such systems as Alpha Proof and most LLM-based code-synthesis systems. In the latter case, the (partial) verification is done with the help of unit tests and code interpreters such as Python interpreters.
>
>
> > **What are the potential ways to make the high inference cost of o1 more manageable in practical settings, besides using o1-mini as a replacement?**
>
> Regarding managing cost of o1, the classical deliberation scheduling techniques may be of relevance if only o1 actually allowed control to users on the amount of compute done at the inference stage. Given that they don’t allow that (other than the discrete o1 vs. o1-preview choice), the other alternatives are to use LLM/LRM-Modulo techniques either on cheaper LRMs or LLMs. For example, from Line 511 to 516, we do mention that using plain old LLMs even a very small one like 4o-mini performs comparatively to o1-preview and slightly better than o1-mini.

---

### Official Review · Reviewer_Y1qU · 2024-11-04

**Soundness:** 2
**Presentation:** 2
**Contribution:** 1
**Rating:** 1
**Confidence:** 4

**Summary:**

This paper evaluates the planning capabilities of OpenAI's latest o1 series models on the PlanBench and TravelPlanner benchmarks, and proposes the use of external verifiers to assist the models in planning.

**Strengths:**

The paper is highly timely and helps readers understand the specific performance of the o1 series models on the two benchmarks.

**Weaknesses:**

The paper's contribution to the community is relatively limited, as it does not provide any new insights.

The first key observation ``while o1 does seem to offer significant improvements over autoregressive
LLMs, this comes at a steep inference cost'' is already highlighted in OpenAI's official post (https://openai.com/index/learning-to-reason-with-llms/).

The second key contribution in Section 4 — that ``combining o1 models with external verifiers–in a so-called LRM-Modulo system–guarantees the correctness of the combined system’s output while further improving performance.'' — is also not new; it simply tests o1 models within the established LLM-Modulo Frameworks [1], without providing any new insights.

[1] LLMs Can't Plan, But Can Help Planning in LLM-Modulo Frameworks

**Questions:**

No questions.

**Details Of Ethics Concerns:**

No concerns

---

> ### Author Response · Authors · 2024-11-25
>
> > **The first key observation ``while o1 does seem to offer significant improvements over autoregressive LLMs, this comes at a steep inference cost'' is already highlighted in OpenAI's official post (https://openai.com/index/learning-to-reason-with-llms/).**
>
> We are a little surprised at your statement that “The paper is highly timely and helps readers understand the specific performance of the o1 series models on the two benchmarks.” and the “strong reject” rating!
> Saying that the OpenAI’s official post makes clear the inference cost considerations may be a little too simplistic. We don’t think anyone realized that the increase would be as severe as what we actually found in our tests. Making the actual cost increase clear–especially when set in juxtaposition to the cost of solving with guarantees with external solvers–is crucial for us to get a proper understanding of what o1-style approaches offer.
>
> > **The second key contribution in Section 4 — that ``combining o1 models with external verifiers–in a so-called LRM-Modulo system–guarantees the correctness of the combined system’s output while further improving performance.'' — is also not new; it simply tests o1 models within the established LLM-Modulo Frameworks [1], without providing any new insights.**
>
> While it may seem “obvious” in retrospect, it was certainly not clear at the outset that LRM-Modulo will be an effective way to further improve the performance of LRM’s like o1. There really is no reason to believe that o1 will take the critiques from external verifiers, given that OpenAI claims that o1 makes up its own internal chains of thought.
>
>
> Overall, we believe that the reviewer may be unreasonably harsh on our very systematic external study on o1. Afterall, most “LLMs are Zero Shot <XXX>” papers essentially present problems to various LLMs and report their findings. In other words, studies like these are already known to be useful for the research community.

---

> ### Comment · Reviewer_Y1qU · 2024-11-25
> **Reply to the authors**
>
> Thank you for the rebuttal first!
>
> > We don’t think anyone realized that the increase would be as severe as what we actually found in our tests.
>
> Regarding the ``steep inference cost'' of the o1 models, could you further elaborate on the core findings from your tests and explain why these findings were not easily found?
>
> > There really is no reason to believe that o1 will take the critiques from external verifiers, given that OpenAI claims that o1 makes up its own internal chains of thought.
>
> I can't quite grasp the strict causal relationship in this argument.
>
> > Afterall, most “LLMs are Zero Shot <XXX>” papers essentially present problems to various LLMs and report their findings.
>
> Comparing with papers like [1], I think the core difference here lies not in whether the approach is similar but in whether the results are surprising enough to the entire research community.
>
> [1] Large Language Models are Zero-Shot Reasoners
>
> Best,
> Reviewer Y1qU

---

### Author Response · Authors · 2024-11-25
**Global Comment**

We were a little surprised at some of the reviews. The reviewers agree that our evaluation of o1 model is important (e.g. Y1qU “The paper is highly timely and helps readers understand the specific performance of the o1 series models on the two benchmarks.”) and yet seem to have reservations about the relevance of the paper for ICLR. This is quite puzzling. Afterall, most “LLMs are Zero Shot <XXX>” papers essentially present problems to various LLMs and report their findings. In other words, studies like these are already known to be useful for the research community. Given this,  we believe that the reviewers may be unreasonably harsh on our very systematic external study on o1.


We note that being a model that is quite distinct from the prior auto-regressive LLM models, there was considerable mystery/confusion about o1’s performance and limitations on reasoning problems. Our systematic study–which starts with existing benchmarks, but also extends them to better evaluate o1 in places where it saturates the existing benchmarks, and also carefully analyzes the cost/accuracy tradeoffs–will be a valuable contribution in understanding this new breed of models.


The analysis of cost/accuracy tradeoffs are particularly relevant. While the reviewers point out that the o1 announcement did say that it will have costlier inference phase, it is rather a stretch to say that the community understood how significant the cost increases for o1 were compared to the SOTA LLMs. These substantial increases, coupled with the fact that the model still has low accuracy in multiple scenarios (longer plans, unsolvable instances, domains such as travel planning with multiple constraints on schedules)--brings to fore the fact that LRMs may compare unfavorably to classical solvers–that, as we show, seem to solve the instances in a fraction of the time with guarantees; or even autoregressive LLMs coupled with external verifiers in frameworks such as LLM-Modulo.  Finally, we also pointed out that techniques like LLM-Modulo can indeed be adapted to LRMs, and further improve their accuracy with guarantees.


In light of these, we think the reviewers may be too harsh in evaluating the contributions of the paper.

---

### Note · Authors · 2024-12-03

I have read and agree with the venue's withdrawal policy on behalf of myself and my co-authors.